# Mutant KRAS-activated circATXN7 fosters tumor immunoescape by sensitizing tumor-specific T cells to activation-induced cell death

Chi Zhou [1,2,3,11], Wenxin Li[4,5,6,11], Zhenxing Liang [4,5,6,11], Xianrui Wu [4,5,6,11], Sijing Cheng[5], Jianhong Peng[1,2,3], Kaixuan Zeng [7], Weihao Li[1,2,3], Ping Lan [4,5,6], Xin Yang[4,5,6], Li Xiong[4,5,6], Ziwei Zeng [4,5,6], Xiaobin Zheng[4,5,6], Liang Huang[4,5,6], Wenhua Fan[1,2,3], Zhanzhen Liu[4,5,6], Yue Xing [8,9,12] ✉, Liang Kang [4,5,6,12] ✉ & Huashan Liu [4,5,6,10,12] ✉

Mutant KRAS (KRAS[MUT]) is often exploited by cancers to shape tumor immunity, but the underlying mechanisms are not fully understood. Here we report that tumor-specific cytotoxic T lymphocytes (CTLs) from KRAS[MUT] cancers are sensitive to activation-induced cell death (AICD). circATXN7, an NF-κB-interacting circular RNA, governs T cell sensitivity to AICD by inactivating NF-κB. Mechanistically, histone lactylation derived from KRAS[MUT] tumor cell-produced lactic acid directly activates transcription of circATXN7, which binds to NF-κB p65 subunit and masks the p65 nuclear localization signal motif, thereby sequestering it in the cytoplasm. Clinically, circATXN7 upregulation in tumor-specific CTLs correlates with adverse clinical outcomes and immunotherapeutic resistance. Genetic ablation of circAtxn7 in CD8[+] T cells leads to mutant-selective tumor inhibition, while also increases anti-PD1 efficacy in multiple tumor models in female mice. Furthermore, targeting circATXN7 in adoptively transferred tumor-reactive CTLs improves their antitumor activities. These findings provide insight into how lymphocyte-expressed circRNAs contribute to T-cell fate decisions and anticancer immunotherapies.

The human KRAS protein is both friend and foe; the non-mutated form is indispensable in diverse physiological processes, whereas the mutated versions directly underlie multistep processes of tumorigenesis and progression in ~30% of all cancers. Targeting KRAS is considered one of the optimal strategies to combat KRAS-driven tumors and improve advanced cancer patients' outcomes[1,2]. Despite advances in KRAS inhibitors, decades of efforts hitherto did not bring them to the clinic[3,4]. Recent studies revealed that mutant KRAS could be exploited by cancers to orchestrate an immune-suppressive tumor microenvironment (TME)[5,6]. The Cancer Genome Atlas also indicated KRAS mutant colorectal cancer (CRC) are closely associated with

decreased immune infiltration and reactivity[7]. In addition, KRAS inhibition endowed tumors with a remarkable increase in anti-tumor immunity[8]. Therefore, KRAS mutant tumors are especially immune-excluded, and therapeutic approaches aimed at activating anti-tumor immune program might be essential to eliminate the disease.

Cancer pathologies are often orchestrated by various metabolites, and KRAS mutant tumors are especially exposed to dramatically increased levels of lactic acid[9,10]. Cancer-generated lactic acid endows malignancies with an acidic TME, and also acts as a primary carbon fuel source and signaling molecule involved in oncogenic pathways[11,12]. Current researches have also yielded evidence that lactic acid in the

TME was an impediment towards providing an effective anti-tumor immunity[13]. In this respect, tumor-derived lactic acid was found to take effects on tumor-associated macrophages[14,15], regulatory T cells[16], myeloid-derived suppressor cells[17], natural killer cells[18], or dendritic cells[19]. In particular, Kreutz and colleagues pointed to an impact of lactic acid on cytolytic T lymphocytes (CTLs)[20], which directly identify and destroy nascent tumor cells during cancer immunosurveillance. Our previous study significantly advanced our understanding for the involvement of lactic acid in CTL fate decisions and subsequent support for tumor progress[9]. These insights highlighted an intense engagement between lactic acid and CTLs, but the intracellular mechanism of lactic acid action in CTLs remains poorly defined.

Activation-induced cell death (AICD), firstly described in 1987, has been characterized as a mechanistic link with immunological homeostasis[21,22]. Under physiological conditions, AICD is able to eradicate activated T lymphocytes presumed to be no longer required[23]. Abnormality in AICD was discovered in diverse pathological situations, such as viral infection, inflammatory and autoimmune disorders[24–26]. In the context of many cancer types, AICD deregulation was also frequently identified[27–29]. Aberrant AICD of tumor-specific CTLs can be used by cancers to evade immune elimination[28], which accounts for the paradoxical fact that, although the patients mount a specific T cell response against neoplasm, these CTLs fail to control the disease. It is now widely understood that AICD is of much value to decipher cancer pathologies as well as present prognostic insights, or even develop alternative treatments for cancer patients. Along this line, we previously found mutant KRAS-expressing CRC cells exploited tumor-derived lactic acid to sensitize tumor-specific CTLs to AICD, thereby fostering tumor immune escape and immunotherapy resistance[9]. Multiple molecular players, including mitochondrio-nuclear translocation of AIF[30], CD158 receptor[31], or NKILA[28], were identified to participate in an abnormal sensitivity of tumor-specific CTLs to AICD. Despite this knowledge, how lactic acid reprograms AICD of tumor-specific CTLs warrants under further investigation.

Circular RNAs (circRNAs) emerge as a unique class of RNA molecules characterized by their covalently closed ring structure. The interest in studying circRNAs is raised because of several peculiar features, such as evolutionary conservation and tissue-specific expression, but above all, because their deregulated expression was linked to many pathological conditions, particularly cancers[32,33]. Mounting data suggest these molecules are of potential clinical relevance and utility[34,35]. Notably, circRNAs have been identified to be participants in the regulatory networks of tumor immunity[36]. Wang and colleagues demonstrated overexpression of hsa_circ_0020397 in CRC cells could promote the upregulation of PD-L1 by binding and inhibiting miR-138 expression, thereby resulting in tumor immune escape[37]. Furthermore, there is evidence of a correlation between circRNAs and immune cell infiltration in several cancers[38–40]. Recently, a study by Ye et al. identified circRNA profiles and regulatory networks in melanoma patients treated with immune checkpoint blockades, highlighting the clinical application potential of circRNAs as predictive biomarkers for immunotherapeutic efficacy[41]. These advances underscored the link between circRNAs and cancer immunology, yet knowledge of the role played by circRNAs and the mechanism of circRNAs' action in CTLs is limited.

Here, we show that tumor-mediated AICD can be exploited by oncogenic KRAS to dictate the formation of an immune-suppressive tumor microenvironment. In the tumor-specific CTLs of KRAS$^{MUT}$ CRC, histone lactylation turns on the transcription of circATXN7, an NF-κB-interacting circular RNA. The upregulation of circATXN7 increases the sensitivity to AICD of tumor-specific CTLs by binding to NF-κB p65 subunit and masking the p65 nuclear localization signal motif, thereby sequestering it in the cytoplasm. In KRAS$^{MUT}$ tumors-bearing mice, genetic ablation of circAtxn7 in CD8$^+$ T cells leads to mutant-selective tumor inhibition, while also increases anti-PD1 efficacy. Administering tumor-reactive CTLs with circATXN7 knockdown effectively suppresses tumor growth by improving CTL accumulation in CRC patient-derived xenografts in NOD.SCID mice. Clinically, increased circATXN7 in tumor-specific CTLs is associated with adverse clinical outcomes and immunotherapeutic resistance. Together, our findings highlight the importance of circular RNAs in T cell fate decisions and suggest engineering them in T cells represents an exploitable approach for anticancer immunotherapies.

## Results

### Tumor-mediated AICD in KRAS$^{MUT}$ CRC

We previously found an inverse association between mutant KRAS and cytotoxic CD8$^+$ T cell (CTL) tumor infiltrate in stage I–III colorectal cancer (CRC) from the Sixth Affiliated Hospital of Sun Yat-sen University (SYSU-6thAH)[9], which was confirmed herein using a different cohort from Sun Yat-sen University Cancer Center (SYSUCC) containing 101 patients with stage IV CRC (Supplementary Fig. 1A). This work further assessed the prognostic value of CTL tumor infiltrate. In SYSU-6thAH cohort, Kaplan–Meier survival curve analysis found KRAS$^{MUT}$ patients with high versus low CTL-abundance had prolonged disease-free survival (DFS), an effect which was not seen in KRAS$^{WT}$ patients (Supplementary Fig. 1B). Likewise, progression-free survival (PFS) was significantly longer in KRAS$^{MUT}$ patients with high versus low CTL-abundance in the SYSUCC cohort (Supplementary Fig. 1D). Overall survival (OS) analysis showed similar results in the two independent cohorts (Supplementary Fig. 1C, E). Of the patients with low CTL-abundance, KRAS$^{MUT}$ indicated poor patient prognosis (Supplementary Fig. 2A, B), whereas KRAS$^{WT}$ versus KRAS$^{MUT}$ patients displayed comparable prognosis in those with high CTL-abundance (Supplementary Fig. 2A, B). Furthermore, KRAS$^{MUT}$ patients with high CTL-infiltrated tumors did not show a survival disadvantage in comparison with KRAS$^{WT}$ patients (Supplementary Fig. 2C). These prognostic findings suggested the oncogenic effects of the KRAS$^{MUT}$ were related to its capacity to elicit poor immunity.

Our previous findings ascribed the decreased CTLs occurred in KRAS$^{MUT}$ stage I-III CRC to the increased susceptibility to tumor-mediated activation-induced cell death (AICD) of tumor-specific CTLs[9]. A similar phenomenon was observed in a different patient cohort with stage IV CRC (Fig. 1A, Supplementary Fig. 2D). Considering the role of TCR engagement with the MHC-antigen complex, we further tested its contribution to tumor-mediated AICD in KRAS$^{MUT}$ CRC. To this end, CD8$^+$ T cells were activated by autologous dendritic cells (DCs) pulsed with autologous tumor-lysate or CEA peptide (Fig. 1B). Re-stimulation with anti-CD3 or coculturing with autologous tumor cells led to substantial apoptosis in the tumor-antigen-activated CTLs primed by autologous tumor-lysate-pulsed DCs for 6 days, but not those primed for 1 day (Fig. 1C, Supplementary Fig. 2E). In parallel, re-stimulation with anti-CD3 or CEA-loaded T2 cells, a human HLA-A2$^+$ hybridoma cell line used for antigen-specific cytotoxic assays, triggered significant apoptosis of the day-6 CEA-specific CTLs (Fig. 1D, Supplementary Fig. 2F). Furthermore, anti-HLA class I blocking antibodies effectively eliminated the apoptosis of the CTLs induced by coculturing with autologous tumor cells or CEA-loaded T2 cells (Fig. 1E, F). These findings suggested autologous tumor cells elicited AICD in activated CTLs from KRAS$^{MUT}$ tumors through repeated TCR stimulation.

The above findings showed that ACID was significantly increased in T cells exposed to CEA antigen. Yet it is unclear whether the increase in ACID is a CEA-specific mechanism. To address this, the link between AICD sensitivity and CEA expression was further analyzed, and results demonstrated the AICD sensitivity had no significant correlation with CEA expression levels (Fig. 1G). These findings suggested that the increase in AICD in KRAS$^{MUT}$ tumors might be independent of CEA expression. To further confirm this, tumor-specific CTLs were purified from CEA positive and negative expressing tumors using anti-MUC1

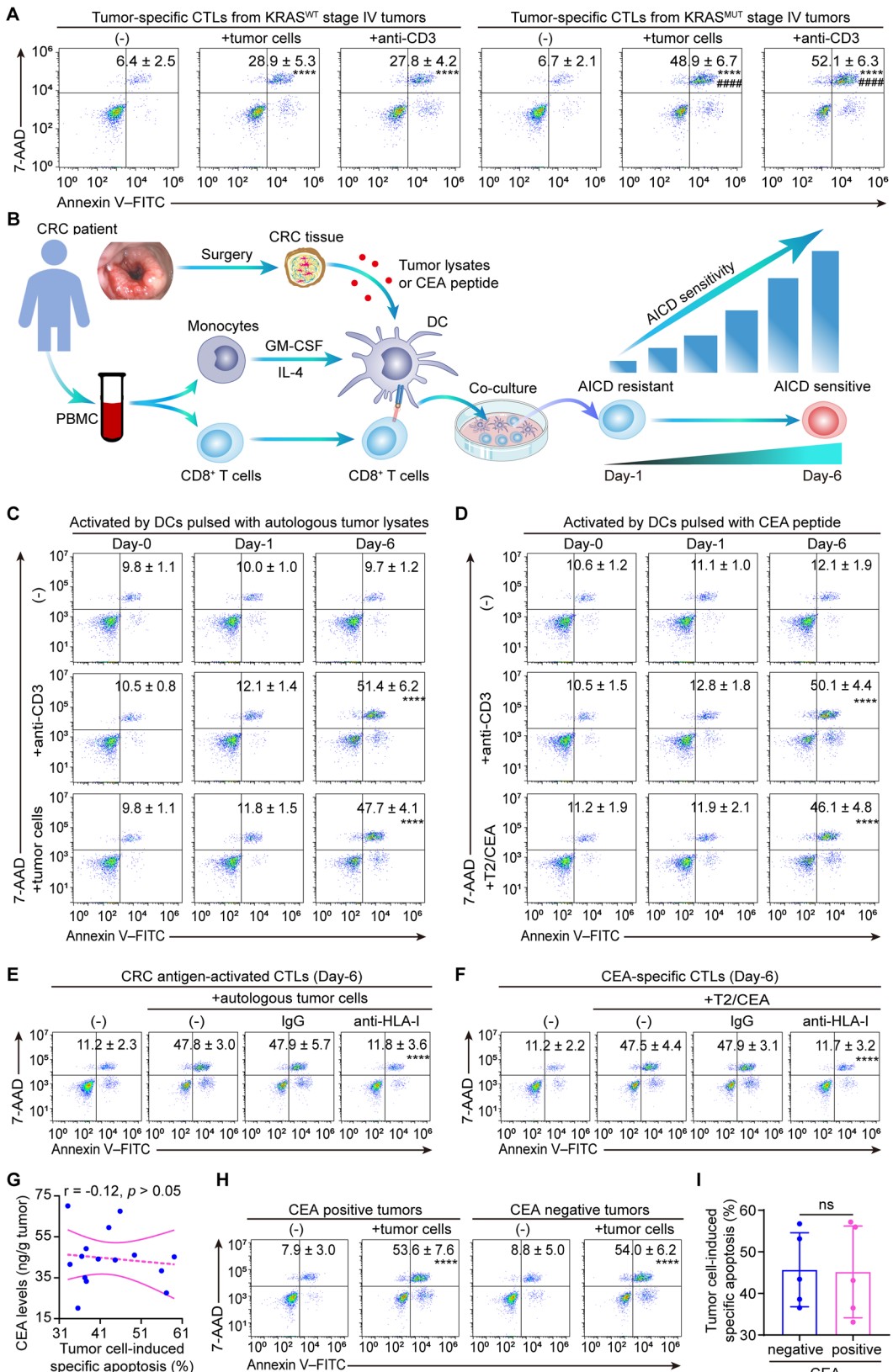

tetramer as described previously[42]. After coculturing with autologous tumor cells, comparable apoptosis was found in the tumor-specific CTLs from CEA-positive versus negative expressing tumors (Fig. 1H, I). Together, these results indicated the increase in AICD in KRAS^MUT tumors was independent of CEA expression. In addition to AICD increase in KRAS^MUT tumors, we found that tumor-specific CTLs from

KRAS^MUT versus KRAS^WT tumors exhibited a significant increase in the expression of perforin and CD107a (Supplementary Fig. 2G), markers associated with cytotoxic activity. Yet the markers of exhaustion (PD1 and TIGIT) and activation (CD25 and CD69) were comparably expressed between the two groups (Supplementary Fig. 2H). These findings suggested the increase in ACID was not correlated with differentiation

**Fig. 1 | Tumor-mediated AICD in KRAS^MUT versus KRAS^WT CRC. A** Tumor-specific CTLs were freshly isolated from KRAS^MUT versus KRAS^WT stage IV CRCs. Apoptosis of CTLs induced by autologous tumor cells or anti-CD3 ($n = 5$ samples; ****$p \leq 0.0001$ compared with the untreated CTLs by one-way ANOVA; ####$p \leq 0.0001$ compared with KRAS^WT tumors-derived CTLs with the indicated treatments by two-tailed Student's $t$-test). **B** Scheme of the induction of tumor-antigen-activated CTLs and AICD sensitivity switch during T cell activation. CRC patients-derived peripheral CD8^+ T cells were activated by autologous DCs pulsed with tumor lysate (**C**, **E**), or CEA peptide (**D**, **F**) for the indicated number of days. **C** Apoptosis of CTLs induced by anti-CD3 or autologous tumor cells ($n = 4$ samples; ****$p \leq 0.0001$ compared with untreated day-6 CTLs by one-way ANOVA). **D** Apoptosis of CTLs induced by anti-CD3 or CEA-loaded T2 cells (T2/CEA) ($n = 4$ samples; ****$p \leq 0.0001$ compared with untreated day-6 CTLs by one-way ANOVA). **E** Apoptosis of the CRC antigen-activated CTLs induced by autologous tumor cells preincubated with anti-HLA-I or IgG ($n = 4$ samples; ****$p \leq 0.0001$ compared with IgG by one-way ANOVA). **F** Apoptosis of the CEA-specific CTLs induced by CEA-loaded T2 cells (T2/CEA) preincubated with anti-HLA-I or IgG ($n = 4$ samples; ****$p \leq 0.0001$ compared with IgG by one-way ANOVA). **G** Correlation between CEA levels and AICD sensitivity of tumor-specific CTLs in KRAS^MUT CRCs ($n = 15$ patients; tested by Pearman correlation). **H, I** Tumor-specific CTLs were isolated from CEA positive and negative expressing CRCs. **H** Representative plots showing autologous tumor cell-induced apoptosis of CTLs (****$p \leq 0.0001$ compared with the untreated CTLs by two-tailed Student's $t$-test). **I** Statistical comparison of tumor cell-induced AICD in CTLs ($n = 5$ samples; ns indicates $p > 0.05$ by two-tailed Student's $t$-test). Numerical values (mean ± SD) denote annexin V^+ cell percentages (**A, C−F, H**). Source data and exact $p$ values are provided as a Source Data file.

towards an exhausted subset, but appeared to indicate an impaired anti-tumor immunity.

## Histone lactylation-activated circATXN7 is upregulated in AICD-sensitive T cells

In light of our above results, we set out to understand how KRAS^MUT sensitized tumor-specific CTLs to AICD. We previously showed bountiful lactic acid in KRAS^MUT stage I-III tumors contributed to the AICD susceptibility via NF-κB inactivation[9]. Using a different patient cohort with stage IV CRC, a lactic acid production advantage was confirmed in KRAS^MUT tumors (Supplementary Fig. 3A). Ex vivo administration of lactic acid significantly increased the AICD sensitivity of tumor-specific CTLs from stage IV CRC (Supplementary Fig. 3B). Further investigation demonstrated CTLs from KRAS^MUT stage IV tumors had an obvious decrease in NF-κB activity (Supplementary Fig. 3C, D). Moreover, NF-κB inhibitors BAY or JSH-23 almost completely abrogated the ability of lactic acid to regulate AICD (Supplementary Fig. 3E). Together, these findings established the lactic acid/NF-κB/AICD axis in stage IV CRC. Yet how lactic acid regulates NF-κB/AICD axis remains unclear. Lactic acid can stimulate cells via the receptor GPR81[43,44] or enter cells via monocarboxylate transport 1 (MCT1)[45,46]. These findings inspired us to test the roles of GPR81 and MCT1 in lactic acid/NF-κB/AICD axis. Results showed MCT1 blockade by AZD3965 significantly reversed the effects of lactic acid on NF-κB/AICD axis, and its combo inhibition of GPR81 with 3-OBA was obviously better than AZD3965 alone (Supplementary Fig. 3F, G). To distinguish their contribution to the difference in NF-κB/AICD axis between KRAS^MUT versus KRAS^WT tumors, the expression levels of *MCT1* and *GPR81* were further assessed. Results found that *MCT1* had significantly higher expression abundance than *GPR81* in tumor-specific CTLs from both KRAS^MUT and KRAS^WT tumors (Supplementary Fig. 3H). Moreover, the expression levels of *MCT1*, but not *GPR81*, was positively associated with NF-κB activity (Supplementary Fig. 3I, J), and NF-κB activity correlated well with intracellular lactic acid concentration in tumor-specific CTLs of KRAS^MUT tumors (Supplementary Fig. 3K), but not in those of KRAS^WT tumors (Supplementary Fig. 3L−N). More importantly, as the key downstream element in lactic acid/GPR81 axis, cAMP and *TCF-1* in tumor-specific CTLs were well balanced between KRAS^MUT versus KRAS^WT tumors (Supplementary Fig. 3O, P). Taken together, these results suggested that MCT1-mediated lactic acid input, but not activating GPR81, contributed to the difference in NF-κB/AICD axis between KRAS^MUT versus KRAS^WT tumors.

To explore the mechanism underlying the difference in NF-κB/AICD axis between KRAS^MUT versus KRAS^WT tumors, several NF-κB signaling-related factors reported in literatures (Supplementary Table 1) were tested. Results found (Supplementary Fig. 4A−L) suggested the differential NF-κB/AICD axis between KRAS^MUT versus KRAS^WT tumors was not governed by the above-mentioned factors, but is likely to be controlled by other factors. CircRNAs, a subclass of endogenous non-coding RNAs, are implicated in numerous patho-physiological conditions, including adaptive immune responses. We next sought to explore whether circRNAs contributed to the lactic acid/NF-κB/AICD axis. To this end, we stimulated peripheral blood (PB) CD8^+ T cells with phytohaemagglutinin (PHA) for 18 h (Day-1) and then cultured them with IL-2 for an additional 5 days (Day-6) (Supplementary Fig. 5A). In line with our previous reports[9,25], re-stimulation with anti-CD3 led to substantial apoptosis of Day-6 T cells (AICD-sensitive), but not Day-1 T cells (AICD-resistant) (Supplementary Fig. 5B). We then preformed circRNA profile analysis in AICD-resistant T cells after lactic acid or vehicle treatment. After filtering differentially expressed circRNAs (fold change (FC) > 2 or <0.5 and false discovery rate (FDR) < 0.05), we identified 130 upregulated and 153 down-regulated circRNAs upon lactic acid treatment (Fig. 2A, B). We further tested the top 10 upregulated circRNA (ranked by *p*-value) expression in Day-6 versus Day-1 T cells (Fig. 2B). A consistent expression trend was seen in 4 circRNAs (circGSE1, circATXN7, circPOLD1, and circPRKAR18) (Supplementary Fig. 5C). Given the crucial role of NF-κB in AICD, we set out to screen circRNAs related to the NF-κB/AICD axis. Through RNA immunoprecipitation (RIP) against p65, 2 circRNAs (circGSE1, circATXN7) were identified as p65-bound circRNAs (Supplementary Fig. 5D). Using lentiviral vectors expressing shRNAs that target the backsplice junction of the circRNAs and deplete the circular rather than their linear transcripts (Supplementary Fig. 5E), functional assays demonstrated that only circATXN7 had the ability to regulate AICD (Fig. 2C, Supplementary Fig. 5F). circATXN7 interference nearly abrogated lactic acid-induced AICD increase (Fig. 2D), but had little effects on T cell proliferation (Supplementary Fig. 5G) and migration (Supplementary Fig. 5H). As compared to the tumor infiltrated CD8^+ T cells, only a slight circATXN7 expression was detected in peripheral CD8^+ T cells (Supplementary Fig. 5I). Clinically, a significant increased expression level of circATXN7 in tumor-specific CTLs derived from KRAS^MUT versus KRAS^WT CRC tissues (Supplementary Fig. 5J), whereas a comparable circATXN7 expression was found in tumor non-specific CTLs from KRAS^MUT versus KRAS^WT CRC tissues (Supplementary Fig. 5K). In addition, none of the 10 down-regulated circRNAs were identified as eligible candidates for further studies through the screening schematic diagram (Supplementary Fig. 6A−E).

CircATXN7 was formed by the back-splicing of two exons (exon 2 and exon 3) of the *ATXN7* gene (chr3: 63898263-638989011) with 405 nt, and highly conserved between human and mouse (85%). Given the lack of open reading frame (http://www.circbank.cn/), circRNAs have no protein-coding ability. The divergent and convergent primers were adopted to amplify the *ATNX7* circular and linear transcripts using genomic DNA (gDNA) and complementary DNA (cDNA), and found that we could only amplify the *ATXN7* circular transcript from cDNA with divergent primers (Supplementary Fig. 6F). The back-spliced junctions were then confirmed by Sanger sequencing (Supplementary Fig. 6F). The circATXN7 level was significantly lower in oligo dT constructed cDNA than in that of random primers constructed cDNA as a result of 3' polyadenylated tail deficiency (Supplementary Fig. 6G). RNase R treatment and a half-life assay demonstrated that circATXN7 was more stable than *ATXN7* linear mRNA (Supplementary Fig. 6H, I).

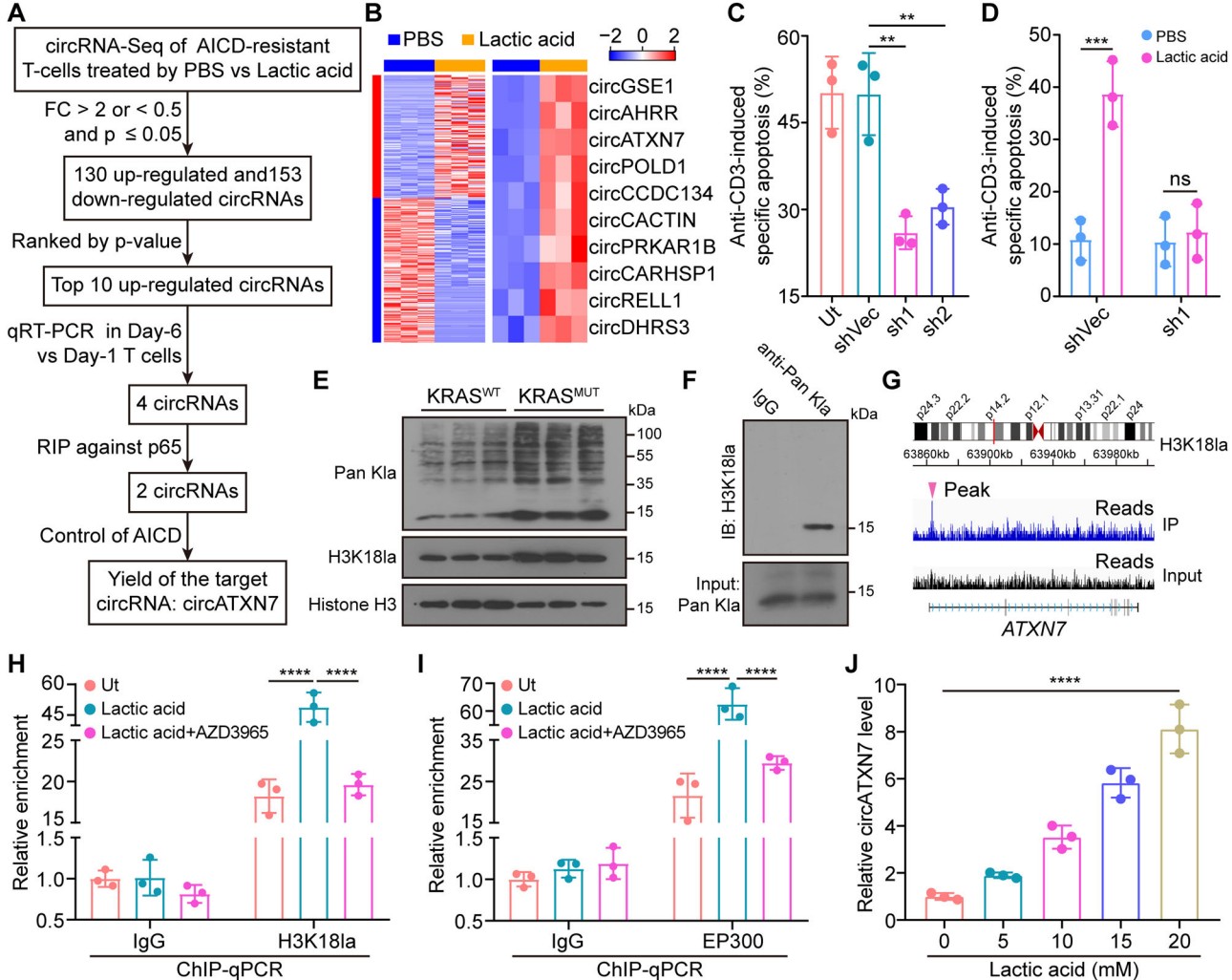

**Fig. 2 | circATXN7 identification and its involvement in AICD. A** The schematic of screening target circRNA which was related to NF-κB/AICD axis. **B** Heatmap of differentially expressed circRNAs in lactic acid- versus PBS-treated Day-1 T cells (*n* = 3 independent samples). **C** Statistics of anti-CD3-induced specific apoptosis for Day-6 T cells transduced with lentivirus carrying an expression cassette for circATXN7 shRNA (sh1 or sh2) or shRNA control vector (shVec) (*n* = 3 independent experiments). Ut, Day-6 T cells without any treatment. **D** Statistics of anti-CD3-induced specific apoptosis for Day-6 T cells transduced with lentivirus carrying an expression cassette for sh1 or shVec, with or without lactic acid (10 mM) treatment (*n* = 3 independent experiments). **E** Western blots showing global histone lactylation and H3K18la levels in tumor-specific CTLs from KRAS^WT or KRAS^MUT CRCs (*n* = 3 samples). **F** Immunoprecipitated lactylated proteins determined by western blots for H3K18la. Three independent experiments were performed and similar

results were obtained. **G** Representative IGV tracks showing enriched H3K18la modification at the *ATXN7* promotor by ChIP-seq analysis of 10 mM lactic acid-treated Day-1 T cells. **H** ChIP-qPCR analysis for H3K18la status at the *ATXN7* promotor of Day-1 T cells treated with 10 mM lactic acid, or lactic acid (10 mM) in combination with 10 nM AZD3965 (*n* = 3 independent experiments). Ut, Day-1 T cells without any treatment. **I** ChIP-qPCR analysis for EP300 status at the *ATXN7* promotor of Day-1 T cells treated with 10 mM lactic acid, or lactic acid (10 mM) in combination with 10 nM AZD3965 (*n* = 3 independent experiments). Ut, Day-1 T cells without any treatment. **J** circATXN7 expression levels in Day-1 T cells treated with indicated concentrations of lactic acid (*n* = 3 independent experiments). Statistical data presented in this figure show mean values ± SD (**C, D, H–J**). **$p \le 0.01$, ***$p \le 0.001$, ****$p \le 0.0001$ and ns indicates $p > 0.05$, by one-way ANOVA (**C, D, H–J**). Source data and exact *p* values are provided as a Source Data file.

Nuclear mass separation assays identified its predominantly cytoplasmic localization (Supplementary Fig. 6J).

We subsequently explored how MCT1-mediated lactic acid uptake induced circATXN7 expression. In view of its contribution of lactic acid-derived histone lactylation to transcription activation[47], we hypothesized that lactic acid-derived histone lactylation might contribute to circATXN7 expression in AICD-sensitive T cells. Interestingly, we found CTLs from KRAS^MUT tumors exhibited high global histone lactylation levels (Fig. 2E), and subsequent immunoprecipitation (IP) identified histone H3K18la as the main target (Fig. 2F). Chromatin immunoprecipitation with sequencing (ChIP-seq) assays demonstrated an obvious enrichment of H3K18la in the *ATNX7* genomic position (Fig. 2G). ChIP-qPCR confirmed H3K18la enrichment in *ATXN7* promoter regions, which could be abolished by blocking lactic acid

uptake using MCT1 inhibitor AZD3965 (Fig. 2H). Furthermore, lactic acid increased histone lactylation writer EP300 binding to the *ATXN7* promoter, and this effect could also be eliminated by AZD3965 (Fig. 2I). In addition, circATNX7 expression was induced in a dose-dependent manner by lactic acid treatment (Fig. 2J). Collectively, our results indicated that lactic acid-derived histone lactylation activates circATXN7 transcription, thereby sensitizing T cells to AICD.

## circATXN7 expression in tumor-specific T cells correlates with adverse clinical outcomes

Next, we sought to evaluate the clinical significance of circATXN7 expression in CRC patients. Using a specific probe to detect the circular rather than known linear transcript of *ATXN7* (Supplementary Fig. 7A, B), RNA in situ hybridization (ISH) assays for circATXN7

expression were performed in paraffin-embedded CRC sections from 269 CRC patients from the SYSU-6thAH cohort. Results demonstrated that circATXN7-positive (circATXN7$^+$) cells were noted scattered in the tumor stroma of 84 out of 87 cases of KRAS$^{MUT}$ CRC (Fig. 3A), but almost no staining was seen in the tumor cells (Fig. 3A). Additionally, they were noted in the stroma of 93 out of 182 KRAS$^{WT}$ cases (Supplementary Fig. 7C), whereas circATXN7$^+$ cells were absent in normal adjacent tissues (Supplementary Fig. 7D, Supplementary Fig. 7E). By sorting each tumor infiltration cell type, RT-PCR (Supplementary Fig. 7F) as well as qRT-PCR (Supplementary Fig. 7G) analysis demonstrated only the whole tumor tissues and CD8 cells had circATXN7 expression, but other components including CD4, macrophages, endothelial cells, and fibroblasts displayed negligible expression of circATXN7. Furthermore, we co-stained frozen sections of CRC tissues using circATXN7 fluorescence in situ hybridization (FISH) and a pentamer carrying the HLA-A2-restricted peptide of human CEA (YLS-GANLNL) that indicates tumor-specific CTLs, and observed circATXN7 and tumor-specific CTLs to be colocalized (Fig. 3B). These results were confirmed by circATXN7 FISH co-stained with CD8, CD4, or EpCAM (Supplementary Fig. 7H). Collectively, we concluded circATXN7 is mainly expressed in tumor-specific CTLs.

We subsequently correlated circATXN7 expression with patient clinicopathological status. In addition to greater circATXN7$^+$ cell counts in KRAS$^{MUT}$ tumors (Fig. 3C) we found that they increased with more advanced TNM stages, and disease relapse recorded within 3 years, and yet this pattern was not seen in KRAS$^{WT}$ cases (Supplementary Fig. 8A, B). Moreover, we identified a negative link between circATXN7$^+$ cells and CTL-abundance in tumors with KRAS$^{MUT}$, but not KRAS$^{WT}$ (Fig. 3D). Tumors with different types of KRAS mutations displayed comparable densities of circATXN7$^+$ cells (Supplementary Fig. 8C). Kaplan−Meier analysis with a follow-up period of 36 months indicated patients with high density of circATXN7$^+$ cells had shortened DFS (Fig. 3E) and OS (Supplementary Fig. 8D). Stratification of the cohort into patients with and without mutant KRAS demonstrated high density of circATXN7$^+$ cells correlated with poor clinical outcomes in KRAS$^{MUT}$ cases, but not in KRAS$^{WT}$ cases (Fig. 3E, Supplementary Fig. 8D). Likewise, KRAS$^{MUT}$ patients with high circATXN7$^+$ cell counts had significantly shortened PFS and OS in the SYSUCC cohort (Fig. 3F, Supplementary Fig. 8E). The clinical significance of circATXN7 was further assessed using circATXN7 ISH staining (Supplementary Fig. 8F) in pancreatic cancer, ~90% of which had KRAS$^{MUT}$[48]. Consistent with the results in CRC, we observed that pancreatic cancer patients with high density of circATXN7$^+$ cells were more likely to have advanced disease (Supplementary Fig. 8G) and poor prognosis (Supplementary Fig. 8H).

On the basis of the link between circATXN7 expression and CTL-abundance, we tried to gain insights into the potential correlation between circATXN7 expression and the clinical efficacy of immune checkpoint inhibitors (ICIs). To address this, we performed circATXN7 ISH staining in paraffin-embedded treatment-naive biopsy samples from 45 CRC patients receiving ICIs, and looked for whether there was an association between circATXN7 expression and their clinical response. In the cohort, 21 (46.7%) patients were non-responders, including 4 (8.9%) with progressive disease (PD) and 17 (37.8%) with stable disease (SD), while 24 (53.3%) patients including 13 (28.9%) with partial response (PR) and 11 (24.4%) with complete response (CR) were identified as responders. Based on the clinical response assessment, we found responder rates were adversely associated with circATXN7 expression (Fig. 3G−J). According to the median level of circATXN7$^+$ cell counts, the cohort was stratified into two categories (patients with circATXN7-low versus -high group). In the circATXN7-high group, 3 (13.6%), 12 (54.5%), 5 (22.7%), and 2 (9.1%) patients had PD, SD, PR, and CR, respectively (Fig. 3K). PD, SD, PR, and CR were recorded in 1 (4.3%), 5 (21.9%), 8 (34.8%), and 9 (39.1%) patients in the circATXN7-low group, respectively (Fig. 3K). These clinical data suggested that high

circATXN7 expression might confer resistance to ICIs for patients with CRC.

## circATXN7 controls AICD by sequestering p65 in the cytoplasm

In light of our findings described above, we aimed to elucidate the contribution of circATXN7 to AICD sensitivity. To this end, circATXN7 expression was firstly monitored in tumor-specific CTLs from KRAS$^{MUT}$ versus KRAS$^{WT}$ tumors, and results showed the former had markedly higher circATXN7 expression levels (Supplementary Fig. 9A). Silencing of circATXN7 in tumor-specific CTLs from KRAS$^{MUT}$ tumors decreased their sensitivity to AICD (Fig. 4A, Supplementary Fig. 9B), whereas circATXN7 loss in tumor-specific CTLs from KRAS$^{WT}$ tumors exerted no effects on their sensitivity to AICD (Supplementary Fig. 9C, D). Conversely, ectopic expression of circATXN7 in tumor-specific CTLs from KRAS$^{WT}$ tumors significantly sensitized them to AICD (Fig. 4A, Supplementary Fig. 9E). In addition, tumor-specific CTL apoptosis induced by coculturing them with autologous tumor cells was abrogated by HLA class I blocking antibodies (Supplementary Fig. 9F). Together, circATXN7 determines the AICD sensitivity of tumor-specific CTLs.

Considering the cytoplasmic localization of circATXN7 (Supplementary Fig. 6J), the copies of circATXN7 and p65 in the cytoplasm of each tumor-specific CTL were further quantified. Results demonstrated that the cytoplasm of each tumor-specific CTL from KRAS$^{MUT}$ tumors contained 1072.4 ± 676.3 and 1978.4 ± 1122.3 copies of circATXN7 and p65 protein, which would allow for an approximately equimolar interaction (Supplementary Fig. 9G, H). However, circATXN7 was 33.8 ± 20.3 copies in the cytoplasm of each tumor-specific CTL from KRAS$^{WT}$ tumors, which was significantly lower than p65 (536.1 ± 171.5 copies per cell; Supplementary Fig. 9G, H). On the basis of the stoichiometry of circATXN7 versus p65 and the fact that each circATXN7 contains one p65-binding motif, we concluded that tumor-specific CTLs of KRAS$^{MUT}$ tumors, but not those of KRAS$^{WT}$ tumors, had sufficient circATXN7 to directly bind p65 for inhibition. Knocking down circATXN7 increased p65 nuclear translocation (Fig. 4B, Supplementary Fig. 9J, K), whereas ectopic expression of circATXN7 prevented p65 nuclear translocation (Supplementary Fig. 9I−K). Subsequent RNA pull-down and western blotting showed p65 was pulled-down by biotinylated probes specific for circATXN7 (Fig. 4C). In vitro RNA/protein interaction analysis indicated the direct interaction of circATXN7 with p65, but not IκBα or p50 (Fig. 4D), which was further confirmed by 3D-structured illumination microscopy (Fig. 4E). We therefore concluded that circATXN7 directly binds to p65.

To identify the structural determinants of the association between circATXN7 and NF-κB p65, we constructed p65 truncates (Fig. 4F). RNA pull-down and RIP assays showed that nuclear localization signal (NLS) motif was essential for the interactions between p65 and circATXN7 (Fig. 4G, Supplementary Fig. 9L). To explore the binding sides of circATXN7 with p65, a computational docking approach was used to show the hydrogen bonds and non-bonded interactions between circATXN7 and p65 (Fig. 4H). To corroborate this prediction, we synthesized blocking oligos that were complimentary to the p65 protein binding sites in circATXN7. RNA pull-down indicated the blocking oligos decreased the interactions between circATXN7 and p65 (Fig. 4I), which was confirmed by RIP assays (Supplementary Fig. 9M). By construction of a flag-tagged p65 with circATXN7-binding site mutation, we found mutated p65 could not be pulled-down by circATXN7 (Fig. 4J). Furthermore, transfection with the blocking oligos significantly abolished the ability of circATXN7 to block p65 subunit translocation (Fig. 4K). Ectopic overexpression of circATXN7 with p65-binding site mutation lost the ability to sequester p65 in the cytoplasm (Fig. 4L). These findings indicated circATXN7 sequesters p65 in the cytoplasm by directly masking its NLS motif, thereby inactivating NF-κB.

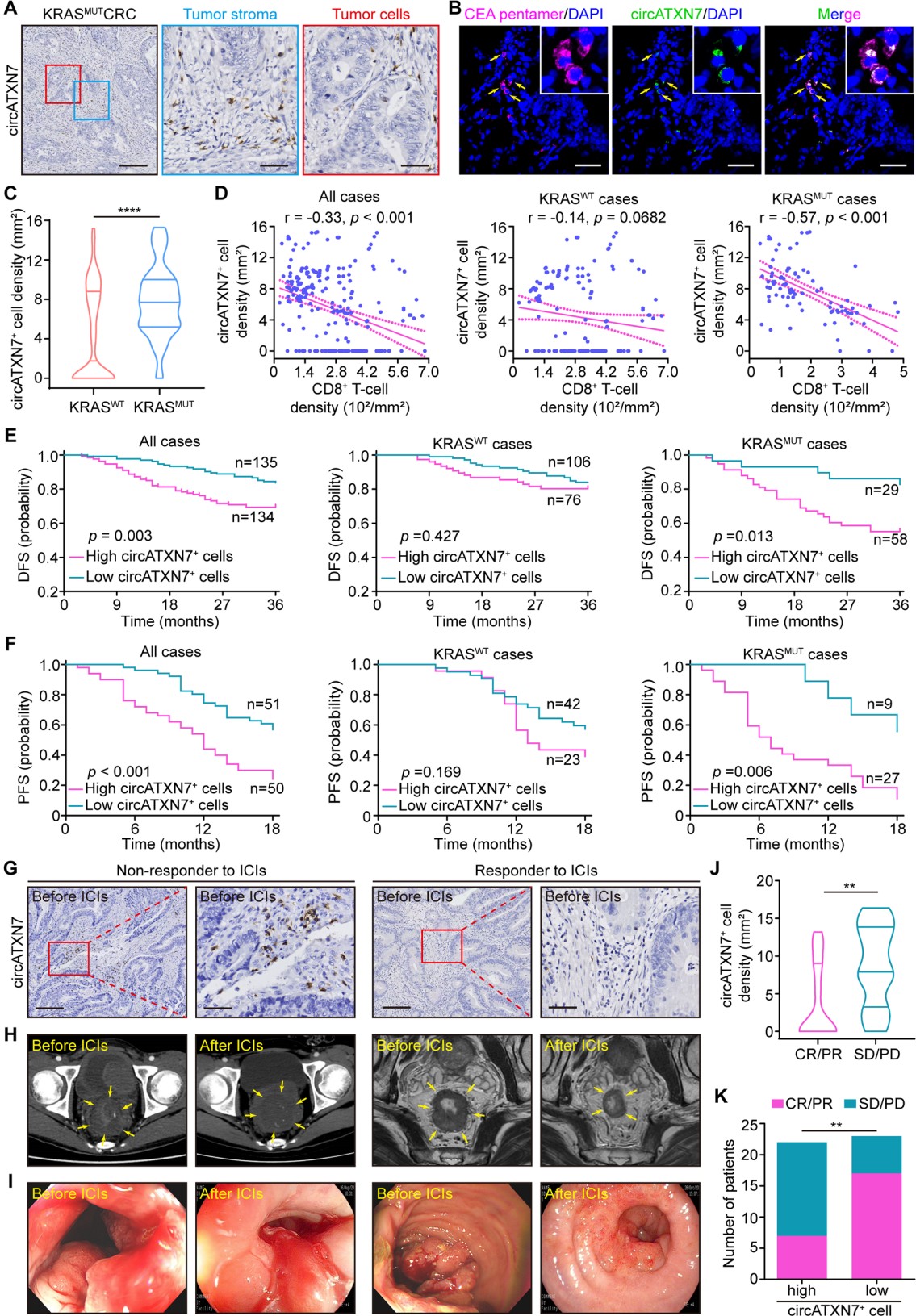

Subsequently, we evaluated whether circATXN7 exerts its effect on AICD by inhibiting NF-κB. In KRAS$^{MUT}$ tumors-derived tumor-specific CTLs, blocking p65 nuclear translocation using NF-κB inhibitors BAY or JSH-23 abolished the AICD reduction upon circATXN7 silencing (Supplementary Fig. 9N). Moreover, BAY or JSH-23 blocked the AICD increase resulting from circATXN7 overexpression in KRAS$^{WT}$ tumors-derived tumor-specific CTLs (Supplementary Fig. 9N). In addition, there were no AICD-promoting effects when tumor-specific CTLs from KRAS$^{WT}$ tumors and the Day-1 T cells overexpressed a mutant cir-cATXN7 version that could not bind to p65 (Supplementary Fig. 9O, P). A similar pattern was observed when p65 blocking oligos were trans-fected (Supplementary Fig. 9Q, R). These results suggested that

**Fig. 3 | Clinical significance of circATXN7 expression. A** Representative images of circATXN7 ISH staining in KRAS^MUT CRC tissues. Scale bars: 200 μm (left), 50 μm (middle and right). **B** circATXN7 fluorescence in situ hybridization (green) and CEA pentamer fluorescence (pink) imaged using confocal microscopy demonstrating circATXN7 and tumor-specific CTL colocalization. Scale bars: 50 μm. **C** Statistics of circATXN7+ cell density in KRAS^WT (n = 182 patients) and KRAS^MUT (n = 87 patients) CRC tumor tissues from SYSU-6thAH. **D** Pearson's correlation analysis between circATXN7+ cell and CD8+ T cell density in KRAS^WT (n = 182 patients) and KRAS^MUT (n = 87 patients) CRC tumor tissues from SYSU-6thAH. **E** Kaplan−Meier curves for DFS layered by circATXN7+ cell density in KRAS^WT (n = 182 patients) and KRAS^MUT (n = 87 patients) CRC cases from SYSU-6thAH. **F** Kaplan−Meier survival curves for PFS layered by circATXN7+ cell density in KRAS^WT (n = 65 patients) and KRAS^MUT (n = 36 patients) CRC patients from SYSUCC. **G** Representative images of circATXN7 ISH staining in tumor biopsy specimens from CRC patients who then received ICIs (n = 45 patients). Scale bars: 200 μm (left), 50 μm (right). **H** Representative computed tomography (left), magnetic resonance imaging (right), and **I** endoscopic images from high- or low- circATXN7+ cell density CRC patients before and after ICIs (n = 45 patients). **J** Statistics of circATXN7+ cells in CRC patients with CR/PR (n = 24 patients) or SD/PD (n = 21 patients). **K** Number of CRC patients with CR/PR or SD/PD in circATXN7-low and -high groups. **≤0.01, and ****p ≤ 0.0001, by two-sided Mann−Whitney test (**C** and **J**), Person's correlation analysis (**D**), two-sided log-rank test (**E**, **F**), or two-sided Chi-Square test (**K**). Source data and exact p values are provided as a Source Data file.

interacting with p65 is indispensable for circATXN7 to sensitize T cells to AICD.

## Targeting circATXN7 in CD8+ T cells selectively inhibits KRAS^MUT tumors

To determine the in vivo function of circATXN7 in T cells, genetically engineered *circAtxn7^loxp/loxp* mice were crossed with *CD8a^cre* mice (Fig. 5A). We genotyped the progenies to obtain *CD8a; circAtxn7^loxp/loxp* mice (Supplementary Fig. 10A) (termed *circAtxn7^CKO* mice). As expected, these mice lacked the circular form *circAtxn7* while the levels of the linear host gene *Atxn7* mRNA and ATXN7 protein were unaltered (Supplementary Fig. 10B, C), and these mice exhibited no marked abnormalities in gestation, birth, development or growth. Moreover, the numbers of thymocytes and peripheral CD4+ and CD8+ T cells were unchanged upon ablation of *circAtxn7^CKO* mice (Supplementary Fig. 10D), as were the frequencies of their corresponding subsets (Supplementary Fig. 10E−H). Therefore, circAtxn7 appears to be dispensable for mouse T cell development. Next, MC38 cells expressing wild-type *Kras* were stably transfected with cDNA encoding Kras^G12D to generate MC38-Kras^G12D cells (designated MC38K). Then MC38 and MC38K cells were subcutaneously grafted into *circAtxn7^CKO* mice or wild-type (WT) control littermates. MC38K tumor growth in *circAtxn7^CKO* mice expanded more slowly and was markedly smaller at the endpoint (Fig. 5B). A similar pattern was obtained using MC38 cells expressing *Kras^G12V* (Supplementary Fig. 11A), or *Kras^G13D*, (Supplementary Fig. 11B). By contrast, loss of circAtxn7 in CD8+ T cells had little effects on MC38 tumors (Fig. 5C). Similar results were obtained in investigations determining effects of targeting circAtxn7 with a different murine pancreatic model (Supplementary Fig. 11C, D). These findings indicated that genetic ablation of circAtxn7 selectively suppresses KRAS^MUT tumors.

The subcutaneous xenograft results inspired us to further explore the in vivo role of circATXN7 by tumor cell orthotopic injection into the cecum wall of *circAtxn7^CKO* and WT mice. Although *circAtxn7^CKO* and WT mice exhibited similar MC38K orthotopic tumor formation incidence, the former had significantly decreased tumor burden as indicated by ~3-fold reduction in the orthotopic tumor sizes (Fig. 5D, E). Gross inspection at the endpoint of 24 days identified no liver metastasis in either group of mice, but hematoxylin-eosin (H&E) staining found a significant decrease in the tumor burden of liver micrometastasis in *circAtxn7^CKO* mice (Fig. 5D). Furthermore, we performed qRT-PCR using primers that specifically amplify the CMV promoter of the stably integrated vector. Results demonstrated that circAtxn7 deletion in CD8+ T cells decreased the disseminated MC38K burdens in the livers (Fig. 5E). Analogous to MC38 subcutaneous tumor model results, we observed insignificant anti-tumor activity of circAtxn7 ablation in MC38 orthotopic tumor models (Fig. 5F, G). Of note, CD8+ T cells contributed to the anti-tumor effects of targeting circAtxn7, as evidenced by equal susceptibility to MC38K tumor inoculation in *circAtxn7^CKO* and WT mice after CD8+ T cell depletion (Supplementary Fig. 11E, F). In contrast, CD4+ T cells were of little importance to the anti-tumor activity of circAtxn7 ablation (Supplementary Fig. 5E, G).

These data confirmed the mutant-selective tumor inhibition of circATXN7 deletion in CD8+ T cells.

Subsequently, we conducted RNA sequencing of bulk RNA extracted from tumor-infiltrating CD8+ T cells sorted from WT and *circAtxn7^CKO* MC38K tumor-bearing mice. Based on the transcriptional changes, gene set enrichment analysis (GSEA) identified a significant enrichment related to NF-kB signaling in circAtxn7-deficient tumor-infiltrating CD8+ T cells (Fig. 5H). Subsequent assays confirmed a NF-κB activation increase in circAtxn7-deficient CD8+ T cells (Supplementary Fig. 11H, I). Moreover, we observed an apoptosis-associated gene signature in circAtxn7-deficient CD8+ T cells (Fig. 5H). Further findings that circAtxn7-deficient CD8+ T cells had higher antiapoptotic gene expression (*Bcl2*, *Bcl2l1*, *Ier3* and *Gadd45b*) characterized these cells as having dampened apoptosis (Supplementary Fig. 11J). Therefore, circAtxn7-deficient CD8+ T cells are reprogrammed to enhance NF-κB activation and decrease apoptosis, further testifying the involvement of circATXN7 in NF-κB/AICD axis. More importantly, we found a substantial increase in tumor-infiltrating CD8+ T cell density in MC38K tumors from *circAtxn7^CKO* mice than those from WT littermates (Fig. 5I, J), as well as in cytotoxic cytokine IFN-γ production (Fig. 5K) and the expression of perforin and CD107a, markers related to cytotoxic activity (Supplementary Fig. 11K), but no significant effects on the exhausted phenotype (Supplementary Fig. 11L). These results demonstrated that circAtxn7 deletion in T cells could shift KRAS^MUT tumors from immunologically "cold" to "hot" by increasing T cell resistance to apoptotic programs, which correlates with a reduction in tumor progression parameters.

## Loss of circATXN7 improves immunotherapy efficacy

Given that targeting circAtxn7 shifted KRAS^MUT tumors from immunologically "cold" to "hot", we therefore questioned whether circATXN7 interference could potentiate ICI responses. To address this, we first performed anti-PD1 treatment in *circAtxn7^CKO* and WT mice with MC38K subcutaneous tumors, an MSI-H model but resistant to immunotherapies[9,49]. As anticipated, anti-PD1 treatment in WT mice had no significant anti-tumor activity against MC38K tumors (Fig. 6A). By contrast, anti-PD1 therapy in *circAtxn7^CKO* mice exhibited significant tumor inhibitory effects, as shown by the smaller tumor size and improved survival (Fig. 6B). These results suggested blocking circAtxn7 expression in CD8+ T cells could be an effective strategy for improving anti-PD1 efficacy. Further support for this possibility comes from experiments assessing anti-PD1 efficacy with two different KRAS^MUT tumor models, including murine pancreatic cancer (Supplementary Fig. 12A, B), and melanoma (Supplementary Fig. 12C, D). We thus concluded circATXN7 represents a key determinant in maintaining anti-PD1 therapy resistance in KRAS^MUT tumors, and targeting circATXN7 might help overcome resistance to PD1 blockade therapy.

We further assessed whether circATXN7 interference in adoptively transferred T cells might prevent AICD and increase adoptive T cell therapy (ACT) efficacy in combating KRAS^MUT tumors. To this end, we used transgenic OT-I expressing mice to purify OT-I cells. After activated by cognate SIINFEKL peptide, OT-I cells were transduced

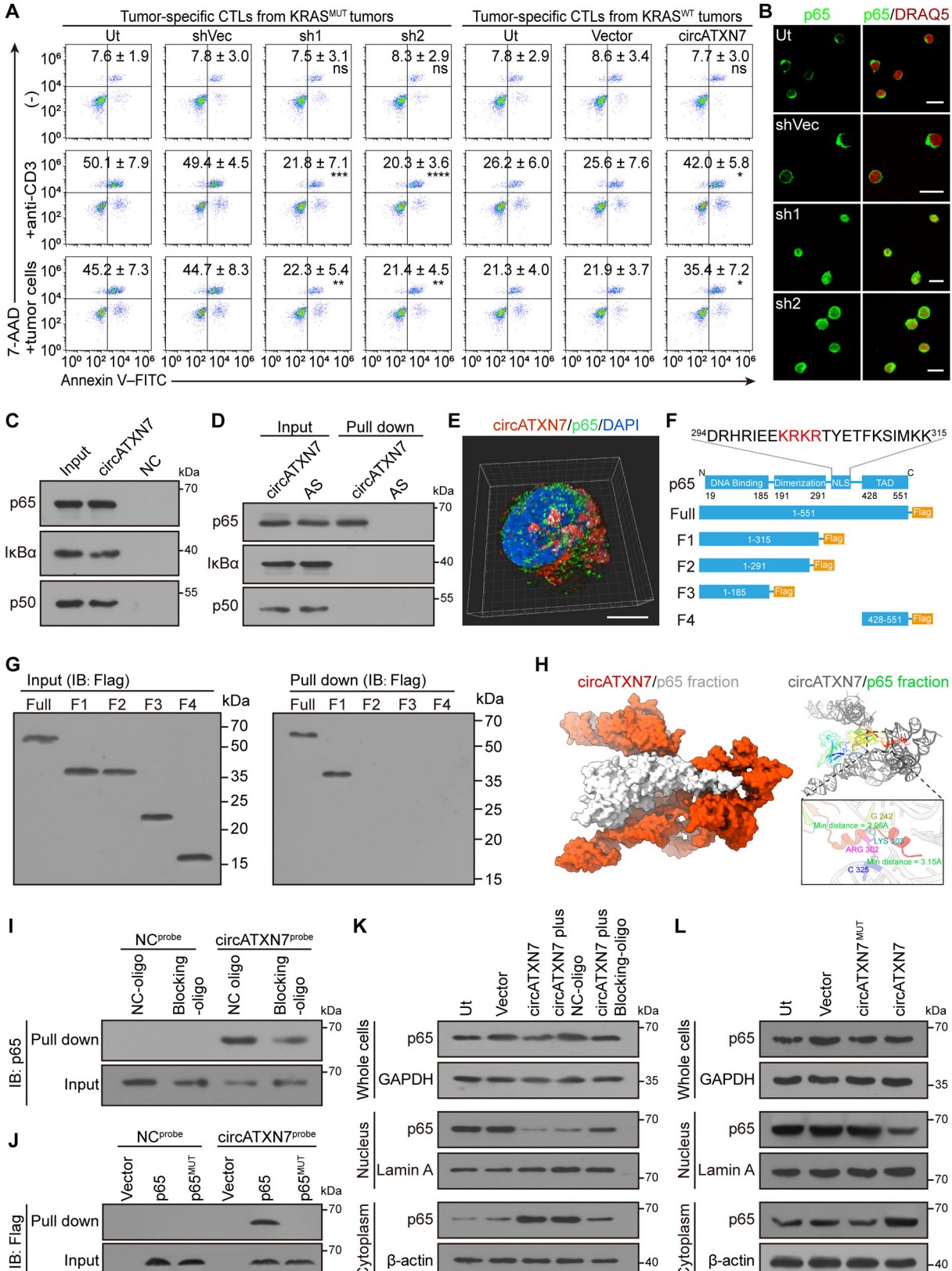

with shRNAs specifically targeting circAtxn7 to dampen its expression (Supplementary Fig. 12E). In vitro experiments demonstrated that although circATXN7 silencing in OT-I cells did not have significant effects on their proliferation (Supplementary Fig. 12F), migration (Supplementary Fig. 12G), exhausted phenotype (Supplementary Fig. 12H) or activation (Supplementary Fig. 12I), it increased the expression of perforin and CD107a, markers related to cytotoxic activity (Supplementary Fig. 12J) and NF-κB activation (Supplementary Fig. 12K), but decreased AICD sensitivity (Supplementary Fig. 12L). OT-I cells transduced with GFP-tagged circAtxn7 shRNA or mCherry-tagged empty vector were intravenously transferred into MC38K-OVA tumor-bearing mice (Fig. 6C). In light of monitoring by flow cytometry, the

**Fig. 4 | circATXN7 inactivates NF-κB by sequestering p65 in the cytoplasm.**
**A** Apoptosis of the indicated tumor-specific CTLs induced by autologous tumor cells or anti-CD3 (n = 4 samples). Ut, cells without any treatment. Numerical values (mean ± SD) denote annexin V$^+$ cell percentages (*p ≤ 0.05, **p ≤ 0.01, ***p ≤ 0.001, ****p ≤ 0.0001 and ns indicates p > 0.05 compared with shVec or Vector by one-way ANOVA). **B** Immunofluorescent staining for p65 (green) nuclear translocation in KRAS$^{MUT}$ tumor-derived tumor-specific CTLs with the indicated treatments (n = 3 patients). Nuclei were stained with DRAQ5 (red). Ut, cells without any treatment. Scale bars: 10 μm. **C** circRNA pull-down assays using biotin-labeled circATXN7 probes indicating circATXN7 binding to p65, IκBα, and p50 in Day-6 T cells. **D** In vitro circRNA-protein binding assay demonstrating circATXN7 directly binding to p65. **E** Representative 3D images for circATXN7 FISH (red) and p65 IF (green) staining showing the colocalization of circATXN7 with p65 in tumor-specific CTLs. Nuclei were stained with DAPI (blue). Scale bar: 10 μm. **F** Schematic illustration of p65 functional domains and corresponding truncation constructs. **G** circRNA pull-down assays were conducted using biotin-labeled circATXN7 probes against cell lysates from Day-6 T cells transfected with full-length p65 or the indicated deletion

mutants. Co-precipitated proteins were detected by immunoblots using anti-Flag antibodies. **H** circATXN7 and p65 docking model. **I** circRNA pull-down assays were conducted using biotin-labeled NC or circATXN7 probes against cell lysates from Day-6 T cells transfected with blocking oligo (5′ CTCCCCGACCGTCGC-CATTGCGGCGGCCGAG 3′ complimentary to 5′ CUCGGCCGCCGCAAUGGCGACG 3′ of circATXN7) or NC oligo. Co-precipitated proteins were detected by immunoblots using anti-p65 antibodies. **J** circRNA pull-down assays were done using biotin-labeled circATXN7 probes against cell lysates from Day-6 T cells transfected with vector, p65, or binding site-mutated p65 (p65$^{MUT}$). Co-precipitated proteins were detected by western blot using anti-Flag antibodies. **K** Immunoblots showing p65 nuclear translocation in circATXN7 overexpressed Day-1 T cells transfected with blocking oligo or NC oligo. **L** Immunoblots showing p65 nuclear translocation in Day-1 T cells transfected with circATXN7 or p65-binding site-mutated circATXN7. In panels **C**, **D**, **G**, **I**–**L**, three independent experiments were performed and similar results were obtained. Source data and exact p values are provided as a Source Data file.

transferred cell distribution at 1, 3, 5, and 7 days after transfer showed comparable GFP- versus mCherry-tagged cell recruitment (Fig. 6D, E). However, at later time points of 14 and 21 days after transfer, mCherry-tagged cells declined dramatically, whereas circAtxn7 silencing significantly prolonged GFP-tagged cell persistence (Fig. 6D, E). At endpoint, loss of circAtxn7 did not alter the transferred cells' exhausted phenotype and activation (Supplementary Fig. 12M), but endowed the tumors with substantially increased CTL densities (Fig. 6F) and increased the expression of perforin and CD107a, markers related to cytotoxic activity (Supplementary Fig. 12M), which correlated with improved circAtxn7-silenced T cell anti-tumor activities (Fig. 6G, H).

According to our findings described above, we sought to recapitulate the results in more humanized models. Therefore, we established CRC patient-derived xenograft (PDX) models implanted in NOD.SCID mice. Tumor-reactive T cells were obtained by co-incubation of PB CD8$^+$ T cells with autologous dendritic cells (DCs) primed by tumor lysates from the same donors, followed by GFP-tagged circATXN7 shRNA or with mCherry-tagged empty vector transduction (Fig. 6I, Supplementary Fig. 13A). When the tumors were palpable, GFP- or mCherry-tagged tumor-reactive T cells were injected intravenously into PDX-bearing mice (Fig. 6I). In PDX tumors generated from CRC patients with KRAS$^{G12D}$ (Supplementary Fig. 13B), flow cytometric analysis revealed that loss of circATXN7 (Supplementary Fig. 13C) endowed T cells with greatly improved persistence in tumors (Supplementary Fig. 13D). At endpoint, we observed a significant improvement in circATXN7-silenced T cell accumulation (Fig. 6J, K), as well as an increase in NF-κB activity (Fig. 6L, Supplementary Fig. 13E) and antiapoptotic gene expression (Supplementary Fig. 13F–I). These results suggested that circATXN7 deletion in transferred T cells could overcome tumor immune evasion by preventing AICD, thereby allowing them to exert stronger anti-tumor activity (Fig. 6M, N). The above observations were faithfully recapitulated by another PDX model generated from CRC patients with KRAS$^{G13D}$ (Supplementary Fig. 13J–M). Together, our results pointed to an encouraging anti-tumor avenue to improve immunotherapy efficacy.

## Discussion

While immunotherapy exhibits anti-tumor activity in some patients with MSI-H CRC, approximately 85% among all CRC patients, the therapeutic benefit is largely restricted, highlighting an unmet need for the study of mechanisms and combination regimens with immunotherapies. Here, our findings demonstrated a clear CTL reduction in KRAS$^{MUT}$ tumors that correlated with shortened survival and poor ICI efficacy. This clinical phenomenon was linked to KRAS$^{MUT}$-driven lactic acid that acted as a precursor to stimulating histone lactylation. As an epigenetic modification, histone lactylation directly elicited cir-cATXN7 expression, which sensitized tumor-specific CTLs to tumor-

mediated AICD by sequestering the NF-κB p65 subunit in the cytoplasm (Fig. 7). This represents a hitherto undescribed causative factor for the inverse association between CTLs and KRAS$^{MUT}$.

The AICD sensitivity can be reprogrammed by several diseases including neoplasia and autoimmune diseases. Current research indicates different T cell subsets display distinct AICD sensitivity. In breast and lung cancer microenvironments, Huang and colleagues demonstrated that tumor-infiltrating CTLs and type 1 T helper (Th1) cells were more sensitive to AICD than Tregs and type 2 T helper cells[28]. Our previous work showed that Th1 and type 17 helper T cells, but not regulatory T cells, were able to evade AICD in patients with Crohn's disease[25]. This study found tumor-specific CTLs in KRAS$^{MUT}$ tumors were susceptible to tumor-mediated AICD. The increase in ACID has also been noted in various conditions. A study by Tan et al. suggested Th1 cells were susceptible to AICD in the context of mouse eye inflammation[50]. Also, virus infections could induce apoptosis in T cells by AICD[51–53]. The increase in AICD of activated CD8$^+$ T cells generated during a viral infection serves to maintain homeostasis of the immune system, so that during the resolution phase of infection, excess activated T cells are deleted[54–56]. These findings indicated that the increase in ACID seemed not to be a tumor-specific mechanism, but appeared to be disease context-dependent.

A salient feature of most solid tumors with KRAS$^{MUT}$ is elevated lactic acid production. This characteristic endows these tumors with lactic acid accumulation and increased tumor acidity. As a common metabolite, lactic acid has been shown to promote tumor immune evasion related to regulatory T cells[16], tumor-associated macrophages[14], and myeloid-derived suppressor cells[17]. Further evidence linking lactic acid with tumor immunoescape comes from our present data in which we showed that elevated lactic acid in KRAS$^{MUT}$ tumors could sensitize tumor-specific CTLs to AICD. The results from our study, as well as previous reports[16,18,57], indicated immune-suppressive roles of lactic acid in the process of tumor immuno-surveillance. Furthermore, we illustrated MCT1-mediated lactic acid uptake, but not lactic acid/GPR81 signaling, plays a major role in regulating AICD. In contrast, a study by Renner et al. reported that MCT1 contributes to the export of lactic acid[58]. It is conceivable that MCT1-mediated lactic acid transport might have distinct context-dependent effects because it is a passive process that depends on the lactic acid gradient over cell membrane.

Since its first description as an epigenetic modification in 2019[47], the contribution of lactic acid-derived histone lactylation to gene transcription has come to light. Subsequent studies have implicated various cell types in histone lactylation, such as monocytes[59], macrophages[60], and myeloid cells[61]. We recently demonstrated histone lactylation boosted oncogene transcription activation in malignant cells[62]. These pioneering reports have moved this relatively young field

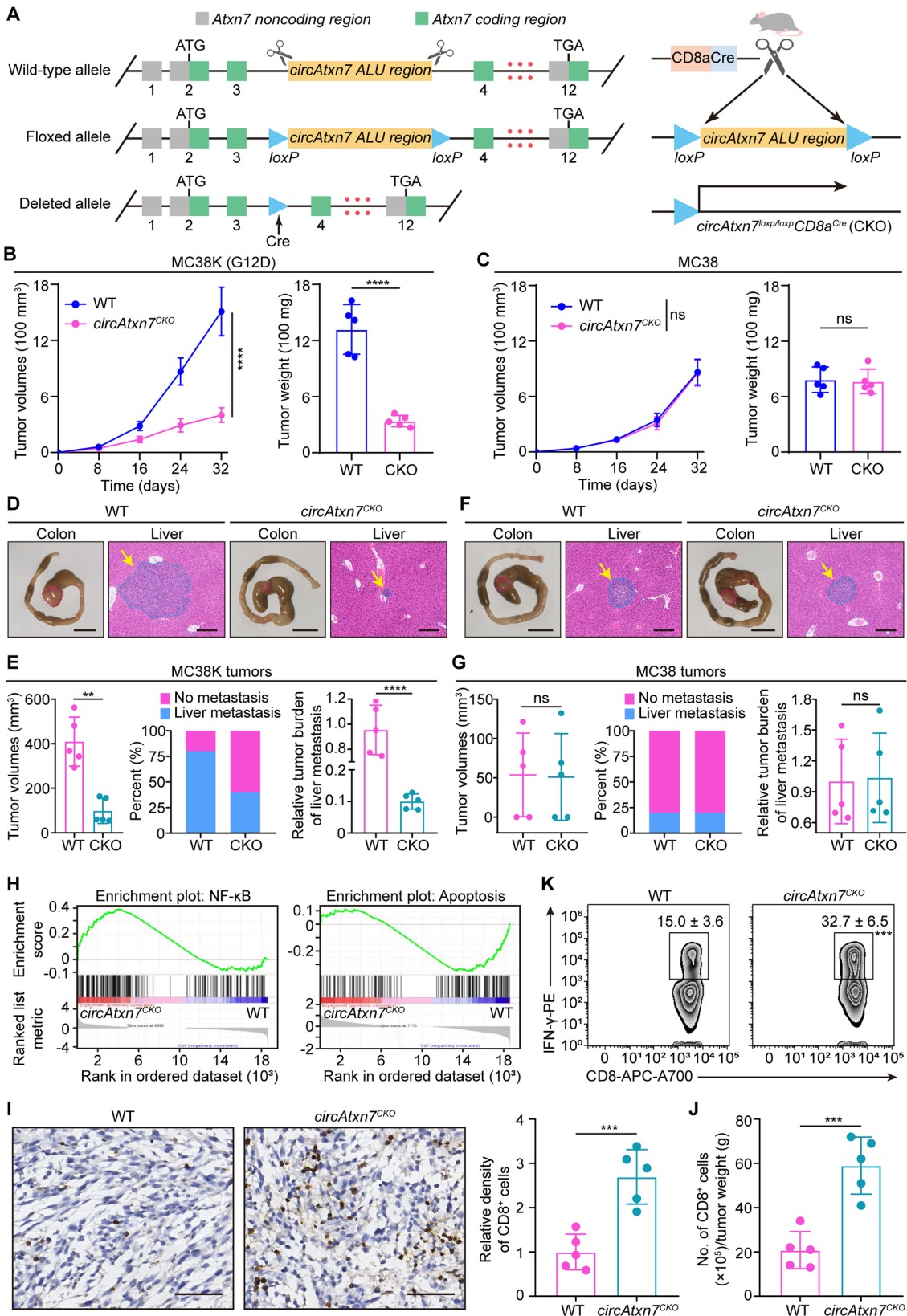

of research forward. However, whether histone lactylation contributes to T cell-mediated cancer immunology remains an important knowledge gap. Here, we reported the involvement of histone lactylation in regulating T cell-mediated tumor immunological escape by showing that histone lactylation-activated circATXN7 sensitized tumor-specific CTLs to AICD. These findings enable a better understanding of the link

between epigenetic modification and tumor immunology. However, it is worthy of further efforts to understand the roles of lactate-lactylation in other cell types in tumor microenvironment.

The emerging roles of circRNAs in cancer and oncology bring them to the forefront of clinical practice. Current research indicates that circRNAs can control various aspects of cancer immunology[63]. A

**Fig. 5 | Mutant-selective tumor inhibition in vivo by targeting circATXN7.**
**A** Schematic diagram demonstrating targeted genome editing at the *Atxn7* gene, followed by intercross with *CD8α^cre^* mice to generate *CD8α^cre^; circAtxn7^loxp/loxp^* (*circAtxn7^CKO^*) mice. MC38K (G12D; **B**) or MC38 cells (**C**) were subcutaneously injected into WT or *circAtxn7^CKO^* mice. During the course of each group (*n* = 5 animals), tumor volumes were monitored, and tumor weights were measured at day 32. **D–G** 5 × 10^5 MC38K(G12D) or MC38 cells were injected into the cecum submucosa of WT or *circAtxn7^CKO^* mice to generate orthotopic xenograft CRC models (*n* = 5 animals). Mice were sacrificed 24 days after injection. Gross inspection of MC38K (**D**) or MC38 (**F**) orthotopic tumors and representative H&E staining of liver micro-metastasis. Scale bars: 1 cm (left panel); 200 μm (right panel). **E–G** At day 24, MC38K or MC38 orthotopic xenografts were subjected to analyses of tumor volumes, liver metastasis rate, and liver CMV expression. **H** At day 24, CD8^+ T cells were purified from MC38K orthotopic xenografts in WT or *circAtxn7^CKO^* mice, and

then subjected to RNA-seq (*n* = 3 samples). GSEA analysis showing an enrichment related to NF-kB signaling and apoptosis in circAtxn7-deficient tumor-infiltrating CD8^+ T cells. **I** IHC staining for CD8 and relative CD8^+ cell density in MC38K orthotopic xenografts in WT or *circAtxn7^CKO^* mice at day 24 (*n* = 5 samples). Scale bars, 50 μm. **J** MC38K orthotopic tumors in WT or *circAtxn7^CKO^* mice at day 24 were subjected to flow cytometry analysis for CD8^+ cell density (*n* = 5 samples). **K** CD8^+ T cells were purified from WT and *circAtxn7^CKO^* MC38K orthotopic tumor-bearing mice at day 24, and then subjected to flow cytometry analysis for IFN-γ expression (*n* = 5 samples). Numerical values denote IFN-γ^+ CD8^+ T cell percentages relative to total CD8^+ cells. Statistical data presented in this figure show mean ± SD (**B, C, E–G, I–K**). ***p ≤ 0.001, ****p ≤ 0.0001, and ns indicates p > 0.05, by two-sided Student's t-test (**B, C,** right panel of **E, G, K, I,** and **J**), or two-sided Mann–Whitney U test (left panel of **E**). Source data and exact p values are provided as a Source Data file.

study by Jia and colleagues showed circFAT1 reduced CD8^+ T cell infiltration by binding to STAT3 in the cytoplasm in squamous cell carcinoma[64]. In non-small cell lung cancer cells, tumor cell-expressed circIGF2BP3 caused immune escape from CD8^+ T cell-mediated tumor killing[65]. These insights shed light on an association between circRNAs and CTL dysfunction. Nevertheless, there is a lack of research focusing on how circRNAs in tumor-infiltrating T cells involve in tumor immunology. Against this background, we determined circRNA expression profile in AICD-sensitive versus -resistant T cells. Subsequent assays provided evidence demonstrating the contribution of tumor-specific CTLs-expressed circATXN7 to tumor immune escape and anti-PD1 therapy resistance. The present work deciphers the role and targeted therapeutic potential of circRNAs in tumor-infiltrating T cells per se, which might advance cancer immunotherapies.

The interaction between proteins and circRNAs can be often seen in the current literatures[66,67]. For instance, Guarnerio and colleagues found that circCsnk1g3 and circAnkib1 can interact with RIG-I at a close molar ratio in the sarcoma cells[38]. The present study proposed a model in which circATXN7 directly binds with p65 in the tumor-specific CD8^+ T cells of KRAS^MUT CRC. Furthermore, the stoichiometry of the circATXN7 versus that of p65 indicated that the interaction between circATXN7 and p65 was approximately equimolar. Although Mann et al. estimated that each HeLa cell contained >180,000 copies of p65[68], this work demonstrated p65 protein was expressed in the tumor-specific CD8^+ T cells of KRAS^MUT CRC at ~2000 copies per cell. These findings suggested a cell type specific protein expression pattern. One protein might have different abundance in different cells. Additional support for this possibility comes from previous studies[69,70] in which they are demonstrated that each A549 cell, and VSV-infected macrophage contained 50, ~1000 copies of RIG-I, respectively. On the other hand, the protein copy number range can span several orders of magnitude in one specific type of cell[71]. It has been reported that the protein copies per HeLa cell vary from 3 to >80,000,000[68]. These findings confirmed the protein abundance was cell context-dependent, but not a general feature in different types of cells.

Therapeutic attempts to tackle KRAS^MUT have been continuing for decades. Due to the benefits of ACT in a subset of cancer patients, much interest is dedicated to the study of T cell receptors targeting KRAS^MUT[72,73]. Along this line, Rosenberg and colleagues[74] demonstrated the tumor regression of metastatic CRC after the administration of cytotoxic T cells targeting mutant KRAS^G12D. A similar pattern in pancreatic cancer was showed in a recent study by Tran et al.[75]. These insights suggest that KRAS-driven tumors can be targeted efficiently by reprogramming immune program. A study by DePinho et al. reinforced this therapeutic strategy by showing that inhibition of myeloid-derived suppressor cell recruitment could overcome resistance of tumors expressing KRAS^G12D to anti-PD1 therapy[49]. These therapeutics, however, require the expression of KRAS^G12D and cannot be used against non-G12C mutants. As such, efforts to seek approach that enables broad inhibition of KRAS^MUT or its related downstream

signaling are continuing. Our work here identified a KRAS^MUT-activated circATXN7 program as an exploitable therapeutic approach to combat KRAS^MUT tumors, which did not correlate with the KRAS mutation type and appeared to be a general feature of KRAS^MUT tumors. In vitro and in vivo experiments showed targeting circATXN7 in T cells protected T cells from tumor-mediated AICD. Accordingly, circATXN7 ablation shifts KRAS^MUT tumors from immunologically "cold" to "hot" and consequently improves immunotherapeutic efficacy. Although emerging data suggest the potential of cancer cell-expressed oncogenic circRNAs as therapeutic targets, this study provides insight into targeting immunocyte-located circRNAs for cancer immunotherapies. The clinical relevance of this therapeutic strategy is further supported by ACT success in PDX models, as well as our findings that high circATXN7 expression in CRC patients correlates with poor response to ICIs.

In summary, this work identified circATXN7 as a major driver of tumor immune evasion that sensitized tumor-specific CTLs to AICD. These findings define a therapeutic strategy by demonstrating that circATXN7-deficient CD8^+ T cells are reprogrammed to long-lived cells by preventing their AICD, thereby improving the therapeutic efficacy of ICIs and ACT.

## Methods
### Patients and tissue samples
The experiments related to human samples were performed with the approval of the Institutional Review Board of The Sixth Affiliated Hospital of Sun Yat-sen University (Guangzhou, China; approval number: G2020008 and G2022024) and Sun Yat-sen University Cancer Center (Guangzhou, China; approval number: G2021-088-01 and B2022-025-01), covering the collection of formalin-fixed, paraffin-embedded tissues, PBMCs and primary CRC specimens. Informed consent was waived for the collection of formalin-fixed, paraffin-embedded tissues, whereas informed consent was obtained from each patient for the collection of PBMCs and primary CRC specimens. Paraffin-embedded tumor samples were obtained from 269 patients with stage I–III CRC and 101 patients with stage IV CRC. Paraffin-embedded biopsy specimens were harvested from 45 patients who underwent ICIs at The Sixth Affiliated Hospital, Sun Yat-sen University. Patient who received radical surgery was managed per local guidelines, and the cohort receiving ICIs was managed according to the most appropriate approach decided by the multidisciplinary team and specific needs of each patient. KRAS status was assessed by Sanger sequencing, and pathologists determined patients' mismatch repair status according to the immunohistochemistry (IHC) staining for four proteins (MLH1, MSH2, MSH6 or PMS2). Shanghai Outdo Biotech (Shanghai, China) provided pancreatic cancer tissues from 87 patients under the approval of the internal ethics review board (approval number: SHYJS-CP-1901008), in which anonymized data were analyzed, and waived the requirement for informed consent.

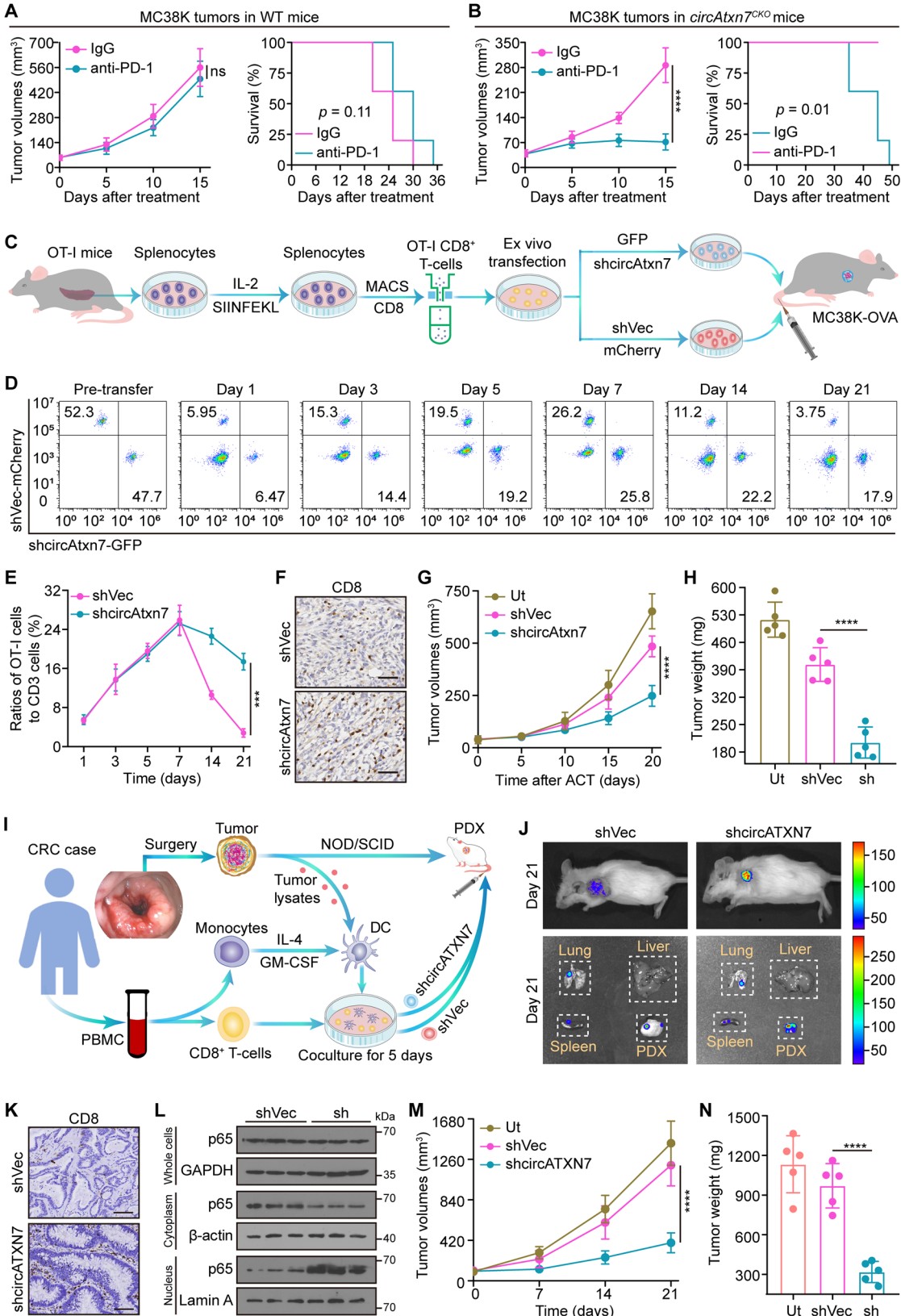

## Mouse strain generation and experiments

*CircAtxn7^loxp/loxp* mice were generated using CRISPR-Cas9-mediated genome editing by Cyagen Biosciences Inc. (China). Immunocompromised NOD.SCID mice were obtained from GemPharmatech (China) and used to establish patient-derived xenograft models. OT-I (C57BL/6-Tg (TcraTcrb)1100Mjb/J), *Cd8a*-Cre (C57BL/6-Tg (Cd8a-cre)1Itan/J) were provided by the Jackson Laboratory. To generate mice with circAtxn7 conditional deletion in CD8+ T cells, *CircAtxn7^loxp/loxp* mice were crossed with *Cd8a*-Cre mice. Conditional knockout of circAtxn7 was confirmed by qRT-PCR using T cells puried from the spleen. We genotyped these experiment cohort strains by PCR amplification methods. PCR primers used in genotyping are listed in Supplementary

**Fig. 6 | Effects of circATXN7 on immunotherapy efficacy. A, B** Tumor growth and survival curves of MC38K subcutaneous xenografts in WT (**A**) or *circAtxn7^CKO* (**B**) mice treated with anti-PD1 antibodies or IgG isotype control antibodies (*n* = 5 animals). **C** Schematic diagram showing circAtxn7 silencing in OT-I CD8⁺ T cells and ACT therapy against MC38K-OVA tumors. **D** 7.5 × 10⁵ OT-I cells transduced with GFP-tagged shcircAtxn7 were mixed with 7.5 × 10⁵ OT-I cells transduced with mCherry-tagged shVec, and co-injected intravenously into MC38K-OVA tumor-bearing mice, followed by flow cytometric analyses of the proportion of OT-I cells in total CD8⁺ cells at the indicated time points. **E** Quantification of OT-I cell number relative to total CD8⁺ cells (*n* = 3 samples). **F–H** 1.5 × 10⁶ OT-I cells with circAtxn7 silencing or shVec were transferred into MC38K-OVA tumor-bearing mice (*n* = 5 animals). At 20 days after transfer, MC38K-OVA tumors were subjected to analyses of CD8 IHC staining (**F**), tumor growth curves during the course (**G**), and

weights (**H**). Scale bars, 50 μm. **I** Scheme of the preparation and adoptive transfer of tumor-reactive CD8⁺ T cells into autologous CRC PDXs. **J** Biodistribution of tumor-reactive CD8⁺ T cells 21 days after transfer. **K** Representative IHC staining for CD8 in harvested CRC PDX samples 21 days after transfer. Scare bars: 50 μm. **L** NF-κB activity in tumor-infiltrating CD8⁺ T cells purified from ACT-treated PDXs 21 days after transfer (*n* = 3 samples). **M** Growth curves of ACT-treated PDXs (*n* = 5 animals). **N** Weights of ACT-treated PDXs measured 21 days after transfer (*n* = 5 animals). Ut, cells without any treatment (**G**, **H**, and **M**, **N**). Statistical data presented in this figure show mean values ± SD (**A**, **B**, **E**, **G**, **H**, **M**, **N**). ***p ≤ 0.001, ****p ≤ 0.0001 and ns indicates p > 0.05, by two-tailed Student's *t*-test (left panels of **A**, **B**, and **E**), one-way ANOVA (**G**, **H**, and **M**, **N**), or two-sided log-rank test (right panels of **A**, **B**). Source data and exact p values are provided as a Source Data file.

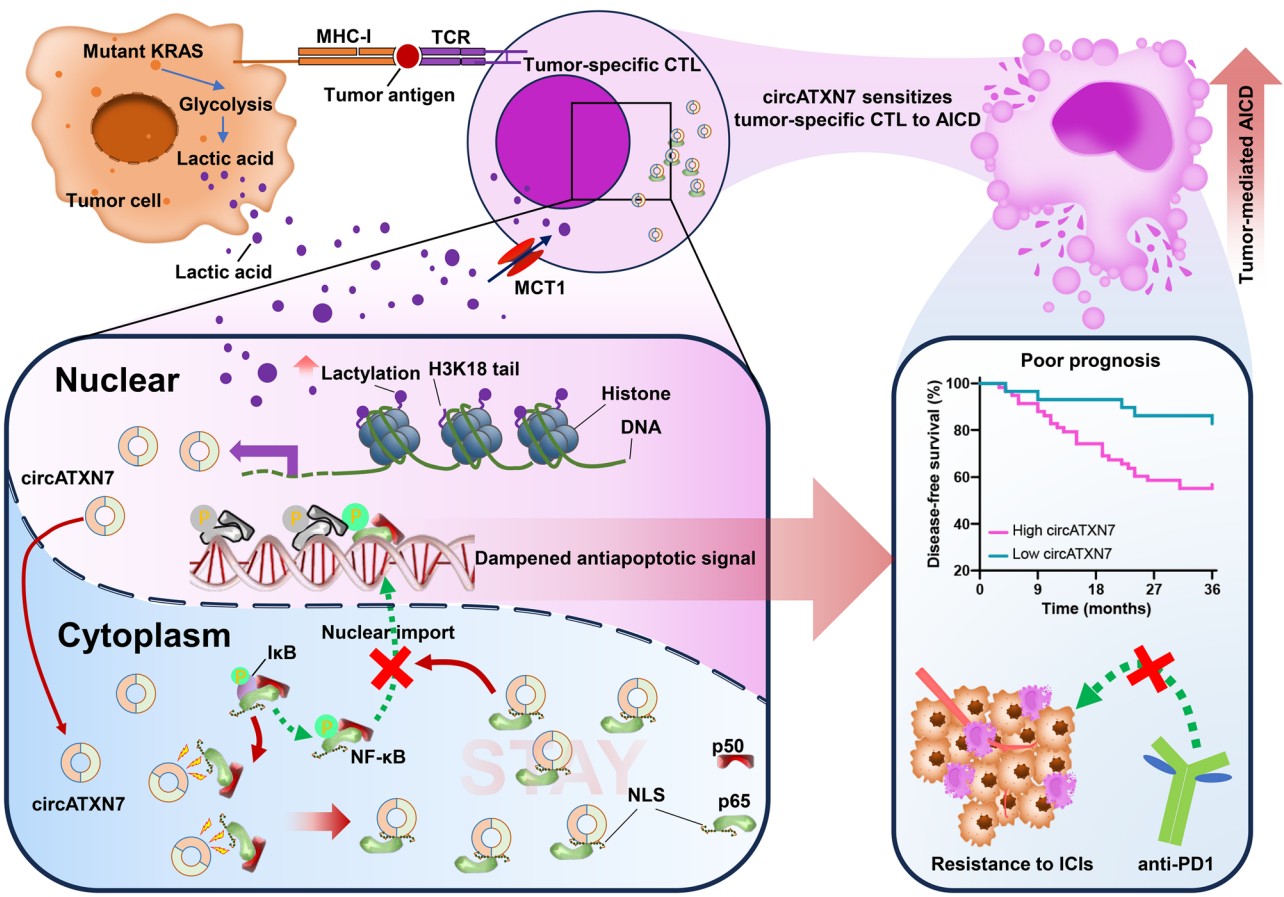

**Fig. 7 | The scheme of the mechanism underlying circATXN7 expression and contribution to the tumor-infiltrating T cell fate decisions.** An NF-κB-interacting circRNA that is activated by histone lactylation sensitizes tumor-specific CTLs to AICD by sequestering p65 in the cytoplasm, thereby causing adverse clinical outcomes and immunotherapeutic resistance.

Table 2. Animals were bred under specific pathogen-free conditions at the Experimental Animal Center of Sun Yat-sen University. Female mice were used for all animal work under the protocols approved by the Institutional Animal Care and Use Committee (IACUC), Sun Yat-sen University (approval number: SYSU-IACUC-2020-000438, SYSU-IACUC-2021-000285 and SYSU-IACUC-2021-000642).

## AICD induction and determination

An ex vivo model system was established to mimic AICD according to previous reports with modification[28,76]. Briefly, purified T cells were activated by PHA (1 μg/ml; Sigma) for 18 h and then cultured with the cytokine IL-2 (25 U/ml; PeproTech) for an additional 5 days. To induce AICD, activated T cells were stimulated with anti-CD3 (10 μg/mL; BD Bioscience), and tumor-specific CTLs were treated with anti-CD3 (10 μg/mL; BD Bioscience) or autologous tumor cells pre-dyed by

CellTtracker Deep Red Dye (Thermo Fisher Scientific) at a 1:1 ratio for 18 h at 37 °C. Anti-CD3-induced T cell apoptosis was determined by flow cytometry analysis using annexin V/7-AAD staining kits (MULTI-SCIENCES). For tumor cell-induced AICD, cocultures of tumor-specific CTLs and autologous tumor cells were subjected to flow cytometric cell sorting to exclude tumor cells and retrieve tumor-specific CTLs. Afterwards, purified CTLs were stained with annexin V/7-AAD, followed by flow cytometry analysis. Apoptotic cell percentages included the percentages of early (annexin V⁺ 7-AAD⁻) and late apoptotic cells (annexin V⁺ 7-AAD⁺). Specific apoptosis was calculated as: (induced apoptosis percentage minus spontaneous apoptosis percentage) / (100% minus percentage of spontaneous apoptosis) × 100%. To evaluate specific signaling pathways effects on AICD, T cells were treated with vehicle, 2 mM Bay11-7082 (MedChemExpress), or 6 mM JSH-23 (MedChemExpress) for 1 h, or 10 mM lactic acid (Sigma) for 12 h prior

to AICD induction. To inhibit tumor-mediated AICD, tumor cells were incubated with HLA class I blocking antibodies (10 μg/ml, Thermo Fisher Scientific), or isotype antibodies for 2 h at 37 °C before the addition of tumor-specific CTLs. In some experiments, T cells were treated with 3 mM 3-hydroxy-butyrate (3-OBA; Sigma), or 10 nM AZD3965 (MedChemExpress) for 4 h, followed by 10 mM lactic acid stimulation for 12 h before AICD induction.

## circRNA profile

For circRNA-sequencing, total RNA was extracted from PHA-activated CD8$^+$ T cells with 10 mM lactic acid (Sigma) or PBS treatment using Trizol (Invitrogen). The rRNA and linear RNA were removed using an epicenter Ribo-Zero rRNA Removal Kit (Illumina) and RNAse R (Epicenter), respectively. Afterwards, a cDNA library was constructed, followed by deep sequencing using an Illumina HiSeq 3000 (Illumina) at RiboBio Co. Ltd (Guangzhou, China). circRNAs were identified using CIRI2 and CIRCexplorer2 software.

## Chromatin immunoprecipitation sequencing

The ChIP assay was performed using a Pierce™ Magnetic ChIP Kit (Thermo Fisher Scientific) according to the manufacturer's instructions. Briefly, after cross-linking and chromatin digestion, digested chromatin was incubated with 5 μg anti-H3K18la (PTM Bio), anti-EP300 (Cell Signaling Technology), or anti-IgG (Cell Signaling Technology) antibodies at 4 °C overnight. Protein A/G magnetic beads were then added into the lysate the following morning and incubated for another 4 h. The immunoprecipitated DNA was purified and then used to perform qRT-PCR or RNA sequencing.

## RNA sequencing

For RNA-Seq, tumor-infiltrating CD8$^+$ T cells purified by flow cytometry ($n$ = 600–800 cells per samples) from WT or $circAtxn7^{CKO}$ MC38K tumor-bearing mice were captured directly into lysis buffer containing 10 μM dNTP mix, 10 μM Oligo dT primer, 1% Triton X-100, and 40 IU/ml RNase inhibitor. Full-length mRNA was reverse transcribed, amplified, and sequenced using the Smart-seq2 protocol. Reads were aligned to the mouse reference genome version mm10 using STAR (version 2.6.1b) and quantified using HTSeq (version 0.11.0). Gene counts were normalized using DESeq2 (version 1.32.0) to estimate gene expression levels and identify differentially expressed genes.

## Cell lines

The murine colon carcinoma cell line MC38 were obtained from Kerast Inc. The murine pancreatic adenocarcinoma cell line Pan02 were kindly provided by Prof. Qiongcong Xu (The First Affiliated Hospital, Sun Yat-sen University). The murine melanoma cell line B16F10, and human embryonic kidney 293 T (HEK293T) and T2 cells were originally obtained from the American Type Culture Collection. Cells were cultured in DMEM medium (Gibco) supplemented with 10% fetal bovine serum (Gibco or Nanjing Ozfan Biotechnology Co., Ltd.). All cell lines were identified by short tandem repeat profiling, tested negative for mycoplasma contamination, and grown according to standard protocols.

## Primary cells

Human peripheral blood mononuclear cells (PBMCs) were donated by healthy donors or CRC patients, and isolated using Ficoll-Paque (GE Healthcare) according to the manufacturer's instructions. CD8$^+$ T cells were obtained from PBMCs using CD8$^+$ T cell Isolation Kits (Miltenyi Biotec). Primary CRC cells and tumor-specific CTLs were purified from freshly resected tumor samples of HLA-A2$^+$ patients with CRC expressing CEA using EpCAM$^+$ microbeads (Miltenyi Biotec) and anti-CEA Pentamer-PE (YLSGANLNL; ProImmune), respectively. OT-I cells were acquired with the spleens from transgenic OT-I expressing mice

using CD8$^+$ T cell Isolation Kits (Miltenyi Biotec). Cell populations were confirmed to be > 90% pure by flow cytometric analysis.

## Dendritic cell (DC) and tumor-reactive T cell preparation

The isolated PBMCs were cultured in RPMI-1640 medium supplemented with 1% fetal bovine serum for 1 h, after which adherent monocytes were cultured for 5 days in VIVO medium (Lonza Walkersville) containing 100 ng/mL GM-CSF (PeproTech) and 30 ng/mL IL-4 (PeproTech). Half of the culture medium was replaced with fresh medium and cytokines every two days. Afterwards, the obtained DCs were matured through incubation with 10 ng/mL TNF-α (Peprotech) for 24 h and then pulsed for 24 h with autologous primary tumor cell lysates by freeze-thawing with liquid nitrogen (200 μg protein/1 × 10$^6$ cells/ml) to generate autologous tumor-antigen-loaded DCs. CD8$^+$ T cells purified from PBMCs of the same donors were activated by coculture with autologous tumor-antigen-loaded DCs at a ratio of 5:1 in VIVO medium supplemented with 25 IU/ml IL-2 (Peprotech) for 5 days to obtain tumor-reactive T cells.

## T cell transduction

Lentivirus was used for T cell transduction. To produce lentivirus, packaging vectors and lentiviral transfer vectors were transfected into 293 T cells using polyethyleneimine (Polysciences). After 48 h and 72 h post-transfection, viral supernatants were collected and filtered using a 0.45 μm syringe filter (Millipore), and then they were concentrated by centrifugation at 1600 × $g$ in ultrafiltration tubes (Millipore). Activated T cells were cultured with concentrated lentivirus (multiplicity of infection of 25) supplemented with 8 μg/ml polybrene (Sigma), centrifuged at 850 × $g$ for 80 min at 32 °C, and cultured for 9 h. The transduction was repeated on two consecutive days and cells were cultured in X-VIVO medium supplemented with 100 IU/ml IL-2. For OT-I T cell transductions, total splenocytes from OT-I mice were stimulated with 10 nM peptide (OVA)$_{257–264}$ (SIINFEKL; Sigma) in the presence of 100 IU/ml IL-2 for 5 days. Afterwards, they were purified and subjected to lentivirus transduction as described above.

## Plasmids construction and lentivirus infection

Lentiviral vectors with or without Luc, mCherry or GFP expression were used to transduce shRNAs against circATXN7 (human), circAtxn7 (mouse), or scrambled vectors into T cells. To generate the circATXN7 or circAtxn7 overexpression plasmid, the full-length circATXN7 or circAtxn7 with or without p65-binding site mutant cDNA (5′ GGTCGGGG 3′ were altered to 5′ AAAAAAAA 3′) was cloned into the pLO-ciR vector (Guangzhou Bioyard Biotechnology Development Co., Ltd). A series of human p65 gene deletion mutations with flag-tag were cloned into the pSin-puro vector (Guangzhou Bioyard Biotechnology Development Co., Ltd). The circATXN7-binding site-mutated p65 (R303A) were generated as well. The above plasmids were used for T cell transduction as described above.

To generate constructs for mutant Kras overexpression, the coding sequence of $Kras^{G12D}$, $Kras^{G12V}$, or $Kras^{G13D}$ was amplified using ClonExpress II One Step Cloning Kit (Vazyme), and then the cDNA was inserted into the lentiviral expression vector pCDH-CMV-Puro. The constructs were verified by DNA sequencing. For stable transfection, the pCDH-CMV-Puro lentiviral vectors encoded $Kras^{G12D}$, $Kras^{G12V}$, or $Kras^{G13D}$ were adopted to produce lentiviral particles carrying the above-mentioned vectors in HEK293T cells using Lipofectamine 3000 (Invitrogen). MC38, Pan02, and B16F10 cells were infected with lentivirus followed by selection with puromycin to generate the cells expressing mutant Kras (designated as MC38K, Pan02K, and B16F10K, respectively), while cells with transfection of the corresponding empty vectors were used as controls (MC38, Pan02, and B16F10 hereafter, respectively). The successful generation of Kras mutant cells was confirmed by Sanger sequencing. In addition, to generate MC38K-OVA and Pan02K-OVA cell lines, the MC38K and Pan02K were infected with

lentivirus produced by HEK293T cells using pLEX307 lentiviral vector encoded OVA and selected by neomycin.

## RNA isolation and qRT-PCR
Total RNA was extracted from cells using TRIzol (Invitrogen). The nuclear and cytoplasmic fractions were purified by NE-PER™ Nuclear and Cytoplasmic Extraction Reagents (Thermo Fisher Scientific) as per the manufacturer's protocol. Total RNAs were then reverse-transcribed using a reverse transcription kit (Takara). We performed qRT-PCR using a SYBR Green PCR Kit (Takara). All reactions were performed in a 10 μl reaction volume in triplicate and GAPDH or served as the reference. The $2^{-\Delta\Delta CT}$ method was applied to calculate relative expression. The primer sequences are shown in Supplementary Table 2.

## RNase R treatment
RNase R (Epicentre Technologies, Madison, WI, USA) was used to assess the stability of circRNA. Total RNA (2 μg) was mixed with 0.6 μl 10 × RNase R Reaction Buffer and 0.2 μl RNase R or DEPC-treated water (control group). The samples were then incubated at 37 °C for 15 min. The circATXN7 and linear ATXN7 expression levels were determined by qRT-PCR.

## Actinomycin D assay
To assess circRNA half-life, gene transcription was blocked by adding 2 mg/mL Actinomycin D (Sigma) to the cell culture medium. DMSO was used as a negative control. Cells were harvested at 0, 4, 8, 12, and 24 h, and circATXN7 and linear *ATXN7* stabilities were analyzed by qRT-PCR.

## circATXN7-p65 structure modeling
The p65 crystal structure was downloaded from the Protein Data Bank. The secondary structure of circATXN7 was formed using the Vienna RNA web server, and 3dRNA predicted the 3D structure of circATXN7. Protein-nucleic acid structures interaction sites were calculated using HDOCK and visualized by chimera.

## FISH assay
The FISH assay was carried out using a Fluorescent In Situ Hybridization Kit (RiboBio) as per the manufacturer's protocols. Hybridization was performed with fluorescence-labeled circATXN7 probes (GenePharma), followed by analysis using confocal microscopy. The probe sequences are shown in Supplementary Table 2.

## Immunoblots
Cells were collected, washed, and lysed in radioimmunoprecipitation assay (RIPA) lysis buffer containing Protease and Phosphatase Inhibitor (Thermo Fisher Scientific). A bicinchoninic acid (BCA) kit (CWBio) was then used to detect the concentration of protein. Equivalent amounts of protein were separated on 10% SDS-PAGE and then transferred to a polyvinylidene fluoride (PVDF) membrane (Millipore). The membrane was then blocked with Tris-buffered saline with Tween 20 (TBST) buffer containing 5% skim milk powder and incubated with corresponding primary antibodies at 4 °C overnight. The primary antibodies used were anti-p65 (Cell Signaling Technology), anti-IκBα (Cell Signaling Technology), anti-p50 (Cell Signaling Technology), anti-Histone-H3 (Abcam), anti-β-actin (Abcam), anti-Flag (Cell Signaling Technology), GAPDH (Cell Signaling Technology), anti-L-Lactyl Lysine (PTM Bio Inc), anti-Lactyl-Histone H3 (Lys18) Rabbit mAb (PTM Bio Inc), anti-KRAS (Abcam), anti-KRAS$^{G12D}$ (Abcam), anti-Lamin A (Abcam). Membranes were then washed with TBST three times and incubated with horseradish peroxidase (HRP)-conjugated anti-rabbit or anti-mouse secondary antibody (Cell Signaling Technology) for 1 h at room temperature. Signals were developed with ECL Blotting Detection Reagents (Thermo Fisher Scientific).

## Immunohistochemistry (IHC)
IHC was performed on formalin-fixed, paraffin-embedded (FFPE) tissue sections. The primary antibodies in this work were anti-human CD8 (Abcam), anti-mouse CD8 (BD Bioscience). Tissue sections were incubated with primary antibodies at 4 °C overnight and then incubated with secondary antibody. DAB complex was used as the chromogen. The nuclei were counterstained with hematoxylin. For quantification, the slides were assessed by two independent pathologists who were blinded to the patients' clinical information. The CD8$^+$ cell density was calculated as the number per mm$^2$ of CD8$^+$ cells on each slide.

## Transwell migration assay
Cell migration assays were performed using a 5 μm pore Transwell filter system (BD Biosciences). $1 \times 10^5$ T cells transduced with GFP-shcircATXN7 (in human cells) or GFP-shcircAtxn7 (in murine cells) were mixed with $1 \times 10^5$ T cells transduced with mCherry-shVec and seeded on the upper chambers. Media containing recombinant human or murine CXCL10 was placed in the lower well. Following an incubation period of 3 h, the migrated cells were collected and analyzed by flow cytometry.

## RNA in situ hybridization (ISH) assay
The circATXN7$^+$ cells in paraffin-embedded CRC tissues were detected using digoxin-labeled circATXN7 probes and an ISH Detection Kit (BosterBio) on the basis of the manufacturer's instructions. The sections were dewaxed and rehydrated, followed by digestion with pepsin. The sections were hybridized with circATXN7 probes at 37 °C overnight. The sections were incubated with an anti-digoxin monoclonal antibody conjugated with alkaline phosphatase and then incubated with 3, 3'-diaminobenzidine (DAB). For quantification, the slides were assessed by two independent pathologists who were blinded to the patients' clinical information. The circATXN7$^+$ cell density was calculated as the number per mm$^2$ of circATXN7$^+$ cells on each slide.

## Immunofluorescence (IF)
Cells were fixed with 4% paraformaldehyde for 15 min at room temperature and permeabilized using 0.2% Triton X-100. Frozen sections were obtained from fresh tissues after surgery. Paraffin-embedded tumor sections were deparaffinized and antigen repaired. Afterwards, the tissue sections or cell-adherent slides were blocked with 10% bovine serum albumin (BSA; Sigma) for 1 h and incubated with anti-human CD8 (Abcam) and anti-p65 (Cell Signaling Technology). After rigorous washing with PBS, sections or slides were incubated with fluorescently conjugated secondary antibodies (Thermo Fisher Scientific). Isotype matched antibodies were used as controls. After counterstaining with DAPI (Abcam), we acquired images using a confocal laser-scanning microscope (Leica TCS-SP8) with a core data acquisition system (Applied Precision).

## RNA Immunoprecipitation (RIP)
Anti-p65 (Cell Signaling Technology) and anti-Flag (Cell Signaling Technology) were used for the p65 and Flag RIP assays, respectively. RIP was done using the Magna RIP RNA-binding protein immunoprecipitation kit (Millipore) in light of the manufacturer's introductions. The isolated RNA was purified and then subjected to qRT-PCR. The enrichment values were normalized to the level of background RIP, as detected by IgG isotype control.

## circRNA pull-down
Biotin-labeled circATXN7 probes and control probes were used for circRNA pull-down. In brief, the cells were lysed in co-immunoprecipitation (CoIP) buffer and incubated with the circATXN7 probe at room temperature for 2 h. Then, the cell lysate was incubated with 50 μl of Streptavidin C1 magnetic beads (Invitrogen) at room temperature for 1 h. The beads were washed briefly five times with co-IP buffer, and the bound proteins in the pull-down material were analyzed by immunoblots.

### In vitro transcription and cyclization of circATXN7

The DNA template used for biotinylated circATXN7 in vitro synthesis was generated by PCR and purified using a DNA Gel Extraction Kit (Axygen). In vitro transcription was performed using the T7-Flash BiotinRNA Transcription Kit (Epicentre, biotin labeling) according to the manufacturer's instructions. RNA was subsequently purified by phenol-chloroform extraction. For linear RNA in vitro cyclization, the RNA products were incubated with the indicated DNA splints (molar ratio = 1:1.5) at 90 °C for 5 min, and then cooled to room temperature over 20 min. Ligation to form circRNAs was then performed overnight at 16 °C with T4 DNA ligase (NEB). The sample was then treated with RNase R and DNase I at 37 °C for 30 min and subsequently purified by phenol-chloroform extraction.

### Determination of NF-κB activity

NF-κB activity was measured using a NF-κB p65 Transcription Factor Assay Kit (Abcam) as per the manufacturer's instructions. NF-κB p65 contained in a nuclear extract, binds to the NF-κB p65 response element, and is detected using an anti-NF-κB p65 antibody. A secondary antibody conjugated to HRP is added to provide a colorimetric readout at 450 nm using a Microplate Reader. Each sample was determined in triplicate.

### In vitro circRNA-protein binding assay

For the in vitro binding assay, His-labeled p65, p50, or IκBα and biotinylated circATXN7 were incubated in 500 μl RIP buffer at room temperature for 1 h. Afterwards, we added 50 μl of washed Dynabeads M-280 Streptavidin (Invitrogen) to each binding reaction and incubated at room temperature for another 1 h. The beads were washed briefly five times with RIP buffer and then boiled in SDS buffer. The bound proteins were detected by immunoblots.

### Lactic acid and cytokine detection

Fresh tumor tissues were collected from CRC patients and homogenized. In light of the manufacturer's introductions, the protein concentrations in each homogenized tissue sample were determined and the lactic acid, CXCL9, CXCL10 and CXCL12 level equal of sample each was determined using a Lactate Assay Kit (BioVision), Human CXCL9 ELISA Kit (Abcam,), Human IP-10 ELISA Kit (Abcam) and Human SDF1 alpha ELISA Kit (Abcam), respectively.

### PCR and Agarose gel electrophoresis

PCR assays were conducted using Premix Taq (Ex Taq II) (Takara) according to the manufacturer's instructions. The PCR products were then submitted to agarose gel electrophoresis in 2% agarose with TAE buffer by using an electrophoresis system (BIO-RAD) and visualized by using an imaging system (BIO-RAD).

### Clinical response assessment

The clinical response for CRC patients receiving immune checkpoint inhibitors (ICIs) was evaluated by computed tomography or magnetic resonance imaging radiologic data according to Formal Response Evaluation Criteria in Solid Tumors (RECIST), version 1.1, as follow: complete response (CR), disappearance of all target lesions; partial response (PR), ≥30% decrease; PD, ≥20% increase over smallest sum observed; and SD, meeting none of the other criteria. Responders were defined as patients achieving PR or CR.

### Subcutaneous xenografts

To generate subcutaneous xenografts, MC38 ($5 \times 10^5$ cells per mouse), MC38K ($5 \times 10^5$ cells per mouse), Pan02 ($3 \times 10^5$ cells per mouse), Pan02K ($3 \times 10^5$ cells per mouse), and B16F10K ($2 \times 10^5$ cells per mouse) cells were subcutaneously injected into dorsal part of wild-type C57BL/6 J mice or age- and sex- circAtxn7$^{CKO}$ mice (-6−8 weeks old). Each group consisted of five mice. Tumor growth was monitored by digital calipers, and tumor volumes were recorded using the following

formula: Volume = (longer diameter × shorter diameter$^2$)/2. If animals appeared moribund or the diameter of the tumors reached 15 mm, the mice were sacrificed. In some cases, the maximal tumor burden permitted has been exceeded the last day of measurement and the mice were immediately euthanized. At the indicated endpoints, mice were sacrificed and tumors or tumor-infiltrating T cells were subjected to gross inspection, IHC analysis, flow cytometry, qRT-PCR, or RNA-seq.

### Genetically engineered CRC mouse model

To generate the genetically engineered CRC mouse model, the Villin-Cre$^{ERT2}$ mice were crossed with LSL-Kras$^{G12D/+}$ mice and Apc$^{flox/+}$ mice to obtain Villin-Cre$^{ERT2}$Apc$^{flox/+}$ (Kras$^{WT}$) or Villin-Cre$^{ERT2}$ Kras$^{G12D/+}$Apc$^{flox/+}$ (Kras$^{MUT}$) mice. Each group consisted of five mice. When mice were at the age of 8 weeks, 1 mg/mL 4-hydroxytamoxifen (4-OHT) was introduced into the adult colon via enema. All mice were sacrificed 10 weeks later, and the colonic tumors were for IHC analysis and flow cytometry.

### Orthotopic xenograft CRC mouse model

For the construction of the orthotopic xenograft CRC mouse model, the cecum of anesthetized circAtxn7$^{CKO}$ or WT mice was exteriorized through an abdominal laparotomy. MC38 ($5 \times 10^5$ cells per mouse), MC38K ($5 \times 10^5$ cells per mouse) in 50 μl PBS were injected into the cecum submucosa using 30-G insulin-gauge syringe. Each group consisted of five mice. Mice were sacrificed 24 days after injection. Intestines, livers, and lungs were harvested to assess the tumor burden. Cryosections of the harvested organs were stained using H&E for histological assessment. Genomic DNAs were isolated from the rest of the organs for qRT-PCR analysis of CMV (only present in the injected cells transduced with the pCDH-CMV-Puro lentiviral vectors). In some experiments, to deplete CD8$^+$ or CD4$^+$ T cells in mice in vivo, two doses (150 μg/dose) of either YTS-191 (anti-CD4 depletion antibody) or YTS-169 (anti-CD8 depletion antibody) were injected intraperitoneally before orthotopic injection of MC38K cells, followed by eight consecutive injections every three days.

### Tumor-infiltrating T cell isolation in mice

To isolate tumor-infiltrating T cells in mouse tumor models, subcutaneous or orthotopic xenografts were harvested, minced and digested with 0.5 mg/ml Collagenase IV (Sigma) plus 200 IU/ml DNase I (Sigma) for 1 h at 37 °C, and then passed through 40 μm filters to remove undigested tumor tissues. Tumor-infiltrating T cells were then isolated by Ficoll-Paque PLUS (GE Healthcare) density gradient separation and purified by CD8$^+$ T cell Isolation Kits (Miltenyi Biotec) or Fluorescence Activating Cell Sorter. Purified cells were subjected to further analyses as indicated.

### Anti-PD1 therapy

The effects of circATXN7 on anti-PD1 therapy were assessed using subcutaneous xenografts generated from MC38K, Pan02K, and B16F10K cells in wild-type C57BL/6 J mice or age- and sex- circAtxn7$^{CKO}$ mice (-6−8 weeks old). When the tumors were palpable, the mice were randomly assigned to the indicated groups, and anti-PD1 monoclonal antibody (BioXcell) or IgG isotype control antibody (BioXcell) was intraperitoneally injected every 3 days at a dose of 100 μg/injection (n = 5 mice per group). Tumor volume was measured as described above, and mice were sacrificed when the tumor longer diameter reached 15 mm, recorded as death for survival curve.

### Patient-derived-xenograft implantation

Primary CRC specimens were collected from treatment-naive patients with CRC cancer who received tumor resection at The Sixth Affiliated Hospital of Sun Yat-sen University. NOD.SCID mice (5−7 weeks old) under pathogen-free conditions were used for patient-derived xenograft transplantation, with a minimum of five mice per group. The time from patient collection to mouse implantation did not exceed 120 min.

Tumor formation was monitored weekly with calipers in the three months following implantation. When tumors reached ~15 mm, the mice were killed, and tissue fragments were retransplanted into another cohort of mice. After 3–5 repeated cycles, no lymphocytes could be found in the PDX tumor grafts, which were used here for further experiments.

## PDX dissociation

PDX tumors were cut into ~1 mm³ fragments and incubated for 20 min with collagenase type III (Sigma), in RPMI-1640 medium containing 2% FBS (5 ml/g tumor tissue) at 37 °C. The tumor pieces were transferred to a tissue digestion C-tube (Miltenyi Biotec) and further dissociated enzymatically and mechanically on a gentle-MACS Dissociator (Miltenyi Biotec) to generate a single-cell suspension. Afterwards, CD8⁺ T cells were purified with human CD8 Microbeads (Miltenyi Biotec) or Fluorescence Activating Cell Sorter. Purified cells were subjected to flow cytometry, qRT-PCR, immunoblots, or NF-κB activity assay.

## Adoptive T cell transfer (ACT) therapy

Adoptive T cell transfer therapy was performed in PDX tumors, MC38K and Pan02K tumors. In ACT therapy for PDX tumors, we firstly prepared tumor-reactive T cells and autologous tumor-antigen-loaded DCs as indicated above. Then, tumor-reactive T cells were transduced with GFP-tagged circAtxn7 shRNA or mCherry-tagged empty vector before transfer. After tumor formation, $2.5 \times 10^6$ tumor-reactive T cells were co-injected with $0.5 \times 10^6$ antigen-loaded DCs into the PDX-bearing mice via the tail vein. Each group consisted of five mice. Tumor growth was monitored and recorded every week and the tumor volume was recorded. At 3 weeks after transfer, mice were sacrificed and tumors or tumor-infiltrating T cells were subjected to further analysis.

In ACT therapy against MC38K and Pan02K tumors, we firstly generated MC38K and Pan02K cells expressing the cognate antigen (MC38K-OVA and Pan02K-OVA). Then MC38K-OVA ($5 \times 10^5$ cells per mouse) and Pan02K-OVA ($3 \times 10^5$ cells per mouse) cells were used to establish subcutaneous xenografts. Purified OT-I cells were activated by cognate SIINFEKL peptide, and then were transduced with GFP-tagged circAtxn7 shRNA or mCherry-tagged empty vector. When MC38K-OVA or Pan02K-OVA tumors were palpable, tumor-bearing mice were intravenously injected with $1.5 \times 10^6$ OT-I CD8⁺ T cells. Each group consisted of five mice. Tumor growth was monitored and the tumor volume was recorded. On 20 days after transfer, mice were sacrificed and tumors or tumor-infiltrating T cells were subjected to further analysis.

## Monitoring transferred T cell distribution in PDX tumors

We monitored transferred T cell distribution using in vivo bioluminescent imaging as well as flow cytometry analysis. Firstly, lentiviruses carrying luciferase plasmids were pretransduced into tumor-reactive T cells to determine the distribution of transferred T cells in vivo via bioluminescent flux in PDX-bearing mice by IVIS Lumina imaging (Xenogen IVIS Lumina System). One minute before imaging, mice were injected intraperitoneally with D-luciferin (PerkinElmer) and anesthetized with 3% isoflurane. Living Image software version 3.0 (Caliper Life Sciences) was used for image analysis.

Then, in some experiments with PDX tumors, tumor-reactive T cells transduced with GFP-shcircATXN7 and mCherry-shVector were mixed in vitro at a 1:1 ratio and then injected intravenously into PDX-bearing mice after palpable tumor formation. In subcutaneous xenografts generated from MC38K-OVA cells, activated OT-I CD8⁺ T cells transduced with GFP-shcircAtxn7 were mixed with OT-I CD8⁺ T cells transduced with mCherry-shVector at a 1:1 ratio and then co-injected intravenously into MC38K-OVA-bearing mice. At 1, 3, 5, 7, 14, and 21 days after transfer, CD8⁺ T cells in PDX tumors or MC38K-OVA tumors were purified as described above, and then the purified cells

were subjected to flow cytometry to analyze the disruption difference between GFP- and mCherry-tagged T cells.

## Statistics and reproducibility

No statistical methods were used to predetermine sample size. Data were presented as mean ± standard deviation (SD) except were stated otherwise. Statistical analysis was conducted using SPSS statistical software (version 16.0; SPSS Inc., Chicago, Illinois) or GraphPad Prism (version 8.0.2; GraphPad Software, San Diego, CA, USA). Microscopy images shown are representative of at least 3 independent experiments. Western Blot images are representative of three independent experiments. Detailed data processing, sample size and statistical methods for each result were shown in the corresponding figure legends. All $p$ values were two-sided and $p$ value ≤ 0.05 was considered as statistically significant.

## Reporting summary

Further information on research design is available in the Nature Portfolio Reporting Summary linked to this article.

## Data availability

Source data for the circRNA-seq, ChIP-seq, and RNA-seq have been deposited in the Genome Sequence Archive under the accession numbers HRA003320, HRA003223, and CRA007181, respectively. Datasets HRA003320 and HRA003223 are available under restricted access for research purposes only, access can be obtained by the DAC (Data Access Committees) of the GSA-human database. The approximate response time for accession requests is about two weeks. Once access has been approved, the data will be available to download for research purpose, and can only be used for the research group and its research collaborators. The user can also contact the corresponding author directly upon request. The remaining data generated in this study are provided in the Article, its Supplementary Information and Source Data file. Source data are provided with this paper.

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

## Acknowledgements

This work was supported by grants from the National Natural Science Foundation of China (82200569, 82103273, 82203626, 82303440, 82303060, 82000515 and 82370675), Guangdong Basic and Applied Basic Research Foundation (2023A1515010523, 2023A1515010473, 2021A1515110509, 2019A1515110043, 2021A1515111011, 2019A1515110144 and 2022A1515012498), China Postdoctoral Science Foundation (2021M703723 and 2022M723616), the Open Fund of Guangdong Provincial Key Laboratory of Digestive Cancer Research (2021B1212040006), Science and Technology Projects in Guangzhou (202206010062), Medical Scientific Research Foundation of Guangdong Province, China (no. A2021130), Sun Yat-sen University Clinical Research 5010 Program (2016005), Natural Science Basic Research Program of Shaanxi Province (2022JQ-825), Shenzhen "San Ming Projects" Research (Grant No.lc202002), Key Research and Development Program of Guangzhou (No. SL2024B03J00078), Fundamental Research Funds for the Central Universities, Sun Yat-sen University (No. 23xkjc023) and National Key Clinical Discipline.

## Author contributions

H.S.L., Y.X., L.K., and C.Z. conceived the ideas and designed the experiments. H.S.L., W.X.L., Z.X.L., S.J.C., J.H.P., K.X.Z., W.H.L., X.Y., Z.W.Z., X.B.Z., and L.X. performed the experiments. H.S.L., X.R.W., L.X., L.K., and C.Z. analyzed and interpreted the data. H.S.L., W.X.L., and C.Z. wrote the manuscript. H.S.L., W.X.L., Z.X.L., Y.X., S.J.C., J.H.P., K.X.Z., W.H.L., Z.W.Z., X.B.Z., W.H.F., L.H., Z.Z.L., X.R.W., P.L., Y.X., L.K. and C.Z. revised the paper. Z.W.Z., X.Y., X.B.Z., and L.K. performed the statistical analysis. All authors read and approved the final paper.

## Competing interests

The authors declare no competing interests.

## Additional information

[1]Department of Colorectal Surgery, Sun Yat-sen University Cancer Center, Guangzhou, China. [2]State Key Laboratory of Oncology in South China, Collaborative Innovation Center for Cancer Medicine, Sun Yat-sen University Cancer Center, Guangzhou, China. [3]State Key Laboratory of Oncology in South China,

Guangdong Provincial Clinical Research Center for Cancer, Sun Yat-sen University Cancer Center, Guangzhou, China. [4]Department of General Surgery (Colorectal Surgery), The Sixth Affiliated Hospital, Sun Yat-sen University, Guangzhou, Guangdong, China. [5]Guangdong Provincial Key Laboratory of Colorectal and Pelvic Floor Diseases, The Sixth Affiliated Hospital, Sun Yat-sen University, Guangzhou, Guangdong, China. [6]Biomedical Innovation Center, The Sixth Affiliated Hospital, Sun Yat-sen University, Guangzhou, Guangdong, China. [7]Precision Medical Research Institute, the Second Affiliated Hospital of Xi' an Jiaotong University, Xi'an, China. [8]Guangdong Provincial Key Laboratory of Malignant Tumor Epigenetics and Gene Regulation, Medical Research Center, Sun Yat-Sen Memorial Hospital, Sun Yat-Sen University, Guangzhou, China. [9]Breast Tumor Center, Sun Yat-Sen Memorial Hospital, Sun Yat-Sen University, Guangzhou, China. [10]Guangdong Provincial Key Laboratory of Digestive Cancer Research, the Seventh Affiliated Hospital of Sun Yat-sen University, Shenzhen, Guangdong, China. [11]These authors contributed equally: Chi Zhou, Wenxin Li, Zhenxing Liang, Xianrui Wu. [12]These authors jointly supervised this work: Yue Xing, Liang Kang, Huashan Liu. ✉e-mail: xingy28@mail.sysu.edu.cn; kangl@mail.sysu.edu.cn; liuhshan@mail2.sysu.edu.cn

