## [Peer Review File · Nature Communications]

Mutant KRAS-activated circATXN7 fosters tumor immunoescape by sensitizing tumor-specific T cells to activation-induced cell deathREVIEWER COMMENTS

Reviewer #1 (Remarks to the Author): with expertise in cancer, lactylation

In this manuscript, Zhou et al., reports that tumor-specific T cells from KRAS^{MUT} cancers are susceptible to activation-induced cell death (AICD). They found that a circular RNA called circATXN7 is responsible for T cell sensitivity to AICD by inactivating the transcription factor NF- κ B. They also discovered that KRAS mutant tumors produce lactic acid, which leads to histone lactylation and activates the transcription of circATXN7. The upregulation of circATXN7 in tumor-specific T cells is associated with adverse clinical outcomes and immunotherapeutic resistance. Targeting circATXN7 in T cells improves their anti-tumor activities, suggesting a potential therapeutic strategy for KRAS mutant cancers.

The emerging roles of circRNAs in cancer and other physiology, as well as its potential applications, has generated significant interest. However, to convince this reviewer that the claims are supported by the observations described, several major issues need to be addressed.

1. The issue of novelty arises as the same group recently published in *Advances in Science* (DOI: 10.1002/adv.202203757) on the sensitization of CD8⁺ T cells to activation-induced cell death by lactic acid secreted from mutant KRAS colorectal tumors, which inhibits NF- κ B activity. This study employed a different patient cohort with stage IV CRC compared to the stage I-III CRC cohort used in their previous publication, yet observed a similar phenomenon. However, the authors did not provide convincing clinical analyses as they did in their previous paper.

Specifically, the authors should present the following clearly in the main figures:

- A comparison of CD8⁺ T cell infiltration in KRAS^{MUT} tumors versus KRAS^{WT} tumors.
- A comparison of the sensitivity to AICD in CD8⁺ T cells from KRAS^{MUT} tumors versus KRAS^{WT} tumors.
- The authors should first establish the lactate-NF- κ B/AICD axis in their class IV colorectal tumors. This includes demonstrating the difference in NF- κ B activity in CD8⁺ T cells from

KRASMUT tumors compared to KRASWT tumors, showing that NF- κ B is responsible for AICD in CD8+ T cells, proving that KRASMUT tumors secrete more lactic acid compared to KRASWT tumors, and demonstrating that lactic acid indeed increases sensitivity to activation-induced cell death.

2. The authors proposed a new mechanism involving lactylation-circATXN7 to explain the inhibition of NF- κ B by lactate, but the mechanism is biased and not clearly demonstrated. Specifically, the following issues need to be addressed to improve the paper's credibility:

- The authors focused only on MCT1 and intracellular lactate while ignoring the synergistic effect of lactate with GPR81.
- It is also subjective to focus only on circular RNA and ignore all other regulatory mechanisms in NF- κ B pathway.
- The authors' selection of circATXN7 from among the 130 up-regulated circular RNAs by lactate is subjective and lacks sufficient rationale.
- The specific regulation of lactate and lactylation on circATXN7 in CD8+ T cells and not in other cell types in TME is not clear.
- The mechanism of binding between circATXN7 and p65 is not clear. The authors did not provide sufficient details on the construction of circATXN7 and p65 mutants. The exact sequence of their blocking oligos, the complementarity of the blocking oligos to circATXN7, and the specific sites mutated on circATXN7 and p65 were not provided. The authors also did not explain why the blocking oligos were unable to completely block the interaction between circATXN7 and p65. More information and experimental details are needed to fully understand the mechanism of action of circATXN7 in inhibiting the NF- κ B pathway.

3. There is a lack of clarity on why targeting circATXN7 can specifically inhibit KRASMUT tumors while not KRASWT tumors. Since all mechanistic studies were carried out in WT CD8+ T cells, blocking circATXN7 should be able to restore nuclear NF- κ B activity in all CD8+ T cells. Additionally, a considerable proportion of KRASWT tumors also secrete lactate, and a certain proportion of KRASWT tumors contain circATXN7+ T cells (93 out of 182). Therefore, the authors should provide further clarification on the specificity of the effect of targeting circATXN7 in KRASMUT tumors.

Reviewer #2 (Remarks to the Author): with expertise in cancer, circRNA

In this manuscript, Zhou et al. identified that circATXN7, a NF- κ B-interacting circRNA in CD8+ T cells, was upregulated in KRAS^{MUT} CRC tissues and participated in regulation of T cells sensitivity to AICD. Mechanistically, circATXN7 binds to p65 subunit and isolates it in the cytoplasm thereby inactivating NF- κ B. The work presented here is interesting because it provides a new mode of action of circular RNA in the context of CD8+ T cells fate decision and anti-tumor immunotherapy. While the amount of data presented are impressive, a number of issues need to be resolved. Please see specific comments below.

Major concerns:

(1) Whether the expression level of circATXN7 is different in tumor infiltrated CD8+ T cells derived from KRAS^{MUT} and KRAS^{WT} CRC tissues?

(2) The authors have shown that circATXN7 can directly bind with p65 in CD8+ T cells and sequester it in the cytoplasm. p65 is known as a very abundant protein in CD8+ T cells, but circRNAs usually with a low abundance in cells. How many copies of circATXN7 and p65 are expressed in CRC patients and C57BL/6 mice CD8+ T cells? Can the stoichiometry of these molecules support the proposed model?

(3) The authors explored the roles of circATXN7 and circAtxn7 in CD8+ T cells derived from CRC patients and C57BL/6 mice, respectively. Is the sequence of circATXN7 conserved in mouse or different between human and mouse?

(4) Figures 3A, B, G and Supplementary Figures 3 B, I. How did the authors design the probes used in ISH and FISH experiments to show the localization and level of circATXN7? Of note, as the shared primary sequences between circATXN7 and its cognate mRNA, it's difficult to specifically target circATXN7 without off-target on its cognate mRNAs. How do they exclude the ISH and FISH signals are not circATXN7 cognate mRNAs, in particularly, they both are localized in the cytoplasm?

(5) In consideration of the shared primary sequences between circATXN7 and ATXN7 mRNA, the authors should detect whether shcircATXN7 has effect on the ATXN7 mRNA level.

Minor concerns:

(1) The authors should confirm the KO efficiency of circAtxn7 in the circAtxn7 CKO mice and show this data in the manuscript.

(2)Supplementary Fig. 2: both negative and positive controls should be added in RIP assays to show the specificity.

(3) It is recommended to add a working model in the manuscript for the functions and mechanism of circATXN7 on tumor progression.

Reviewer #3 (Remarks to the Author): with expertise in colorectal cancer, immunotherapy

This study by Zhou et al. identified a novel circular RNA circATXN7 that plays an important role in modulating the antitumor activity of tumor-infiltrated CTLs in Kras-mutant colorectal cancer. The in vitro and in vivo studies of circATXN7 are overall sound. Particularly, the generation of circATXN7 KO mice provided a great animal model to reveal the function of this new circATXN7 in the tumor-associated CTLs. However, the manuscript appears to be disconnected due to lack of supporting evidence that links AICD, circATXN7 and lactic acid in the tumor microenvironment. Below are my major concerns:

1. The previous work by the group and others have identified the role of lactic acid in AICD of CTLs. The Fig 1 and Fig 2A-D seem to further support the previous studies, but not directly connect to the circATXN7, which therefore could be moved to supplemental information.
2. There is no evidence why circRNAs play an essential role in AICD of CTLs. No strong rationale was provided although the identified circATXN7 appears to have important functions.
3. Given the importance of circATXN7 in CTLs, it is crucial to know the copy numbers of this circRNA, ensuring that their amount is enough to directly bind p65 for inhibition at a close molar ratio.
4. While circATXN7 is only expressed and function in CTLs, all the studies in the manuscript do not involve autologous CTLs in the tumor. For example, Fig. 1c, it is not understandable why tumor-specific CTLs are not used. The expansion of tumor-specific CTLs are well established for this type of study. The same weakness also applies to the Fig. 2c study.
5. FISH staining of circATXN7 is not accurate to compare the cell-specific expression among various cell types. It is recommended to conduct scRNA-seq of total tumor and stroma cells or QRT-PCR of each cell type isolated by cell sorting.
6. It is unclear if the expression and function of Atxn7 gene is affected in the circATXN7 KO mice.

In addition to the above major concerns, there are a few minor issues:

1. The introduction part does not provide any useful information. It should include more previous studies in ACID, lactic acid in TME, and etc...
2. In Fig.2B, the inhibition by 3-OBA is significant and its combo treatment with AZD3965 is obviously better than AZD3965 alone. It is not consistent with the text in the manuscript " Results found MCT1 blockade by AZD3965, but not 3-OBA...."
3. MC38 is a MSS colon cancer model, while only MSI-high CRCs are treated by ICIs in clinic. This should be discussed.

Reviewer #4 (Remarks to the Author): with expertise in cancer immunology, colorectal cancer

The authors presents a comprehensive study on the role of circATXN7 in tumor immunoevasion, decoding its role in promoting activation-induced cell death (ACID) in tumor specific CTLs. The study exploits well-designed experimental in vitro and in vivo approaches and holds the potential to have a substantial impact on the development of new immune-therapeutic strategies. However, there are some questions that remained unanswered and need to be addressed to make this study more impactful.

MAJOR COMMENTS:

1. Figure 1D and E. Experiments are convincing but a graph showing biological replicates and deviation would make their conclusions more solid and also help the reader understanding the reproducibility across and within CRC patients. Also, CD8 isolated from healthy donors PBMCs should be included as an additional control to test whether CD8 from CRC patients are more sensitive to ACID in comparison to CD8 from cancer-free donors.
2. At Line 117-118, the authors concluded "autologous tumor cells elicited AICD in activated CTLs from KRASMUT tumors through repeated TCR stimulation". To make this statement I believe the authors should compare side by side the CD8 ACID sensibility upon exposure to tumor coming from KRASMUT and KRASWT CRC tumor. Instead, unless the Reviewer has overlooked (if so please add a more careful description of the exp detailing the source of the CD8, type of tumor ..), it looks like that the exp has been done with CRC tumors that have

not previously classified in KRASMUT and KRASWT. In this case, it would be advisable to show results from CD8 exposed to KRASMUT versus KRASWT and/or correlate the ACID sensitivity to the KRAS mutation.

3. ACID is significantly increased also in CD8 exposed to CEA antigen. Is then ACID increase a common mechanism happening in CD8 exposed to several type of tumors or for instance to those that overexpressed CEA (i.e. breast, lung ovarian)? I think this is an important point that needs to be clarified in order to be able to state that there is an association between mutant KRAS and CD8 ACID sensitivity within the tumor. Also, is the increase in ACID a tumor-specific mechanism or also viral antigen give rise to the same phenomenon?

4. Beside measurement of ACID and NF- κ B activity, it would be important to assess phenotype, function and activation of CD8 T cells exposed to KRASMUT versus KRASWT to understand if the increased in ACID is correlated with loss of functionality and/or differentiation towards an exhausted subset. In this regard and in relationship with the interesting results on the association with response to ICB therapy, it would be important to assess the expression of different co-inhibitory receptors or at least PD1.

5. The author showed that MCT1 blockade by AZD3965 and GPR81 inhibition by 3-OBA significantly reversed the effects of lactic acid on NF- κ B/AICD axis. Does this have a functional effect on CD8 anti-tumor immunity? In other words, would AZD3965 boost the activity of tumor-specific CTLs and impact on their anti-tumor potential against CRC KRASMUT tumors?

5. Results on circATXN7 expression confined to the tumor stroma and specifically to the CD8 compartment are very intriguing. However, the RT-PCR approach needs to be integrated with other methodologies to look at circATXN7 protein expression on the different cell compartment (i.e. IHC on tumor section looking at the co-localization circATXN7 with CD8 or CD4 or other immune cells or stromal cells). Also, in order to conclude that circATXN7 expression is specific to the tumor and confined to the stroma, a normal adjacent tissue should be included as control as well as assess the expression of circATXN7 in peripheral CD8 T cells.

6. Figure 5. Is thymic development, differentiation and frequencies of CD8 T cells normal in circAtxn7CKO mice? Is circAtxn7 deficiency impacting on T cell biology?

7. Figure 5B-C. Results of tumor inhibition after targeting circAtxn7 are solid and reproduced in several different models. However, experiments are stopped when the tumor are very

small, around 500mm³ and sometime even at 300mm³. I believe the author should keep the exp going at least until 1000mm³ to claim a possible curative role for circAtxn7 targeting. Also, author should try to use the same (or similar) scale for comparison purposes. Same applies to the other tumor models used in the manuscript.

8. Figure 5K. Can the author show other T cell features that can be correlated with immunosurveillance (i.e. less exhausted phenotype, increased frequencies of infiltrating T cells)?

9. Figure 6. I would suggest the author to include data on the characterization (function, activation, phenotype...) of OT1 upon circAtxn7 silencing prior and post infusion in tumor-bearing mice?

MINORS COMMENTS:

- INTRODUCTION: author should make an effort to write a more compelling introduction for their study. At the moment it is very poor and does not place their study in the context of the available literature.

- Line 367 and 371 (and somewhere else in the text): “cool” tumors, I guess the authors mean “Cold” tumors.

- The rationale of choosing to study CirRNAs in the context of anti-tumor immunity should be better detailed to make the study more accessible to both expert and non-specialist in the field.

- Figure 2N: What is the viability of cells treated with increasing concentration of lactic acid, especially at 10 mM- which is the concentration used in the exp to determine the link between Lactic acid and circATXN7?

- Line 129-131: the authors state: “Results found MCT1 blockade by AZD3965, but not GPR81 inhibition by 3-OBA, significantly reversed the effects of lactic acid on NF-κB/AICD axis”. To my understanding there is a significant effect also when 3-OBA is used; thus I would suggest the author to temper their conclusion and rephrase the concept.

- seminal results have been published from Steve Rosenberg and Eric Tran on the potential of using T-cell receptors (TCRs) targeting mutant KRAS G12D expressed by the tumors as a tool to increase efficacy of ACT for the treatment of tumor still refractory. Authors should cite these papers as well as others studies investigating the role KRAS mutation on anti-

tumor T cell responses and they should make an effort to comment their results in the context of the available literature.

Point-by-point responses to the comments from Reviewers

We sincerely thank all of the Reviewers for their constructive comments and helpful suggestions. We have addressed all of the raised issues at our best efforts, and hope that our revised manuscript now meets your expectations.

=====

Responses to comments (Reviewer #1)

In this manuscript, Zhou et al., reports that tumor-specific T cells from KRAS^{MUT} cancers are susceptible to activation-induced cell death (AICD). They found that a circular RNA called circATXN7 is responsible for T cell sensitivity to AICD by inactivating the transcription factor NF- κ B. They also discovered that KRAS mutant tumors produce lactic acid, which leads to histone lactylation and activates the transcription of circATXN7. The upregulation of circATXN7 in tumor-specific T cells is associated with adverse clinical outcomes and immunotherapeutic resistance. Targeting circATXN7 in T cells improves their anti-tumor activities, suggesting a potential therapeutic strategy for KRAS mutant cancers.

The emerging roles of circRNAs in cancer and other physiology, as well as its potential applications, has generated significant interest. However, to convince this reviewer that the claims are supported by the observations described, several major issues need to be addressed.

Reply: We truly thank you for the constructive comments and helpful suggestions. We have addressed all of the raised issues at our best efforts. Please find our point-by-point responses to the comments as follows.

Comment 1: *The issue of novelty arises as the same group recently published in Advanced Science (DOI: 10.1002/adv.202203757) on the sensitization of CD8⁺ T cells to activation-induced cell death by lactic acid secreted from mutant KRAS colorectal tumors, which inhibits NF- κ B activity. This study employed a different patient cohort with stage IV CRC compared to the stage I-III CRC cohort used in their previous publication, yet observed a similar phenomenon. However, the authors did not provide convincing clinical analyses as they did in their previous paper. Specifically, the authors should present the following clearly in the main figures:*

Reply 1: We sincerely appreciate the Reviewer's comments. Based on the Reviewer's advice, we performed additional experiments to support our conclusion. Please find our point-by-point responses to the comments as follows.

- *A comparison of CD8⁺ T cell infiltration in KRAS^{MUT} tumors versus KRAS^{WT} tumors.*

Reply 1.1: According to the Reviewer's advice, we performed a comparison of CD8⁺

T cell infiltration in KRAS^{MUT} tumors versus KRAS^{WT} stage IV tumors. As anticipated, the CD8⁺ densities were significantly lower in KRAS^{MUT} stage IV tumors than those of KRAS^{WT} tumors (**Supplementary Fig. 1A**). These findings were incorporated in the revised manuscript.

- *A comparison of the sensitivity to AICD in CD8+ T cells from KRAS^{MUT} tumors versus KRAS^{WT} tumors.*

Reply 1.2: As suggested by the Reviewer, we compared the AICD sensitivity of CTLs from KRAS^{MUT} versus KRAS^{WT} stage IV tumors. After coculturing with autologous tumor cells, we found significantly increased apoptosis in tumor-specific CTLs from KRAS^{MUT} versus KRAS^{WT} stage IV tumors (**Fig. 1A, Supplementary Fig. 2D**). In line with this, treatment with anti-CD3 antibodies elicited massive apoptosis of tumor-specific CTLs from KRAS^{MUT} stage IV tumors (**Fig. 1A, Supplementary Fig. 2D**). These results indicated tumor-specific CTLs from KRAS^{MUT} stage IV tumors were more sensitive to AICD. Based on the Reviewer's comments, these results were added to the revised manuscript, as follows:

(Manuscript, Results, Page 7, line 154-155)

“A similar phenomenon was observed in a different patient cohort with stage IV CRC (Fig. 1A, Supplementary Fig. 2D).”

- *The authors should first establish the lactate-NF-κB/AICD axis in their class IV colorectal tumors. This includes demonstrating the difference in NF-κB activity in CD8+ T cells from KRAS^{MUT} tumors compared to KRAS^{WT} tumors, showing that NF-κB is responsible for AICD in CD8+ T cells, proving that KRAS^{MUT} tumors secrete more lactic acid compared to KRAS^{WT} tumors, and demonstrating that lactic acid indeed increases sensitivity to activation-induced cell death.*

Reply 1.3: Good suggestion! Accordingly, the lactate/NF-κB/AICD axis was further examined in light of the Reviewer's guidance. These data were added to the revised manuscript (**Supplementary Fig. 3A-E**), as follows:

(Manuscript, Results, Page 8 line 197 - Page 9 205)

“Using a different patient cohort with stage IV CRC, a lactic acid production advantage was confirmed in KRAS^{MUT} tumors (Supplementary Fig. 3A). Ex vivo administration of lactic acid significantly increased the AICD sensitivity of tumor-specific CTLs from stage IV CRC (Supplementary Fig. 3B). Further investigation demonstrated CTLs from KRAS^{MUT} stage IV tumors had an obvious decrease in NF-κB activity (Supplementary Fig. 3C-D). Moreover, NF-κB inhibitors BAY or JSH-23 almost completely abrogated the ability of lactic acid to regulate AICD (Supplementary Fig. 3E). Together, these findings established the lactic acid/NF-κB/AICD axis in stage IV

CRC.”

Comment 2: *The authors proposed a new mechanism involving lactylation-circATXN7 to explain the inhibition of NF-κB by lactate, but the mechanism is biased and not clearly demonstrated. Specifically, the following issues need to be addressed to improve the paper's credibility:*

Reply 2: Many thanks to the Reviewer's constructive comments. Please find our point-by-point responses to the comments as follows.

- *2.1 The authors focused only on MCT1 and intracellular lactate while ignoring the synergistic effect of lactate with GPR81.*

Reply 2.1: We thank the Reviewer's good comments. We completely agree with the Reviewer that it is important to evaluate the synergistic effect of lactic acid with MCT1 and GPR81. Current researches have indicated that lactic acid can inactivate NF-κB via MCT1-mediated input, or activating the lactate receptor GPR81. As displayed in **Supplementary Fig. 3F-G**, GPR81 inhibition by 3-OBA is significant and its combo inhibition of MCT1 with AZD3965 is obviously better than AZD3965 alone. These findings seemed to suggest a synergistic effect of lactic acid with MCT1 and GPR81 on NF-κB/AICD axis. However, this does not necessarily indicate both MCT1 and GPR81 contributed to the difference in NF-κB/AICD axis between KRAS^{MUT} versus KRAS^{WT} tumors. To distinguish their contribution, we first tested the expression levels of MCT1 and GPR81. Results found that MCT1 had significantly higher expression abundance than GPR81 in tumor-specific CTLs from both KRAS^{MUT} and KRAS^{WT} tumors (**Supplementary Fig. 3H**). Furthermore, the expression levels of MCT1, but not GPR81, was positively associated with the NF-κB activity (**Supplementary Fig. 3I-J**), and the NF-κB activity correlated well with intracellular lactic acid concentration in tumor-specific CTLs of KRAS^{MUT} tumors (**Supplementary Fig. 3K**), but not in those of KRAS^{WT} tumors (**Supplementary Fig. 3L-N**). More importantly, as the key downstream element in lactic acid/GPR81 axis, cAMP and TCF-1 in tumor-specific CTLs were well balanced between KRAS^{WT} versus KRAS^{MUT} tumors (**Supplementary Fig. 3O-P**). Taken together, these results suggested that MCT1-mediated lactic acid input, but not activating GPR81, contributed to the difference in NF-κB/AICD axis between KRAS^{WT} versus KRAS^{MUT} tumors. These results were incorporated in the revised manuscript, as follows:

(Manuscript, Results, Page 9, line 209-225)

“MCT1 blockade by AZD3965 significantly reversed the effects of lactic acid on NF-κB/AICD axis, and its combo inhibition of GPR81 with 3-OBA was obviously better than AZD3965 alone (**Supplementary Fig. 3F-G**). To distinguish their contribution to the difference in NF-κB/AICD axis between KRAS^{MUT} versus KRAS^{WT} tumors, the expression levels of MCT1 and GPR81 were further assessed. Results found that MCT1

had significantly higher expression abundance than GPR81 in tumor-specific CTLs from both KRAS^{MUT} and KRAS^{WT} tumors (**Supplementary Fig. 3H**). Moreover, the expression levels of MCT1, but not GPR81, was positively associated with NF-κB activity (**Supplementary Fig. 3I-J**), and NF-κB activity correlated well with intracellular lactic acid concentration in tumor-specific CTLs of KRAS^{MUT} tumors (**Supplementary Fig. 3K**), but not in those of KRAS^{WT} tumors (**Supplementary Fig. 3L-N**). More importantly, as the key downstream element in lactic acid/GPR81 axis, cAMP and TCF-1 in tumor-specific CTLs were well balanced between KRAS^{MUT} versus KRAS^{WT} tumors (**Supplementary Fig. 3O-P**). Taken together, these results suggested that MCT1-mediated lactic acid input, but not activating GPR81, contributed to the difference in NF-κB/AICD axis between KRAS^{MUT} versus KRAS^{WT} tumors.”

•2.2 *It is also subjective to focus only on circular RNA and ignore all other regulatory mechanisms in NF-KB pathway.*

Reply 2.2: Thank you very much for your comments. As a matter of fact, before exploring circRNAs involved in the differential NF-κB/AICD axis between KRAS^{MUT} versus KRAS^{WT} tumors, we had evaluated several NF-κB signaling-associated factors reported in literatures, including lncRNAs, microRNAs and proteins (**Supplementary Table 1**). To explore whether these factors contributed to the NF-κB/AICD axis, we first evaluated the mRNA expression of the above-mentioned factors in tumor-specific CD8⁺ T cells of KRAS^{MUT} versus KRAS^{WT} tumors. As determined through qRT-PCR, the expression levels of TSPAN15, DCLK1, TRINGS, ASB16-AS1 and miR-132 were increased in tumor-specific CD8⁺ T cells of KRAS^{MUT} versus KRAS^{WT} tumors (**Supplementary Fig. 4A**), whereas the expression levels of PP4R1, miR-26 and miR-155 were decreased (**Supplementary Fig. 4A**). We then investigated whether the 8 differently expressed factors could make a significant impact on NF-κB/AICD axis in CRC. Among the 5 upregulated factors in KRAS^{MUT} tumors, TSPAN15 and ASB16-AS1 silencing elicited a slight increase in NF-κB activity in tumor-specific CD8⁺ T cells from KRAS^{MUT} tumors (**Supplementary Fig. 4B-F**), but neither of them could make a significant impact on AICD sensitivity (**Supplementary Fig. 4J-K**). Among the 3 downregulated factors in KRAS^{MUT} tumors, only overexpression of miR-155 in tumor-specific CD8⁺ T cells from KRAS^{WT} tumors decreased NF-κB activity (**Supplementary Fig. 4G-I**), but paradoxically overexpression of miR-155 led to a slight decrease in AICD sensitivity (**Supplementary Fig. 4L**). Together, these findings suggested the differential NF-κB/AICD axis between KRAS^{WT} versus KRAS^{MUT} tumors is not governed by the above-mentioned factors, but is likely to be controlled by other factors.

Over the past decade, circRNAs have emerged as a large class of primarily non-coding RNA molecules. The interest in studying circRNAs is raised because of several peculiar features, such as evolutionary conservation and tissue-specific expression, but above all, because their deregulated expression was linked to many pathological conditions,

particularly cancer. Current data from in vitro as well as in vivo studies along with analysis of clinical cancer tissues suggest that these molecules are of potential clinical relevance and utility (Cell. 2022;185(12):2016-2034; Nat Rev Clin Oncol. 2022;19(3):188-206; Cell. 2022;185(10):1728-1744.e16.). In particular, circRNAs have also been identified to be participants in the regulatory networks of various anti-tumor immune responses. Wang and colleagues found overexpression of hsa_circ_0020397 in CRC cells could promote the upregulation of PD-L1 by binding and inhibiting miR-138 expression, thereby resulting in tumor immune escape (Cell Biol Int. 2017;41(9):1056-1064.). Furthermore, there is evidence of a correlation between circRNAs and the infiltration of immune cells in several cancers (Nat Commun. 2022;13(1):7243; J Med Genet. 2019;56(1):32-38; Biomolecules. 2019;9(9):429.). Recently, a study by Ye et al identified circRNA profiles and regulatory networks in advanced melanoma patients treated with immune checkpoint blockades, highlighting the clinical utility of circRNAs as predictive biomarkers of cancer immunotherapy (Nat Commun. 2023; 14: 2540.). However, the roles of circRNAs in tumor-infiltrating T cells related to cancer immunology are still poorly understood. These knowledge gaps need to be addressed to move this relatively young field of research forward and bring circRNAs to the forefront of clinical practice. In this study, we firstly characterized the contribution of lymphocyte-expressed circRNAs to NF- κ B pathway and its downstream biological function. This regulatory pattern advances the current understanding of their cellular roles of circRNAs, as well as the molecular basis of T-cell fate decision. Of clinical importance pointing toward a relevant therapeutic utility, targeting circATXN7 in CD8⁺ T cells could shift KRAS^{MUT} tumors from immunologically “cool” towards “hot”, thereby improving immunotherapeutic efficacy. Based on the Reviewer’s comments, the following sentences were added to the revised manuscript, as follows:

(Manuscript, Results, Page 9, line 227-232)

“To explore the mechanism underlying the difference in NF- κ B/AICD axis between KRAS^{MUT} versus KRAS^{WT} tumors, several NF- κ B signaling-related factors reported in literatures (**Supplementary Table 1**) were tested. Results found (**Supplementary Fig. 4A-L**) suggested the differential NF- κ B/AICD axis between KRAS^{MUT} versus KRAS^{WT} tumors was not governed by the above-mentioned factors, but is likely to be controlled by other factors.”

(Manuscript, Introduction, Page 5 line 110 - Page 6 line 128)

“Circular RNAs (circRNAs) emerge as a unique class of RNA molecules characterized by their covalently closed ring structure. The interest in studying circRNAs is raised because of several peculiar features, such as evolutionary conservation and tissue-specific expression, but above all, because their deregulated expression was linked to many pathological conditions, particularly cancers^{32, 33}. Mounting data suggest these molecules are of potential clinical relevance and utility^{34, 35}. Notably, circRNAs have been identified to be participants in the regulatory networks of tumor immunity³⁶. Wang and colleagues demonstrated overexpression of hsa_circ_0020397 in CRC cells could promote the upregulation of PD-L1 by binding and inhibiting miR-138 expression,

thereby resulting in tumor immune escape³⁷. Furthermore, there is evidence of a correlation between circRNAs and immune cell infiltration in several cancers^{38, 39, 40}. Recently, a study by Ye et al identified circRNA profiles and regulatory networks in melanoma patients treated with immune checkpoint blockades, highlighting the clinical application potential of circRNAs as predictive biomarkers for immunotherapeutic efficacy⁴¹. These advances underscored the link between circRNAs and cancer immunology, yet knowledge of the role played by circRNAs and the mechanism of circRNAs' action in CTLs is limited. Using circRNA sequencing and CD8-conditional circRNA knockout mice, this work set an example of how circRNAs regulate AICD of CTLs and subsequently influence immunotherapy.”

• *2.3 The authors' selection of circATXN7 from among the 130 up-regulated circular RNAs by lactate is subjective and lacks sufficient rationale.*

Reply 2.3: We thank the Reviewer for pointing out this issue. To study whether circRNAs contribute to the lactic acid-regulated AICD sensitivity in a manner related to NF- κ B activity, we performed circRNA profile analysis in AICD-resistant T cells after lactic acid or vehicle treatment. After filtering differentially expressed circRNAs (FC > 2 or < 0.5 and FDR < 0.05), the top 10 up-regulated as well as the top 10 down-regulated circRNAs were selected for further investigation in the present study. As determined by qRT-PCR in AICD-sensitive versus AICD-resistant T cells, 5 of the top 10 down-regulated circRNAs exhibited a consistent expression trend (**Supplementary Fig. 6A**). Through RIP against p65, 3 of the 5 circRNAs were identified as p65-bound circRNAs (**Supplementary Fig. 6B**). However, subsequent function assays showed none of them had the ability to regulate AICD (**Supplementary Fig. 6C-E**). Previous studies (Nature. 2019;574(7779):575-580; Genome Biol. 2021;22(1):85.) as well as ours (Int J Biol Sci . 2022;18(8):3470-3483.) have indicated lactic acid could promote gene expression by histone lactylation. Considering the contribution of lactic acid to transcriptional activation, our initial manuscript only showed the results regarding the selection of circATXN7 from among the up-regulated circRNAs. We apologize for not showing the results regarding the exclusion of the down-regulated circRNAs, and we have added this information in the revised manuscript.

To strengthen the clinical relevance of circRNAs by lactic acid, the expression level of circATXN7 was further tested in CD8⁺ T cells derived from KRAS^{MUT} versus KRAS^{WT} CRC tissues (**Supplementary Fig. 5J-K**). In light of the Reviewer's comments, these results were incorporated in the revised manuscript.

(Manuscript, Results, Page 10 line 256 - Page 11 line 264)

“Clinically, a significant increased expression level of circATXN7 in tumor-specific CTLs derived from KRAS^{MUT} versus KRAS^{WT} CRC tissues (**Supplementary Fig. 5J**), whereas a comparable circATXN7 expression was found in tumor non-specific CTLs from KRAS^{MUT} versus KRAS^{WT} CRC tissues (**Supplementary Fig. 5K**). In addition,

none of the 10 down-regulated circRNAs were identified as eligible candidates for further studies through the screening schematic diagram (Supplementary Fig. 6A-E).”

• 2.4 The specific regulation of lactate and lactylation on circATXN7 in CD8+ T cells and not in other cell types in TME is not clear.

Reply 2.4: Many thanks to the Reviewer’s constructive comments. According to the Reviewer’s comments, we explored the regulation of lactate-lactylation on circATXN7 in other cell types in TME. To this end, the key cell types in TME were sorted, including CD4+ T cells, macrophages, endothelial cells, fibroblasts, and NK cells. Results from western blots found all of the cell types had histone H3K18la (See Figure 1A# below). However, ChIP-qPCR identified none of them had H3K18la enrichment in ATXN7 promoter regions (See Figure 1B# below). These findings were in line with our conclusion that circATXN7 is mainly expressed in tumor-specific CTLs (Fig. 3A-B, Supplementary Fig. 7C-H). We agree with the Reviewer that it is interesting to evaluate the roles of lactate-lactylation in other cell types in TME. This is an important scientific question which is more appropriate to be answered in future studies. Based on the Reviewer’s comments, the following sentences were added to the revised manuscript, as follows:

(Manuscript, Discussion, Page 22, line 614-616)

“However, it is worthy of further efforts to understand the roles of lactate-lactylation in other cell types in tumor microenvironment.”

Figure 1#. (A) Western blots showing H3K18la levels in the indicated cells isolated from CRC tissues. (B) ChIP-qPCR analysis for H3K18la status at the ATXN7 promoter of the indicated cells isolated from CRC tissues (n = 3). ns indicates $p > 0.05$, by two-tailed Student’s t-test (B).

• 2.5 The mechanism of binding between circATXN7 and p65 is not clear. The authors did not provide sufficient details on the construction of circATXN7 and p65 mutants. The exact sequence of their blocking oligos, the complementarity of the blocking oligos to circATXN7, and the specific sites mutated on circATXN7 and p65 were not provided. The authors also did not explain why the blocking oligos were unable to completely block the interaction between circATXN7 and p65. More information and experimental details are needed to fully understand the mechanism of action of circATXN7 in

inhibiting the NF- κ B pathway.

Reply 2.5: Thank you for noting this issue. In light of the Reviewer's comments, the details on the construction of circATXN7 and p65 mutants, the exact sequence of their blocking oligos, the complementarity of the blocking oligos to circATXN7, and the specific sites mutated on circATXN7 and p65 were added to the revised manuscript (**Manuscript, Figures and Legends, Page 43, line 1141-1142; Supplementary Material, Page 3, line 76 and Page 4, line 80**).

For the concern regarding why the blocking oligos were unable to completely block the interaction between circATXN7 and p65, additional experiments were performed to explain the confusion. As showed in the previous literatures (Nucleic Acids Res. 2019;47(7):3580-3593; Cell Death Differ. 2019;26(12):2758-2773; Mol Ther. 2020;28(5):1287-1298; Cell Death Differ.; 24(2): 357–370.), it is a common event that the blocking oligos did not completely block the interaction between circRNAs and its-bound proteins. This phenomenon is likely to correlate with the transfection dose of blocking oligos. To test this hypothesis, a dose response experiment was performed. Day 6 T cells were treated with a series of blocking oligo concentrations ranging from 2 nM to 16 μ M. Post-treated cells were then subjected to pull-down assays by biotin-labeled circATXN7 probes. Results found that blocking oligo transfection at dose lower than 50 nM was unable to block the interaction between circATXN7 and p65 (**See Figure 2# below**), and blocking oligo transfection at dose in the range of 100 nM to 1 μ M could significantly block the interaction between circATXN7 and p65 in a dose-dependent manner (**See Figure 2# below**), while the interaction between circATXN7 and p65 was completely blocked upon blocking oligo transfection at dose greater than 2 μ M (**See Figure 2# below**). According to these findings, we concluded that the partial blocking effect on the interaction between circATXN7 and p65 was due to the transfection with 800 nM blocking oligos.

Figure 2#. circRNA pull-down assays were conducted using biotin-labeled circATXN7 probes against cell lysates from Day 6 T cells transfected with the indicated dose of blocking oligos. Co-precipitated proteins were detected by immunoblots using anti-p65 antibodies.

Comment 3: *There is a lack of clarity on why targeting circATXN7 can specifically inhibit KRASMUT tumors while not KRASWT tumors. Since all mechanistic studies were carried out in WT CD8+ T cells, blocking circATXN7 should be able to restore nuclear NF-KB activity in all CD8+ T cells. Additionally, a considerable proportion of*

KRAS^{WT} tumors also secrete lactate, and a certain proportion of KRAS^{WT} tumors contain circATXN7⁺ T cells (93 out of 182). Therefore, the authors should provide further clarification on the specificity of the effect of targeting circATXN7 in KRAS^{MUT} tumors.

Reply 3: We appreciate the Reviewer for raising this important question. We fully agree with the Reviewer that it is important to clarify the specific effect of targeting circATXN7 in KRAS^{MUT} tumors. In this work, we demonstrated that tumor-specific CD8⁺ T cells from KRAS^{MUT} tumors exhibited a circATXN7^{high}/p65^{low} expression pattern, as compared with those from KRAS^{WT} tumors. The distinct expression patterns of circATXN7/p65 axis between KRAS^{MUT} versus KRAS^{WT} tumors might contribute to the specific effect of targeting circATXN7 in KRAS^{MUT} tumors. A further test of this model would be to assess the absolute number of the circATXN7 versus that of p65 molecules per cell. Absolute qRT-PCR in tumor-specific CD8⁺ T cell of KRAS^{MUT} tumors showed there were 926.8 ± 148.2 circATXN7 molecules per cell versus 1063.6 ± 112.6 p65 molecules per cell (**Supplementary Fig. 9G**). However, circATXN7 was 80.8 ± 26.0 copies per tumor-specific CD8⁺ T cell of KRAS^{WT} tumors (**Supplementary Fig. 9G**), which was significantly lower than p65 (1018.8 ± 105.9 copies per cell). According to the stoichiometry of the circATXN7 versus that of p65, we concluded that tumor-specific CD8⁺ T cells of KRAS^{MUT} tumors, but not those of KRAS^{WT} tumors, have sufficient circATXN7 abundance to directly bind p65 for inhibition.

The Reviewer is right that WT CD8⁺ T cells were used for the mechanistic studies, including circRNA RNA pull-down and RIP experiments, in which the expression of circATXN7 expression was enforced to make circATXN7 abundance enough to directly bind p65 for inhibition. Hence, enforcing circATXN7 expression in WT CD8⁺ T cells could result in a significant increase in circATXN7 expression (**Supplementary Fig. 9E**), and thereby elicit an effect on NF- κ B/AICD axis (**Fig. 4A, Supplementary Fig. 9H**). However, due to the low amounts of circATXN7, silencing circATXN7 expression in WT CD8⁺ T cells was unable to restore nuclear NF- κ B p65 activity (**See Figure 3A# below**) as well as to affect AICD sensitivity (**Supplementary Fig. 9C**). This also reflects the specific effect of targeting circATXN7, which resulted from the distinct expression patterns of circATXN7/p65 axis between KRAS^{MUT} and KRAS^{WT} tumors.

It's true that a proportion of KRAS^{WT} tumors also secrete lactate. This work showed that MCT1-mediated lactate input contributed to the difference in NF- κ B/AICD axis between KRAS^{MUT} versus KRAS^{WT} tumors. MCT1 played a crucial role in this model. However, as a passive transporter, MCT1 can operate bidirectionally and has also been reported to facilitate lactic acid export from cells (Proc Natl Acad Sci U S A 2011;108:16663–8; Clin Cancer Res 2014;20:926–37.). The directionality of MCT1-driven lactic acid transport depends on a gradient between cytoplasmic and extracellular lactic acid concentrations (Cancer Res. 2017;77(20):5591-5601; Blood. 2007;109(9):3812-9.). Previous studies (Cell. 2012;149(3):656-70; EMBO Rep. 2019;20(6):e47451.) as well as ours (Signal Transduct Target Ther. 2021;6(1):144.) have

identified the lactate production from KRAS^{WT} tumors was significantly lower as compared with KRAS^{MUT} tumors. Furthermore, intracellular lactate concentration in tumor-specific CD8⁺ T cells of KRAS^{MUT} tumors was markedly higher than that of KRAS^{WT} tumors (See **Figure 3B# below**). In addition, tumor-specific CD8⁺ T cells from KRAS^{MUT} tumors exhibited high histone lactylation levels as compared with that in KRAS^{WT} tumors (**Fig. 2E-F**). Taken together, we concluded that the relatively low lactate concentration in KRAS^{WT} tumors might be unable to trigger the NF-κB/AICD axis.

As a qualitative test, RNA in situ hybridization (ISH) was used to demonstrate that a certain proportion of KRAS^{WT} tumors contain circATXN7⁺ T cells. However, absolute quantification analysis showed the stoichiometry of the circATXN7 in tumor-specific CD8⁺ T cells of KRAS^{WT} tumors had significantly lower circATXN7 abundance, as compared with that of KRAS^{MUT} tumors (See **Figure 3C# below**). These data, collectively, advanced our understanding of the specific effect of targeting circATXN7 in KRAS^{MUT} tumors.

In light of the Reviewer’s comments, we have added the corresponding statement to the revised manuscript, as follows:

(Manuscript, Results, Page 15, line 379-387)

“Absolute quantification in tumor-specific CTLs of KRAS^{MUT} tumors showed circATXN7 molecules per cell (926.8 ± 148.2) were similar to p65 molecules per cell (1063.6 ± 112.6), which would allow for an approximately equimolar interaction (**Supplementary Fig. 9G**). However, circATXN7 was 80.8 ± 26.0 copies per tumor-specific CTL of KRAS^{WT} tumors (**Supplementary Fig. 9G**), which was significantly lower than p65 (1018.8 ± 105.9 copies per cell). On the basis of the stoichiometry of the circATXN7 versus that of p65, tumor-specific CTLs of KRAS^{MUT} tumors, but not those of KRAS^{WT} tumors, had sufficient circATXN7 abundance to directly bind p65 for inhibition.”

Figure 3#. (A) NF-κB activity in KRAS^{WT} CRCs-derived tumor-specific CTLs transduced with lentivirus carrying an expression cassette for the shRNAs targeting circATXN7 (sh1 or sh2) or shRNA control vector (shVec) (n = 3). Ut, KRAS^{WT} CRCs-derived tumor-specific CTLs without any treatment. (B) Intracellular lactic acid levels in tumor-specific CTLs of KRAS^{WT} and KRAS^{MUT} tumors. (C) Total RNA was extracted from tumor-specific CTLs of KRAS^{WT} and KRAS^{MUT} tumors, and then

absolute quantitation of circATXN7 copy number by qRT-PCR is shown. $**p \leq 0.01$, $***p \leq 0.001$, and ns indicates $p > 0.05$, by one-way ANOVA (A), or two-tailed Student's t-test (B-C).

=====
Responses to comments (Reviewer #2)

In this manuscript, Zhou et al. identified that circATXN7, a NF- κ B-interacting circRNA in CD8⁺ T cells, was upregulated in KRAS^{MUT} CRC tissues and participated in regulation of T cells sensitivity to AICD. Mechanistically, circATXN7 binds to p65 subunit and isolates it in the cytoplasm thereby inactivating NF- κ B. The work presented here is interesting because it provides a new mode of action of circular RNA in the context of CD8⁺ T cells fate decision and anti-tumor immunotherapy. While the amount of data presented are impressive, a number of issues need to be resolved. Please see specific comments below.

Reply: We really appreciate the Reviewer for the constructive comments on our work. We have provided point-by-point responses to your comments.

Comment 1: *Whether the expression level of circATXN7 is different in tumor infiltrated CD8⁺ T cells derived from KRAS^{MUT} and KRAS^{WT} CRC tissues?*

Reply 1: Thank you. In light of the Reviewer's comments, we analyzed the expression levels of circATXN7 in tumor infiltrated CD8⁺ T cells derived from KRAS^{MUT} and KRAS^{WT} CRC tissues. Results demonstrated a significant increased expression level of circATXN7 in tumor-specific CD8⁺ T cells derived from KRAS^{MUT} versus KRAS^{WT} CRC tissues (**Supplementary Fig. 5J**), whereas a comparable circATXN7 expression was found in tumor non-specific CD8⁺ T cells from KRAS^{MUT} versus KRAS^{WT} CRC tissues (**Supplementary Fig. 5K**). Based on the Reviewer's comments, these results were added to the revised manuscript, as follows:

(Manuscript, Results, Page 10 line 258 - Page 11 line 262)

“Clinically, a significant increased expression level of circATXN7 in tumor-specific CTLs derived from KRAS^{MUT} versus KRAS^{WT} CRC tissues (Supplementary Fig. 5J**), whereas a comparable circATXN7 expression was found in tumor non-specific CTLs from KRAS^{MUT} versus KRAS^{WT} CRC tissues (**Supplementary Fig. 5K**).”**

Comment 2: *The authors have shown that circATXN7 can directly bind with p65 in CD8⁺ T cells and sequester it in the cytoplasm. p65 is known as a very abundant protein in CD8⁺ T cells, but circRNAs usually with a low abundance in cells. How many copies of circATXN7 and p65 are expressed in CRC patients and C57BL/6 mice CD8⁺ T cells? Can the stoichiometry of these molecules support the proposed model?*

Reply 2: We appreciate the Reviewer’s comments. Current researches have indicated circRNAs often exhibit tissue-restricted and cell-type specific expression patterns (Nat Rev Clin Oncol. 2022;19(3):188-206.). Although the majority of circRNAs are expressed at low levels in most tissues, individual circRNAs can accumulate to high levels in various cell types (PLoS Genet. 2013;9(9):e1003777; Nat Commun. 2022;13(1):4711; J Mol Med. 2017;95(11):1179-1189.). As reported, circRNA-protein interaction appears to be a common event (Proc Natl Acad Sci U S A. 2023;120(13):e2215132120; Nat Commun. 2022;13(1):7243; Cell. 2020;183(1):76-93.e22.). As suggested by the Reviewer, we further assessed the absolute number of the circATXN7 versus that of p65 molecules per cell to assess. Absolute qRT-PCR in tumor-specific CD8⁺ T cell of KRAS^{MUT} tumors showed there were 926.8 ± 148.2 circATXN7 molecules per cell versus 1063.6 ± 112.6 p65 molecules per cell (**Supplementary Fig. 9G**). However, circATXN7 was 80.8 ± 26.0 copies per tumor-specific CD8⁺ T cell of KRAS^{WT} tumors (**Supplementary Fig. 9G**), which was significantly lower than p65 (1018.8 ± 105.9 copies per cell). A similar pattern was seen in tumor infiltrated CD8⁺ T cells derived from tumor xenograft models in C57BL/6 mice (**See Figure 4# below**), which would allow for an approximately equimolar interaction. According to the stoichiometry of the circATXN7 versus that of p65, we concluded that tumor-specific CD8⁺ T cells of KRAS^{MUT} tumors, but not those of KRAS^{WT} tumors, have sufficient circATXN7 abundance to directly bind p65 for inhibition. These results were incorporated in the revised manuscript, as follows:

(Manuscript, Results, Page 15, line 379-387)

“Absolute quantification in tumor-specific CTLs of KRAS^{MUT} tumors showed circATXN7 molecules per cell (926.8 ± 148.2) were similar to p65 molecules per cell (1063.6 ± 112.6), which would allow for an approximately equimolar interaction (**Supplementary Fig. 9G**). However, circATXN7 was 80.8 ± 26.0 copies per tumor-specific CTL of KRAS^{WT} tumors (**Supplementary Fig. 9G**), which was significantly lower than p65 (1018.8 ± 105.9 copies per cell). On the basis of the stoichiometry of the circATXN7 versus that of p65, tumor-specific CTLs of KRAS^{MUT} tumors, but not those of KRAS^{WT} tumors, had sufficient circATXN7 abundance to directly bind p65 for inhibition.”

Figure 4#. Total RNA was extracted from tumor infiltrated CD8⁺ T cells derived from tumor xenograft models in C57BL/6 mice, and absolute quantitation of circATXN7 and p65 copy number by qRT-PCR is shown. ** $p \leq 0.01$, and ns indicates $p > 0.05$, by two-tailed Student’s t-test.

Comment 3: *The authors explored the roles of circATXN7 and circAtxn7 in CD8+ T cells derived from CRC patients and C57BL/6 mice, respectively. Is the sequence of circATXN7 conserved in mouse or different between human and mouse?*

Reply 3: Thank you. NCBI BLAST (<https://blast.ncbi.nlm.nih.gov/Blast.cgi>) was used to depict circATXN7 and its analogs in human and mouse, and results showed circATXN7 was highly conserved between human and mouse (85%). In light of the Reviewer's comments, these results were incorporated in the revised manuscript, as follows:

(Manuscript, Results, Page 11, line 266-268)

“CircATXN7 was formed by the back-splicing of two exons (exon 2 and exon 3) of the ATXN7 gene (chr3: 63898263-638989011) with 405 nt, and highly conserved between human and mouse (85%).”

Comment 4: *Figures 3A, B, G and Supplementary Figures 3 B, I. How did the authors design the probes used in ISH and FISH experiments to show the localization and level of circATXN7? Of note, as the shared primary sequences between circATXN7 and its cognate mRNA, it's difficult to specifically target circATXN7 without off-target on its cognate mRNAs. How do they exclude the ISH and FISH signals are not circATXN7 cognate mRNAs, in particularly, they both are localized in the cytoplasm?*

Reply 4: Thank you. For detecting circRNAs, we designed a specific probe to detect the circular rather than known linear transcript of ATXN7 as described previously (Cell. 2020;183(1):76-93.e22.). The probe is able to target the backsplice junction of the circRNAs to detect them. Importantly, the probe was specifically designed to hybridize only to the sequence of the backsplice junction, while leaving the linear transcripts unaffected (**Supplementary Fig. 7A**). We completely agree with the Reviewer that it is important to confirm that the probes could detect the circular rather than its linear transcript. To this end, we performed a pull-down using the probe. Afterwards, qRT-PCR was used to test the enrichment for circATXN7 and ATXN7 mRNA. Results found circATXN7 was specifically enriched with the probe, if compared with its linear counterpart or other non-specific RNAs, such as GAPDH mRNA (**Supplementary Fig. 7B**). These findings further confirmed that the probe used in this study could specifically target circATXN7 without off-target on its cognate mRNAs. To make it more clearly, the following texts were rephrased in the revised manuscript, as follows:

(Manuscript, Results, Page 12, line 302-306)

“Next, we sought to evaluate the clinical significance of circATXN7 expression in CRC patients. Using a specific probe to detect the circular rather than known linear transcript of ATXN7 (**Supplementary Fig. 7A-B**), RNA in situ hybridization (ISH) assays for circATXN7 expression were performed in paraffin-embedded CRC sections from 269

CRC patients from the SYSU-6thAH cohort.”

Comment 5: *In consideration of the shared primary sequences between circATXN7 and ATXN7 mRNA, the authors should detect whether shcircATXN7 has effect on the ATXN7 mRNA level.*

Reply 5: We thank the Reviewer’s good advice. For silencing circRNAs, we designed lentiviral vectors expressing shRNAs which target the backsplice junction of the circRNAs to deplete them. Importantly, these shRNAs were specifically designed to hybridize only to the sequence of the backsplice junction, while leaving the linear transcripts unaffected (**Supplementary Fig. 5E**). The specific and efficient knockdown of the circRNAs was confirmed by qRT-PCR. As anticipated, shcircATXN7 had no significant effects on the ATXN7 mRNA level (**Supplementary Fig. 5E**). These results were incorporated in the revised manuscript, as follows:

(Manuscript, Results, Page 10, line 250-253)

“Using lentiviral vectors expressing shRNAs that target the backsplice junction of the circRNAs and deplete the circular rather than their linear transcripts (**Supplementary Fig. 5E**), functional assays demonstrated that only circATXN7 had the ability to regulate AICD (**Fig. 2C, Supplementary Fig. 5F**).”

Comment 6: *The authors should confirm the KO efficiency of circAtxn7 in the circAtxn7 CKO mice and show this data in the manuscript.*

Reply 6: Good suggestion! As suggested by the Reviewer, the KO efficiency of circAtxn7 in the circAtxn7 CKO mice was added to the revised manuscript, as follows:

(Manuscript, Results, Page 16, line 431-433)

“As expected, these mice lacked the circular form *circAtxn7* while the levels of the linear host gene *Atxn7* mRNA and ATXN7 protein were unaltered (**Supplementary Fig. 10B-C**)”

Comment 7: *Supplementary Fig. 2: both negative and positive controls should be added in RIP assays to show the specificity.*

Reply 7: We thank the Reviewer for the comments. In light of the Reviewer’s advice, the negative and positive controls were added in RIP assays (**Supplementary Fig. 5D**).

Comment 8: *It is recommended to add a working model in the manuscript for the functions and mechanism of circATXN7 on tumor progression.*

Reply 8: Many thanks to the Reviewer's comments. Based on the Reviewer's advice, a working model was added to the revised manuscript (**Fig. 7**).

=====

Responses to comments (Reviewer #3)

This study by Zhou et al. identified a novel circular RNA circATXN7 that plays an important role in modulating the antitumor activity of tumor-infiltrated CTLs in Kras-mutant colorectal cancer. The in vitro and in vivo studies of circATXN7 are overall sound. Particularly, the generation of circATXN7 KO mice provided a great animal model to reveal the function of this new circATXN7 in the tumor-associated CTLs. However, the manuscript appears to be disconnected due to lack of supporting evidence that links AICD, circATXN7 and lactic acid in the tumor microenvironment. Below are my major concerns:

Reply: Thank you. We truly appreciate the reviewer for your comments. We have provided point-by-point responses to your comments.

Comment 1: *The previous work by the group and others have identified the role of lactic acid in ACID of CTLs. The Fig 1 and Fig 2A-D seem to further support the previous studies, but not directly connect to the circATXN7, which therefore could be moved to supplemental information.*

Reply 1: We thank the Reviewer's comments. As suggested by the Reviewer, the **Fig 2A-D** in our initial manuscript were moved to supplemental information in the revised manuscript. The Reviewer is right that our previous study has partially identified the role of lactic acid in ACID of CTLs (Adv Sci. 2023;10(6):e2203757.). Based on these findings, the **Fig. 1** provided the following advances.

1) Our previous study demonstrated an inverse link of intratumoral cytotoxic CD8⁺ T-cells with mutant KRAS in the context of CRC (Adv Sci. 2023;10(6):e2203757.). However, due to the limited sample size of clinical cohort, our previous report failed to evaluate the potential difference in the prognostic value of the CTL tumor infiltrate between KRAS^{MUT} versus KRAS^{WT} tumors. The current work distinguished the difference in the prognostic value of the CTL tumor infiltrate between KRAS^{MUT} versus KRAS^{WT} tumors. In light of the Reviewer's comments, these results were moved to supplemental information (**Supplementary Fig. 1B-E, Supplementary Fig. 2A-C**).

2) ACID is significantly increased also in T cells exposed to CEA antigen. Yet it is unclear whether the increase in ACID is a CEA-specific mechanism. To address this, the link between AICD sensitivity and CEA expression was further analyzed, and results demonstrated that the AICD sensitivity had no significant correlation with CEA expression levels (**Fig. 1G**). These findings suggested that the increase in AICD in

KRAS^{MUT} tumors might be independent of CEA expression. To further confirm this, tumor-specific CD8⁺ T cells were purified from CEA positive and negative expressing tumors using anti-MUC1 tetramer as described previously (J. Cell. Biochem. 2019;120:8815–8828). After coculturing with autologous tumor cells, comparable apoptosis was found in the tumor-specific CD8⁺ T cells from CEA positive versus negative expressing tumors (**Fig. 1H-I**). Taken together, these results indicated that the increase in AICD in KRAS^{MUT} tumors was independent of CEA expression. We incorporated these new data in the new **Fig 1G-I**.

3) Our previous study ascribed the decreased CTLs occurred in KRAS^{MUT} CRC to the increased susceptibility to tumor-mediated AICD of tumor-specific CTLs. Current researches indicated that AICD is influenced by the nature of the initial T-cell activation events, and TCR engagement with the MHC-tumor antigen complex played a vital role in the activation of tumor-specific CTLs. The contribution of TCR engagement with the MHC-tumor antigen complex to tumor-mediated AICD in KRAS^{MUT} CRC has thus attracted our interest. **Fig. 1B-F** in this study further demonstrated that autologous tumor cells elicited AICD in activated CTLs from KRAS^{MUT} tumors through repeated TCR stimulation. These results improved the understanding of how oncogenic KRAS affects the immune tumour microenvironment. We believe that our findings in the **Fig 1** would spur keen interests among a broad spectrum of readers, and we would like to leave the new **Fig 1** in the main text. We are also open to the Editors' suggestion for this issue.

Comment 2: *There is no evidence why circRNAs play an essential role in AICD of CTLs. No strong rationale was provided although the identified circATXN7 appears to have important functions.*

Reply 2: We appreciate the Reviewer's comments. As a matter of fact, before exploring circRNAs involved in the differential NF- κ B/AICD axis between KRAS^{MUT} versus KRAS^{WT} tumors, we had evaluated several NF- κ B signaling-associated factors reported in literatures, including lncRNAs, microRNAs and proteins (**Supplementary Table 1**). To explore whether these factors contributed to the NF- κ B/AICD axis, we first evaluated the mRNA expression of the above-mentioned factors in tumor-specific CD8⁺ T cells of KRAS^{MUT} versus KRAS^{WT} tumors. As determined through qRT-PCR, the expression levels of TSPAN15, DCLK1, TRINGS, ASB16-AS1 and miR-132 were increased in tumor-specific CD8⁺ T cells of KRAS^{MUT} versus KRAS^{WT} tumors (**Supplementary Fig. 4A**), whereas the expression levels of PP4R1, miR-26 and miR-155 were decreased (**Supplementary Fig. 4A**). We then investigated whether the 8 differently expressed factors could make a significant impact on NF- κ B/AICD axis in CRC. Among the 5 upregulated factors in KRAS^{MUT} tumors, TSPAN15 and ASB16-AS1 silencing elicited a slight increase in NF- κ B activity in tumor-specific CD8⁺ T cells from KRAS^{MUT} tumors (**Supplementary Fig. 4B-F**), but neither of them could make a significant impact on AICD sensitivity (**Supplementary Fig. 4J-K**). Among the 3

downregulated factors in KRAS^{MUT} tumors, only overexpression of miR-155 in tumor-specific CD8⁺ T cells from KRAS^{WT} tumors decreased NF-κB activity (**Supplementary Fig. 4G-I**), but paradoxically overexpression of miR-155 led to a slight decrease in AICD sensitivity (**Supplementary Fig. 4L**). Together, these findings suggested the differential NF-κB/AICD axis between KRAS^{WT} versus KRAS^{MUT} tumors is not governed by the above-mentioned factors, but is likely to be controlled by other factors.

Over the past decade, circRNAs have emerged as a large class of primarily non-coding RNA molecules. The interest in studying circRNAs is raised because of several peculiar features, such as evolutionary conservation and tissue-specific expression, but above all, because their deregulated expression was linked to many pathological conditions, particularly cancer. Current data from in vitro as well as in vivo studies along with analysis of clinical cancer tissues suggest that these molecules are of potential clinical relevance and utility (Cell. 2022;185(12):2016-2034; Nat Rev Clin Oncol. 2022;19(3):188-206; Cell. 2022;185(10):1728-1744.e16.). In particular, circRNAs have also been identified to be participants in the regulatory networks of various anti-tumor immune responses. Wang and colleagues found overexpression of hsa_circ_0020397 in CRC cells could promote the upregulation of PD-L1 by binding and inhibiting miR-138 expression, thereby resulting in tumor immune escape (Cell Biol Int. 2017;41(9):1056-1064.). Furthermore, there is evidence of a correlation between circRNAs and the infiltration of immune cells in several cancers (Nat Commun. 2022;13(1):7243; J Med Genet. 2019;56(1):32-38; Biomolecules. 2019;9(9):429.). Recently, a study by Ye et al identified circRNA profiles and regulatory networks in advanced melanoma patients treated with immune checkpoint blockades, highlighting the clinical utility of circRNAs as predictive biomarkers of cancer immunotherapy (Nat Commun. 2023; 14: 2540.). However, the roles of circRNAs in tumor-infiltrating T cells related to cancer immunology are still poorly understood. These knowledge gaps need to be addressed to move this relatively young field of research forward and bring circRNAs to the forefront of clinical practice. In this study, we firstly characterized the contribution of lymphocyte-expressed circRNAs to NF-κB pathway and its downstream biological function. This regulatory pattern advances the current understanding of their cellular roles of circRNAs, as well as the molecular basis of T-cell fate decision. Of clinical importance pointing toward a relevant therapeutic utility, targeting circATXN7 in CD8⁺ T cells could shift KRAS^{MUT} tumors from immunologically “cool” towards “hot”, thereby improving immunotherapeutic efficacy. Based on the Reviewer’s comments, the following sentences were added to the revised manuscript, as follows:

(Manuscript, Results, Page 9, line 227-232)

“To explore the mechanism underlying the difference in NF-κB/AICD axis between KRAS^{MUT} versus KRAS^{WT} tumors, several NF-κB signaling-related factors reported in literatures (**Supplementary Table 1**) were tested. Results found (**Supplementary Fig. 4A-L**) suggested the differential NF-κB/AICD axis between KRAS^{MUT} versus KRAS^{WT} tumors was not governed by the above-mentioned factors, but is likely to be

controlled by other factors.”

(Manuscript, Introduction, Page 5 line 110- Page 6 line 128)

“Circular RNAs (circRNAs) emerge as a unique class of RNA molecules characterized by their covalently closed ring structure. The interest in studying circRNAs is raised because of several peculiar features, such as evolutionary conservation and tissue-specific expression, but above all, because their deregulated expression was linked to many pathological conditions, particularly cancers^{32, 33}. Mounting data suggest these molecules are of potential clinical relevance and utility^{34, 35}. Notably, circRNAs have been identified to be participants in the regulatory networks of tumor immunity³⁶. Wang and colleagues demonstrated overexpression of hsa_circ_0020397 in CRC cells could promote the upregulation of PD-L1 by binding and inhibiting miR-138 expression, thereby resulting in tumor immune escape³⁷. Furthermore, there is evidence of a correlation between circRNAs and immune cell infiltration in several cancers^{38, 39, 40}. Recently, a study by Ye et al identified circRNA profiles and regulatory networks in melanoma patients treated with immune checkpoint blockades, highlighting the clinical application potential of circRNAs as predictive biomarkers for immunotherapeutic efficacy⁴¹. These advances underscored the link between circRNAs and cancer immunology, yet knowledge of the role played by circRNAs and the mechanism of circRNAs’ action in CTLs is limited. Using circRNA sequencing and CD8-conditional circRNA knockout mice, this work set an example of how circRNAs regulate AICD of CTLs and subsequently influence immunotherapy.

Comment 3: *Given the importance of circATXN7 in CTLs, it is crucial to know the copy numbers of this circRNA, ensuring that their amount is enough to directly bind p65 for inhibition at a close molar ratio.*

Reply 3: We thank the Reviewer’s good comments. As suggested by the Reviewer, we further assessed the absolute number of the circATXN7 versus that of p65 molecules per cell. Absolute qRT-PCR in tumor-specific CD8⁺ T cell of KRAS^{MUT} tumors showed there were 926.8 ± 148.2 circATXN7 molecules per cell versus 1063.6 ± 112.6 p65 molecules per cell (**Supplementary Fig. 9G**). However, circATXN7 was 80.8 ± 26.0 copies per tumor-specific CD8⁺ T cell of KRAS^{WT} tumors (**Supplementary Fig. 9G**), which was significantly lower than p65 (1018.8 ± 105.9 copies per cell). According to the stoichiometry of the circATXN7 versus that of p65, we concluded tumor-specific CD8⁺ T cells of KRAS^{MUT} tumors, but not those of KRAS^{WT} tumors, have sufficient circATXN7 abundance to directly bind p65 for inhibition. These results were incorporated in the revised manuscript, as follows:

(Manuscript, Results, Page 15, line 379-387)

“Absolute quantification in tumor-specific CTLs of KRAS^{MUT} tumors showed circATXN7 molecules per cell (926.8 ± 148.2) were similar to p65 molecules per cell (1063.6 ± 112.6), which would allow for an approximately equimolar interaction

(Supplementary Fig. 9G). However, circATXN7 was 80.8 ± 26.0 copies per tumor-specific CTL of KRAS^{WT} tumors (Supplementary Fig. 9G), which was significantly lower than p65 (1018.8 ± 105.9 copies per cell). On the basis of the stoichiometry of the circATXN7 versus that of p65, tumor-specific CTLs of KRAS^{MUT} tumors, but not those of KRAS^{WT} tumors, had sufficient circATXN7 abundance to directly bind p65 for inhibition.”

Comment 4: *While circATXN7 is only expressed and function in CTLs, all the studies in the manuscript do not involve autologous CTLs in the tumor. For example, Fig. 1c, it is not understandable why tumor-specific CTLs are not used. The expansion of tumor-specific CTLs are well established for this type of study. The same weakness also applies to the Fig. 2c study.*

Reply 4: We thank the Reviewer for the comments. Based on the Reviewer’s comments, the experiments in Fig. 1c and Fig. 2c were recapitulated in the tumor-specific CTLs. Similar findings were obtained (Fig. 1A, Supplementary Fig. 2D, Supplementary Fig. 5J-K) and we have incorporated these results in the revised manuscript, as follows:

(Manuscript, Results, Page 7, line 154-155)

“A similar phenomenon was observed in a different patient cohort with stage IV CRC (Fig. 1A, Supplementary Fig. 2D).”

(Manuscript, Results, Page 10 line 258 - Page 11 line 262)

“Clinically, a significant increased expression level of circATXN7 in tumor-specific CTLs derived from KRAS^{MUT} versus KRAS^{WT} CRC tissues (Supplementary Fig. 5J), whereas a comparable circATXN7 expression was found in tumor non-specific CTLs from KRAS^{MUT} versus KRAS^{WT} CRC tissues (Supplementary Fig. 5K).”

Comment 5: *FISH staining of circATXN7 is not accurate to compare the cell-specific expression among various cell types. It is recommended to conduct scRNA-seq of total tumor and stroma cells or QRT-PCR of each cell type isolated by cell sorting.*

Reply 5: We appreciate the Reviewer’s comments. According to the Reviewer’s advice, qRT-PCR of each cell type isolated by cell sorting was performed. Results demonstrated that circATXN7 was mainly expressed in tumor-specific CTLs (Supplementary Fig. 7G). These results were incorporated in the revised manuscript, as follows:

(Manuscript, Results, Page 12, line 311-316)

“By sorting each tumor infiltration cell type, RT-PCR (Supplementary Fig. 7F) as well as qRT-PCR (Supplementary Fig. 7G) analysis demonstrated only the whole tumor tissues and CD8 cells had circATXN7 expression, but other components including CD4, macrophages, endothelial cells, and fibroblasts displayed negligible expression of

circATXN7.”

Comment 6: *It is unclear if the expression and function of Atxn7 gene is affected in the circATXN7 KO mice.*

Reply 6: We appreciate the Reviewer’s suggestion. As compared with wild-type mice, the circAtxn7 KO mice lacked the circular form circAtxn7, while the levels of the linear host gene Atxn7 mRNA and Atxn7 protein were unaltered (**Supplementary Fig. 10B-C**). These results were incorporated in the revised manuscript, as follows:

(Manuscript, Results, Page 16, line 431-433)

“As expected, these mice lacked the circular form *circAtxn7* while the levels of the linear host gene *Atxn7* mRNA and ATXN7 protein were unaltered (**Supplementary Fig. 10B-C**)”

Comment 7: *The introduction part does not provide any useful information. It should include more previous studies in ACID, lactic acid in TME, and etc...*

Reply 7: We appreciate the Reviewer for raising this issue. Based on the Reviewer’s comments, the following sentences were rephrased in the “**Introduction**” section of the revised manuscript, as follows:

(Manuscript, Introduction, Page 4 line 75 - Page 5 line 108)

“Cancer pathologies are often orchestrated by various metabolites, and KRAS mutant tumors are especially exposed to dramatically increased levels of lactic acid^{9,10}. Cancer-generated lactic acid endows malignancies with an acidic TME, and also acts as a primary carbon fuel source and signaling molecule involved in oncogenic pathways^{11,12}. Current researches have also yielded evidence that lactic acid in the TME was an impediment towards providing an effective antitumor immunity¹³. In this respect, tumor-derived lactic acid was found to take effects on tumor-associated macrophages^{14,15}, regulatory T cells¹⁶, myeloid-derived suppressor cells¹⁷, natural killer cells¹⁸, or dendritic cells¹⁹. In particular, Kreutz and colleagues pointed to an impact of lactic acid on cytolytic T lymphocytes (CTLs)²⁰, which directly identify and destroy nascent tumor cells during cancer immunosurveillance. Our previous study significantly advanced our understanding for the involvement of lactic acid in CTL fate decisions and subsequent support for tumor progress⁹. These insights highlighted an intense engagement between lactic acid and CTLs, but the intracellular mechanism of lactic acid action in CTLs remains poorly defined.

Activation-induced cell death (AICD), firstly described in 1987, has been characterized as a mechanistic link with immunological homeostasis^{21, 22}. Under physiological conditions, AICD is able to eradicate activated T lymphocytes presumed to be no longer

required²³. Abnormality in AICD was discovered in diverse pathological situations, such as viral infection, inflammatory and autoimmune disorders^{24, 25, 26}. In the context of many cancer types, AICD deregulation was also frequently identified^{27, 28, 29}. Aberrant AICD of tumor-specific CTLs can be used by cancers to evade immune elimination²⁸, which accounts for the paradoxical fact that, although the patients mount a specific T-cell response against neoplasm, these CTLs fail to control the disease. It is now widely understood that AICD is of much value to decipher cancer pathologies as well as present prognostic insights, or even develop alternative treatments for cancer patients. Along this line, we previously found mutant KRAS-expressing CRC cells exploited tumor-derived lactic acid to sensitize tumor-specific CTLs to AICD, thereby fostering tumor immune escape and immunotherapy resistance⁹. Multiple molecular players, including mitochondrio-nuclear translocation of AIF³⁰, CD158 receptor³¹, or NKILA²⁸, were identified to participate in an abnormal sensitivity of tumor-specific CTLs to AICD. Despite this knowledge, how lactic acid reprograms AICD of tumor-specific CTLs warrants under further investigation.”

Comment 8: *In Fig.2B, the inhibition by 3-OBA is significant and its combo treatment with AZD3965 is obviously better than AZD3965 alone. It is not consistent with the text in the manuscript " Results found MCT1 blockade by AZD3965, but not 3-OBA...."*

Reply 8: Thank you for your comments. We are sorry for the confusion, and to make it more clearly, the following sentences were rephrased in the revised manuscript, as follows:

(Manuscript, Results, Page 9, line 209-225)

“MCT1 blockade by AZD3965 significantly reversed the effects of lactic acid on NF- κ B/AICD axis, and its combo inhibition of GPR81 with 3-OBA was obviously better than AZD3965 alone (**Supplementary Fig. 3F-G**). To distinguish their contribution to the difference in NF- κ B/AICD axis between KRAS^{MUT} versus KRAS^{WT} tumors, the expression levels of MCT1 and GPR81 were further assessed. Results found that MCT1 had significantly higher expression abundance than GPR81 in tumor-specific CTLs from both KRAS^{MUT} and KRAS^{WT} tumors (**Supplementary Fig. 3H**). Moreover, the expression levels of MCT1, but not GPR81, was positively associated with NF- κ B activity (**Supplementary Fig. 3I-J**), and NF- κ B activity correlated well with intracellular lactic acid concentration in tumor-specific CTLs of KRAS^{MUT} tumors (**Supplementary Fig. 3K**), but not in those of KRAS^{WT} tumors (**Supplementary Fig. 3L-N**). More importantly, as the key downstream element in lactic acid/GPR81 axis, cAMP and TCF-1 in tumor-specific CTLs were well balanced between KRAS^{MUT} versus KRAS^{WT} tumors (**Supplementary Fig. 3O-P**). Taken together, these results suggested that MCT1-mediated lactic acid input, but not activating GPR81, contributed to the difference in NF- κ B/AICD axis between KRAS^{MUT} versus KRAS^{WT} tumors.”

Comment 9: *MC38 is a MSS colon cancer model, while only MSI-high CRCs are treated by ICIs in clinic. This should be discussed.*

Reply 9: Many thanks for the constructive suggestion. As a “workhorse” for cancer immunology research (Cancer Discov. 2016;6(1):71-9; Nat Commun. 2018;9(1):32; Nat Cancer. 2020;1(7):681-691; J Exp Med. 2021;218(11):e20200792.), the MC38 has been identified as an MSI-H model (Nat Commun. 2018 ;9(1):32; Cancer Cell. 2019;35(4):559-572.e7; Gut. 2023;gutjnl-2022-328845.). According to the Reviewer’s comments, we added the corresponding statement to the revised manuscript, as follows:

(Manuscript, Results, Page 19, line 497-499)

“To address this, we first performed anti-PD1 treatment in *circAtxn7^{CKO}* and WT mice with MC38K subcutaneous tumors, an MSI-H model but resistant to immunotherapies⁹.”

(Manuscript, Discussion, Page 21, line 558-561)

“While immunotherapy exhibits antitumor activity in some patients with MSI-H CRC, approximately 85% among all CRC patients, the therapeutic benefit is largely restricted, highlighting an unmet need for the study of mechanisms and combination regimens with immunotherapies.”

=====
Responses to comments (Reviewer #4)

The authors presents a comprehensive study on the role of circATXN7 in tumor immunoevasion, decoding its role in promoting activation-induced cell death (ACID) in tumor specific CTLs. The study exploits well-designed experimental in vitro and in vivo approaches and holds the potential to have a substantial impact on the development of new immune-therapeutic strategies. However, there are some questions that remained unanswered and need to be addressed to make this study more impactful.

Reply: We really appreciate the reviewer for the constructive comments and helpful suggestions on our work. We have provided point-by-point responses to your comments.

Comment 1: *Figure 1D and E. Experiments are convincing but a graph showing biological replicates and deviation would make their conclusions more solid and also help the reader understanding the reproducibility across and within CRC patients. Also, CD8 isolated from healthy donors PBMCs should be included as an additional control to test whether CD8 from CRC patients are more sensitive to ACID in comparison to CD8 from cancer-free donors.*

Reply 1: Thank you. As suggested by the Reviewer, the corresponding graphs were added in the revised manuscript (**Supplementary Fig. 2E-F**). According to the reviewer’s comments, CD8⁺T cells were isolated from healthy donors’ PBMCs and

activated by autologous DCs pulsed with CEA peptide for 6 days. After AICD induction in the day-6 CEA-specific CTLs from healthy donors, we found that re-stimulation with anti-CD3 or CEA-loaded T2 cells could lead to a massive apoptosis (See **Figure 5A# below**), which was comparable to that of CRC patients (See **Figure 5B# below**). These findings indicated that CD8 from CRC patients' PBMCs are not more sensitive to ACID in comparison to CD8 from cancer-free donors, and the success in AICD induction was not limited to T cells from CRC patients as a similar phenomenon can be observed in T cells from cancer-free donors.

Figure 5#. (A) Healthy donors-derived peripheral CD8⁺ T cells were activated by autologous DCs pulsed with CEA peptide for the indicated number of days. Apoptosis for the indicated CTLs induced by anti-CD3 or CEA-loaded T2 cells (T2/CEA) (n = 3). Numerical values denote annexin V⁺ cell percentages (mean ± SD). (B) Statistics of anti-CD3- or CEA-loaded T2 cells-induced apoptosis of CTLs from CRC patients versus healthy donors (n = 3). ns indicates $p > 0.05$, by two-tailed Student's t-test.

Comment 2: At Line 117-118, the authors concluded “autologous tumor cells elicited AICD in activated CTLs from KRASMUT tumors through repeated TCR stimulation”. To make this statement I believe the authors should compare side by side the CD8 ACID sensibility upon exposure to tumor coming from KRASMUT and KRASWT CRC tumor. Instead, unless the Reviewer has overlooked (if so please add a more careful description of the exp detailing the source of the CD8, type of tumor ..), it looks like that the exp has been done with CRC tumors that have not previously classified in KRASMUT and KRASWT. In this case, it would be advisable to showed results from CD8 exposed to KRASMUT versus KRASWT and/or correlate the ACID sensitivity to the KRAS mutation.

Reply 2: We truly appreciate the Reviewer's comments, and we apologize for not presenting clear information. As suggested by the Reviewer, we compared the AICD sensitivity of CTLs from KRAS^{MUT} versus KRAS^{WT} tumors. After coculturing with autologous tumor cells, we found significantly increased apoptosis in tumor-specific CTLs from KRAS^{MUT} versus KRAS^{WT} tumors (**Fig. 1A, Supplementary Fig. 2D**). In line with this, treatment with anti-CD3 antibodies elicited massive apoptosis of tumor-specific CTLs from KRAS^{MUT} tumors (**Fig. 1A, Supplementary Fig. 2D**). These results indicated that tumor-specific CTLs from KRAS^{MUT} tumors were more sensitive to AICD. Accordingly, these results were added to the revised manuscript, as follows:

(Manuscript, Results, Page 7, line 152-155)

“Our previous findings ascribed the decreased CTLs occurred in KRAS^{MUT} stage I-III CRC to the increased susceptibility to tumor-mediated AICD of tumor-specific CTLs¹¹. A similar phenomenon was observed in a different patient cohort with stage IV CRC (**Fig. 1A, Supplementary Fig. 2D**).”

Comment 3: *ACID is significantly increased also in CD8 exposed to CEA antigen. Is then ACID increase a common mechanism happening in CD8 exposed to several type of tumors or for instance to those that overexpressed CEA (i.e. breast, lung ovarian)? I think this is an important point that needs to be clarified in order to be able to state that there is an association between mutant KRAS and CD8 ACID sensitivity within the tumor. Also, is the increase in ACID a tumor-specific mechanism or also viral antigen give rise to the same phenomenon?*

Reply 3: Thank you very much for raising this interesting and important point. In light of the Reviewer's comments, the link between AICD sensitivity and CEA expression was firstly analyzed, and results demonstrated the AICD sensitivity had no significant correlation with CEA expression levels (**Fig. 1G**). These findings suggested that the increase in AICD in KRAS^{MUT} tumors might be independent of CEA expression. To further confirm this, tumor-specific CD8⁺ T cells were purified from CEA positive and negative expressing tumors using anti-MUC1 tetramer as described previously (J. Cell. Biochem. 2019;120:8815–8828). After coculturing with autologous tumor cells, comparable apoptosis was found in the tumor-specific CD8⁺ T cells from CEA positive versus negative expressing tumors (**Fig. 1H-I**). Taken together, these results indicated CEA expression had no significant effects on the association between KRAS^{MUT} and AICD sensitivity.

AICD merges as a negative regulator of activated T cells upon activation through the T-cell receptor, and is influenced by the nature of the initial T-cell activation events (Immunol Rev. 2003;193:70-81.). Following an initial stimulation triggered by an antigen, AICD typically ensues as a result of a secondary activation (re-stimulation) of the TCR that is brought about by the persistence of the antigen (Theranostics. 2020; 10(10): 4481–4489; Immunological Reviews. 2003;193:70–81.). The current study demonstrated an increase in AICD

in tumor-specific CTLs of KRAS^{MUT} tumors. The increase in ACID has also been noted in various conditions. Our previous report found Treg cells had an increase in ACID in Crohn's disease (J Crohns Colitis. 2020;14(11):1619-1631.). A study by Tan et al suggested Th1 cells were susceptible to AICD in the context of mouse eye inflammation (Cell Immunol 2011;271:210-3.). These findings indicated that the increase in ACID seemed not to be a tumor-specific mechanism, but appeared to be disease context-dependent.

We agree with the Reviewer that it is interesting to evaluate the association between viral antigen and AICD sensitivity. Current researches indicated virus infections could induce apoptosis in T cells by AICD (Cell Death Differ. 2002;9(6):651-60; Apoptosis. 2000;5(5):431-4; J Immunol. 1995;154(11):6013-21.). The increase in AICD of activated CD8⁺ T cells generated during a viral infection could maintain homeostasis of the immune system, so that during the resolution phase of infection, excess activated T cells are deleted (Immunobiology. 2016;221(3):432-9; Curr Opin Microbiol. 1999;2(4):382-7; Adv Virus Res. 1995;45:1-60.). In addition, in response to HIV antigen, McDyer et al found CD4⁺ T cells underwent an AICD increase (Am J Respir Crit Care Med. 2014; 190(7): 744-755.), which represents a critical factor in the loss of CD4⁺ T lymphocytes caused by HIV-infection (J Acquir Immune Defic Syndr. 2021;86(1):128-137.). These insights suggest the ACID increase related to viral antigen occurs frequently. However, the functional contribution of circRNAs to viral antigen-related AICD increase remains unclear, and we believe this is an important scientific question which is more appropriate to be answered in future studies.

Based on the Reviewer's comments, we have added the corresponding statement to the revised manuscript, as follows:

(Manuscript, Results, Page 7, line 171- Page 8, line 182)

“The above findings showed that ACID was significantly increased in T cells exposed to CEA antigen. Yet it is unclear whether the increase in ACID is a CEA-specific mechanism. To address this, the link between AICD sensitivity and CEA expression was further analyzed, and results demonstrated the AICD sensitivity had no significant correlation with CEA expression levels (**Fig. 1G**). These findings suggested that the increase in AICD in KRAS^{MUT} tumors might be independent of CEA expression. To further confirm this, tumor-specific CTLs were purified from CEA positive and negative expressing tumors using anti-MUC1 tetramer as described previously (J. Cell. Biochem. 2019;120:8815-8828). After coculturing with autologous tumor cells, comparable apoptosis was found in the tumor-specific CTLs from CEA positive versus negative expressing tumors (**Fig. 1H-I**). Together, these results indicated the increase in AICD in KRAS^{MUT} tumors was independent of CEA expression.”

(Manuscript, Discussion, Page 21 line 577 - Page 22 line 585)

“The increase in ACID has also been noted in various conditions. A study by Tan et al suggested Th1 cells were susceptible to AICD in the context of mouse eye inflammation⁵⁰. Also, virus infections could induce apoptosis in T cells by AICD⁵¹⁻⁵².⁵³. The increase in AICD of activated CD8⁺ T cells generated during a viral infection

serves to maintain homeostasis of the immune system, so that during the resolution phase of infection, excess activated T cells are deleted^{54, 55, 56}. These findings indicated that the increase in ACID seemed not to be a tumor-specific mechanism, but appeared to be disease context-dependent.”

Comment 4: *Beside measurement of ACID and NF- κ B activity, it would be important to assess phenotype, function and activation of CD8 T cells exposed to KRAS^{MUT} versus KRAS^{WT} to understand if the increased in ACID is correlated with loss of functionality and/or differentiation towards an exhausted subset. In this regard and in relationship with the interesting results on the association with response to ICB therapy, it would be important to assess the expression of different co-inhibitory receptors or at least PD1.*

Reply 4: We thank the Reviewer for these excellent questions. Based on the Reviewer’s advice, the exhausted phenotype, cytotoxicity function and activation of CD8⁺ T cells from KRAS^{MUT} versus KRAS^{WT} tumors were assessed. As tested by flow cytometric analysis, PD-1, a key marker related to exhaustion, was comparably expressed in CD8⁺ T cells from KRAS^{MUT} versus KRAS^{WT} tumors (**Supplementary Fig. 2H**). A similar expression was seen for another exhaustion-related marker TIGIT (**Supplementary Fig. 2H**). In addition, the expression levels of T-cell activation markers CD25 and CD69 were well balanced between the two groups (**Supplementary Fig. 2H**). With regard to the cytotoxicity function, flow cytometric analysis showed that CD8⁺ T cells from KRAS^{MUT} tumors had a significant increase in the expression of CD107a and perforin, markers associated with cytotoxic activity, as compared to those of KRAS^{WT} tumors (**Supplementary Fig. 2G**). Therefore, we concluded that the increase in ACID is not correlated with differentiation towards an exhausted subset, but appeared to indicate an impaired antitumor immunity. Further support for this possibility comes from a study by Song et al in which it is demonstrated that preventing AICD could enhance the antitumor immunity of adoptive T cells in a breast cancer patient-derived xenograft model (Nat Immunol. 2018;19(10):1112-1125.). Based on the Reviewer’s comments, these results were incorporated in the revised manuscript, as follows:

(Manuscript, Results, Page 8, line 182-190)

“In addition to AICD increase in KRAS^{MUT} tumors, we found that tumor-specific CTLs from KRAS^{MUT} versus KRAS^{WT} tumors exhibited a significant increase in the expression of perforin and CD107a (**Supplementary Fig. 2G**), markers associated with cytotoxic activity. Yet the markers of exhaustion (PD-1 and TIGIT) and activation (CD25 and CD69) were comparably expressed between the two groups (**Supplementary Fig. 2H**). These findings suggested the increase in ACID was not correlated with differentiation towards an exhausted subset, but appeared to indicate an impaired antitumor immunity.”

Comment 5: *The author showed that MCT1 blockade by AZD3965 and GPR81 inhibition by 3-OBA significantly reversed the effects of lactic acid on NF- κ B/AICD axis. Does this have a functional effect on CD8 anti-tumor immunity? In other words, would AZD3965 boost the activity of tumor-specific CTLs and impact on their anti-tumor potential against CRC KRAS^{MUT} tumors?*

Reply 5: Thank you very much. In light of the Reviewer's comments, we assessed the effects of AZD3965 on the cytotoxic activity of tumor-specific CTLs. Previous study showed significant impacts of MCT1 deficiency on the proliferation and activation of CD8⁺ T cells (iScience. 2022; 25(6): 104435.), whereas in vitro experiments here demonstrated administration of AZD3965 to tumor-specific CTLs did not affect the expression of CD107a and perforin, markers associated with cytotoxic activity (See **Figure 6A# below**).

Mauro et al found that lactic acid could cause the loss of the cytolytic function of CD8⁺ T cells (PLoS Biol. 2015; 13(7): e1002202.). In line with this, administration in vitro of lactic acid to tumor-specific CTLs elicited a significant decrease in the expression of the cytotoxicity markers (See **Figure 6B# below**), which was effectively blocked by AZD3965 (See **Figure 6B# below**). Together, these findings suggested the effects of AZD3965 on the cytotoxic activity of tumor-specific CTLs depended on the presence of lactic acid. Further support for this conclusion comes from another previous study by Kreutz and colleagues (Blood. 2007;109(9):3812-9.).

Figure 6#. (A) Immunoblots showing the expression levels of CD107a and perforin in CRC-derived tumor-specific CTLs with or without AZD3965 treatment. Ut, tumor-specific CTLs without any treatment. (B) Western blots showing expression levels of CD107a and perforin in CRC-derived tumor-specific CTLs treated with AZD3965 in combination with PBS or lactic acid. β -actin served as loading controls.

Comment 6: *Results on circATXN7 expression confined to the tumor stroma and specifically to the CD8 compartment are very intriguing. However, the RT-PCR approach needs to be integrated with other methodologies to look at circATXN7 protein expression on the different cell compartment (i.e. IHC on tumor section looking at the co-localization circATXN7 with CD8 or CD4 or other immune cells or stromal cells). Also, in order to conclude that circATXN7 expression is specific to the tumor and confined to the stroma, a normal adjacent tissue should be included as control as well as assess the expression of circATXN7 in peripheral CD8 T cells.*

Reply 6: We appreciate the Reviewer's comments. As suggested by the Reviewer, circATXN7 FISH was co-stained with markers indicative of tumor cells or immune cells. Co-staining with EpCAM showed that circATXN7 expression was confined to the tumor stroma (**Supplementary Fig. 7H**). Co-staining with CD8, or CD4 found that circATXN7 expression was specifically to the CD8 compartment (**Supplementary Fig. 7H**). To further exclude circATXN7 expression on other cell compartment, we sorted each tumor infiltration cell types, which were then subjected to RT-PCR. Results from RT-PCR showed other components including CD4, macrophages, endothelial cells, and fibroblasts displayed negligible expression of circATXN7 (**Supplementary Fig. 7F**), which were further confirmed by qRT-PCR (**Supplementary Fig. 7G**).

Also, circATXN7 ISH staining (**Supplementary Fig. 7D**) as well as co-staining with CD8 (**Supplementary Fig. 7E**) in normal adjacent tissues demonstrated circATXN7 were barely expressed. Additional analysis by qRT-PCR indicated peripheral CD8⁺ T cells had only a slight expression of circATXN7, as compared to tumor infiltrated CD8⁺ T cells (**Supplementary Fig. 5I**). Therefore, these results suggested circATXN7 expression was specific to the tumor and confined to the stroma. These results were incorporated in the revised manuscript, as follows:

(Manuscript, Results, Page 12, line 310-311)

“whereas circATXN7⁺ cells were absent in normal adjacent tissues (**Supplementary Fig. 7D, Supplementary Fig. 7E**)”

(Manuscript, Results, Page 13, line 320-321)

“These results were confirmed by circATXN7 FISH co-stained with CD8, CD4, or EpCAM (**Supplementary Fig. 7H**).”

(Manuscript, Results, Page 10, line 256-258)

“As compared to the tumor infiltrated CD8⁺ T cells, only a slight circATXN7 expression was detected in peripheral CD8⁺ T cells (**Supplementary Fig. 5I**).”

Comment 7: *Figure 5. Is thymic development, differentiation and frequencies of CD8 T cells normal in circAtxn7CKO mice? Is circAtxn7 deficiency impacting on T cell biology?*

Reply 7: We truly thank the Reviewer's comments. As described previously, the tool mice E8I-Cre used for cross breeding have Cre activity observed in CD8 α ⁺CD8 β ⁺ $\alpha\beta$ T cells and CD8 α ⁺CD8 β ⁻ $\alpha\beta$ T cells, but not in CD4⁺CD8 α ⁻CD8 β ⁻ $\alpha\beta$ T cells, thus avoiding off-target effects on CD4⁺ T cells (Immunity. 2021;54(10):2209-2217.e6.). As determined by flow cytometry analysis, the absolute numbers of thymocytes and peripheral CD4⁺ and CD8⁺ T cells were unaltered upon ablation of circAtxn7 (**Supplementary Fig. 10D**), as were the frequencies of their corresponding subsets (**Supplementary Fig. 10E-H**).

Thus, circAtxn7 takes no effects on T-cell development, differentiation and frequencies. Next, we stimulated splenic CD8⁺ T cells with α CD3/CD28 in vitro and found that the expression of activation markers (CD25 and CD69) and proliferation (assessed by CFSE) was comparable between WT or circAtxn7 CKO mice (See **Figure 7# below**), indicating that circAtxn7 deficiency appears to not impact on T cell biology. According to the Reviewer's comments, we have added the corresponding statement to the revised manuscript, as follows:

(Manuscript, Results, Page 16 line 434 - Page 15 line 438)

“Moreover, the numbers of thymocytes and peripheral CD4⁺ and CD8⁺ T cells were unchanged upon ablation of *circAtxn7*^{CKO} mice (**Supplementary Fig. 10D**), as were the frequencies of their corresponding subsets (**Supplementary Fig. 10E-H**). Therefore, circAtxn7 appears to be dispensable for mouse T-cell development.”

Figure 7#. CD8⁺ T cells were purified from the spleens of WT and circAtxn7 CKO mice and stimulated with α CD3/CD28 for 36 h. **(A)** The expression of activation markers (CD25 and CD69) was analyzed by flow cytometry (n=4). **(B)** Proliferation of CD8⁺ T cells was measured by CFSE dilution and analyzed by flow cytometry (n=4). Numbers (mean \pm SD) denote the percentage of cells undergoing at least one cellular division. ns indicates $p > 0.05$ compared with WT by 2-tailed Student's t test.

Comment 8: Figure 5B-C. Results of tumor inhibition after targeting circAtxn7 are solid and reproduced in several different models. However, experiments are stopped when the tumor are very small, around 500mm³ and sometime even at 300mm³. I believe the author should keep the exp going at least until 1000mm³ to claim a possible curative role for circAtxn7 targeting. Also, author should try to use the same (or similar) scale for comparison purposes. Same applies to the other tumor models used in the manuscript.

Reply 8: Thank you for your comments. In light of the Reviewer's suggestion, we have optimized the animal experiments to better illustrate the *in vivo* tumor inhibition effect of targeting circAtxn7 and the same scale was used for comparison purposes. The new results were incorporated in the revised manuscript (**Fig. 5B-C, Supplementary Fig. 11A-D**).

Comment 9: *Figure 5K. Can the author show other T cell features that can be correlated with immunosurveillance (i.e. less exhausted phenotype, increased frequencies of infiltrating T cells)?*

Reply 9: Many thanks for the constructive suggestion. Based on the Reviewer's advice, T cell features including exhausted phenotype and cytotoxicity function were further assessed. Results demonstrated that *circAtxn7* deficiency did not affect the exhausted phenotype (**Supplementary Fig. 11L**), but significantly upregulated the expression of perforin and CD107a, markers related to cytotoxic activity (**Supplementary Fig. 11K**). With regard to the frequencies of infiltrating T cells, IHC staining for CD8 in MC38K xenografts in WT or *circAtxn7* CKO mice showed a substantial increase in tumor-infiltrating CD8⁺ T cell density in MC38K tumors from *circAtxn7* CKO mice than those from WT littermates (**Fig. 5I**), which was further confirmed by flow cytometry analysis (**Fig. 5J**). These results were incorporated in the revised manuscript, as follows:

(Manuscript, Results, Page 18, line 484-489)

“More importantly, we found a substantial increase in tumor-infiltrating CD8⁺ T cell density in MC38K tumors from *circAtxn7*^{CKO} mice than those from WT littermates (**Fig. 5I-J**), as well as in cytotoxic cytokine IFN-γ production (**Fig. 5K**) and the expression of perforin and CD107a, markers related to cytotoxic activity (**Supplementary Fig. 11K**), but no significant effects on the exhausted phenotype (**Supplementary Fig. 11L**).”

Comment 10: *Figure 6. I would suggest the author to include data on the characterization (function, activation, phenotype...) of OT1 upon circAtxn7 silencing prior and post infusion in tumor-bearing mice?*

Reply 10: Thank you. As suggested by the Reviewer, the characterization including exhausted phenotype, cytotoxicity function and activation was further tested for OT1 upon *circAtxn7* silencing prior and post infusion in tumor-bearing mice. Flow cytometric analysis that prior infusion in tumor-bearing mice, *circAtxn7* silencing did not affect the exhausted phenotype and activation (**Supplementary Fig. 12H-I**), but significantly upregulated the expression of perforin and CD107a, markers related to cytotoxic activity (**Supplementary Fig. 12J**). A similar pattern was found when post infusion in tumor-bearing mice (**Supplementary Fig. 12M**). In light of the Reviewer's comments, these results were incorporated in the revised manuscript, as follows:

(Manuscript, Results, Page 19, line 516-522)

“*In vitro* experiments demonstrated that although *circATXN7* silencing in OT-I cells did not have significant effects on their proliferation (**Supplementary Fig. 12F**), migration (**Supplementary Fig. 12G**), exhausted phenotype (**Supplementary Fig. 12H**) or activation (**Supplementary Fig. 12I**), it increased the expression of perforin

and CD107a, markers related to cytotoxic activity (**Supplementary Fig. 12J**) and NF- κ B activation (**Supplementary Fig. 12K**), but decreased AICD sensitivity (**Supplementary Fig. 12L**).”

(Manuscript, Results, Page 20, line 529-534)

“At endpoint, loss of circAtxn7 did not alter the transferred cells’ exhausted phenotype and activation (**Supplementary Fig. 12M**), but endowed the tumors with substantially increased CTL densities (**Fig. 6F**) and increased the expression of perforin and CD107a, markers related to cytotoxic activity (**Supplementary Fig. 12M**), which correlated with improved circAtxn7-silenced T cell antitumor activities (**Fig. 6G-H**).”

Comment 11: *INTRODUCTION: author should make an effort to write a more compelling introduction for their study. At the moment it is very poor and does not place their study in the context of the available literature.*

Reply 11: We thank the Reviewer for the helpful comments. Based on the Reviewer’s comments, the following sentences were rephrased in the “**Introduction**” section of the revised manuscript, as follows:

(Manuscript, Introduction, Page 4 line 60 - Page 6 line 128)

“The human KRAS protein is both friend and foe; the non-mutated form is indispensable in diverse physiological processes, whereas the mutated versions directly underlie multistep processes of tumorigenesis and progression in ~30% of all cancers. Targeting KRAS is considered one of the optimal strategies to combat KRAS-driven tumors and improve advanced cancer patients’ outcomes^{1, 2}. Despite advances in KRAS inhibitors, decades of efforts hitherto did not bring them to the clinic^{3, 4}. Recent studies revealed that mutant KRAS could be exploited by cancers to orchestrate an immune-suppressive tumor microenvironment (TME)^{5, 6}. The Cancer Genome Atlas also indicated KRAS mutant colorectal cancer (CRC) are closely associated with decreased immune infiltration and reactivity⁷. In addition, KRAS inhibition endowed tumors with a remarkable increase in anti-tumour immunity⁸. Therefore, KRAS mutant tumors are especially immune-excluded, and therapeutic approaches aimed at activating antitumor immune program might be essential to eliminate the disease.

Cancer pathologies are often orchestrated by various metabolites, and KRAS mutant tumors are especially exposed to dramatically increased levels of lactic acid^{9, 10}. Cancer-generated lactic acid endows malignancies with an acidic TME, and also acts as a primary carbon fuel source and signaling molecule involved in oncogenic pathways^{11, 12}. Current researches have also yielded evidence that lactic acid in the TME was an impediment towards providing an effective antitumor immunity¹³. In this respect, tumor-derived lactic acid was found to take effects on tumor-associated macrophages^{14, 15}, regulatory T cells¹⁶, myeloid-derived suppressor cells¹⁷, natural killer cells¹⁸, or dendritic cells¹⁹. In particular, Kreutz and colleagues pointed to an impact of lactic acid

on cytolytic T lymphocytes (CTLs)²⁰, which directly identify and destroy nascent tumor cells during cancer immunosurveillance. Our previous study significantly advanced our understanding for the involvement of lactic acid in CTL fate decisions and subsequent support for tumor progress⁹. These insights highlighted an intense engagement between lactic acid and CTLs, but the intracellular mechanism of lactic acid action in CTLs remains poorly defined.

Activation-induced cell death (AICD), firstly described in 1987, has been characterized as a mechanistic link with immunological homeostasis^{21, 22}. Under physiological conditions, AICD is able to eradicate activated T lymphocytes presumed to be no longer required²³. Abnormality in AICD was discovered in diverse pathological situations, such as viral infection, inflammatory and autoimmune disorders^{24, 25, 26}. In the context of many cancer types, AICD deregulation was also frequently identified^{27, 28, 29}. Aberrant AICD of tumor-specific CTLs can be used by cancers to evade immune elimination²⁸, which accounts for the paradoxical fact that, although the patients mount a specific T-cell response against neoplasm, these CTLs fail to control the disease. It is now widely understood that AICD is of much value to decipher cancer pathologies as well as present prognostic insights, or even develop alternative treatments for cancer patients. Along this line, we previously found mutant KRAS-expressing CRC cells exploited tumor-derived lactic acid to sensitize tumor-specific CTLs to AICD, thereby fostering tumor immune escape and immunotherapy resistance⁹. Multiple molecular players, including mitochondrio-nuclear translocation of AIF³⁰, CD158 receptor³¹, or NKILA²⁸, were identified to participate in an abnormal sensitivity of tumor-specific CTLs to AICD. Despite this knowledge, how lactic acid reprograms AICD of tumor-specific CTLs warrants under further investigation.

Circular RNAs (circRNAs) emerge as a unique class of RNA molecules characterized by their covalently closed ring structure. The interest in studying circRNAs is raised because of several peculiar features, such as evolutionary conservation and tissue-specific expression, but above all, because their deregulated expression was linked to many pathological conditions, particularly cancers^{32, 33}. Mounting data suggest these molecules are of potential clinical relevance and utility^{34, 35}. Notably, circRNAs have been identified to be participants in the regulatory networks of tumor immunity³⁶. Wang and colleagues demonstrated overexpression of hsa_circ_0020397 in CRC cells could promote the upregulation of PD-L1 by binding and inhibiting miR-138 expression, thereby resulting in tumor immune escape³⁷. Furthermore, there is evidence of a correlation between circRNAs and immune cell infiltration in several cancers^{38, 39, 40}. Recently, a study by Ye et al identified circRNA profiles and regulatory networks in melanoma patients treated with immune checkpoint blockades, highlighting the clinical application potential of circRNAs as predictive biomarkers for immunotherapeutic efficacy⁴¹. These advances underscored the link between circRNAs and cancer immunology, yet knowledge of the role played by circRNAs and the mechanism of circRNAs' action in CTLs is limited. Using circRNA sequencing and CD8-conditional circRNA knockout mice, this work set an example of how circRNAs regulate AICD of

CTLs and subsequently influence immunotherapy.”

Comment 12: *Line 367 and 371 (and somewhere else in the text): “cool” tumors, I guess the authors mean “Cold” tumors.*

Reply 12: Many thanks for the Reviewer’s carefulness. The typo was corrected in the revised manuscript, and we apologized for this mistake.

Comment 13: *The rationale of choosing to study CirRNAs in the context of anti-tumor immunity should be better detailed to make the study more accessible to both expert and non-specialist in the field.*

Reply 13: Thank you. Based on the Reviewer’s comments, the following sentences were rephrased in the revised manuscript, as follows:

(Manuscript, Introduction, Page 5 line 110 - Page 6 line 128)

“Circular RNAs (circRNAs) emerge as a unique class of RNA molecules characterized by their covalently closed ring structure. The interest in studying circRNAs is raised because of several peculiar features, such as evolutionary conservation and tissue-specific expression, but above all, because their deregulated expression was linked to many pathological conditions, particularly cancers^{32, 33}. Mounting data suggest these molecules are of potential clinical relevance and utility^{34, 35}. Notably, circRNAs have been identified to be participants in the regulatory networks of tumor immunity³⁶. Wang and colleagues demonstrated overexpression of hsa_circ_0020397 in CRC cells could promote the upregulation of PD-L1 by binding and inhibiting miR-138 expression, thereby resulting in tumor immune escape³⁷. Furthermore, there is evidence of a correlation between circRNAs and immune cell infiltration in several cancers^{38, 39, 40}. Recently, a study by Ye et al identified circRNA profiles and regulatory networks in melanoma patients treated with immune checkpoint blockades, highlighting the clinical application potential of circRNAs as predictive biomarkers for immunotherapeutic efficacy⁴¹. These advances underscored the link between circRNAs and cancer immunology, yet knowledge of the role played by circRNAs and the mechanism of circRNAs’ action in CTLs is limited. Using circRNA sequencing and CD8-conditional circRNA knockout mice, this work set an example of how circRNAs regulate AICD of CTLs and subsequently influence immunotherapy.

Comment 14: *Figure 2N: What is the viability of cells treated with increasing concentration of lactic acid, especially at 10 mM- which is the concentration used in the exp to determine the link between Lactic acid and circATXN7?*

Reply 14: We appreciate the Reviewer’s comments. As suggested by the Reviewer, we

assess the effects of 10 mM lactic acid on cell viability. In line with previous findings by Mauro et al (PLoS Biol. 2015; 13(7): e1002202.), 10 mM lactic acid did not affect cellular viability (See Figure 8# below).

Figure 8#. Representative Annexin V-FITC staining of Day-1 T cells treated with 10 mM lactic acid (n=3). Numerical values denote annexin V⁺ cell percentages (mean ± SD). ns indicates $p > 0.05$ compared with PBS by one-way ANOVA.

Comment 15: Line 129-131: the authors state: “Results found MCT1 blockade by AZD3965, but not GPR81 inhibition by 3-OBA, significantly reversed the effects of lactic acid on NF- κ B/AICD axis”. To my understanding there is a significant effect also when 3-OBA is used; thus I would suggest the author to temper their conclusion and rephrase the concept.

Reply 15: We appreciate the Reviewer’s comments. We are sorry for the confusion, and to make it more clearly, the following sentences were rephrased in the revised manuscript, as follows:

(Manuscript, Results, Page 9, line 209-225)

“MCT1 blockade by AZD3965 significantly reversed the effects of lactic acid on NF- κ B/AICD axis, and its combo inhibition of GPR81 with 3-OBA was obviously better than AZD3965 alone (Supplementary Fig. 3F-G). To distinguish their contribution to the difference in NF- κ B/AICD axis between KRAS^{MUT} versus KRAS^{WT} tumors, the expression levels of MCT1 and GPR81 were further assessed. Results found that MCT1 had significantly higher expression abundance than GPR81 in tumor-specific CTLs from both KRAS^{MUT} and KRAS^{WT} tumors (Supplementary Fig. 3H). Moreover, the expression levels of MCT1, but not GPR81, was positively associated with NF- κ B activity (Supplementary Fig. 3I-J), and NF- κ B activity correlated well with intracellular lactic acid concentration in tumor-specific CTLs of KRAS^{MUT} tumors (Supplementary Fig. 3K), but not in those of KRAS^{WT} tumors (Supplementary Fig. 3L-N). More importantly, as the key downstream element in lactic acid/GPR81 axis, cAMP and TCF-1 in tumor-specific CTLs were well balanced between KRAS^{MUT} versus KRAS^{WT} tumors (Supplementary Fig. 3O-P). Taken together, these results suggested that MCT1-mediated lactic acid input, but not activating GPR81, contributed to the difference in NF- κ B/AICD axis between KRAS^{MUT} versus KRAS^{WT} tumors.”

Comment 16: *seminal results have been published from Steve Rosenberg and Eric Tran on the potential of using T-cell receptors (TCRs) targeting mutant KRAS G12D expressed by the tumors as a tool to increase efficacy of ACT for the treatment of tumor still refractory. Authors should cite these papers as well as others studies investigating the role KRAS mutation on anti-tumor T cell responses and they should make an effort to comment their results in the context of the available literature.*

Reply 16: Thank you very much for your suggestion. As suggested by the Reviewer, the corresponding papers were cited and the following sentences were rephrased in the revised manuscript, as follows:

(Manuscript, Discussion, Page 23 line 634 – Page 24 line 652)

“Therapeutic attempts to tackle KRAS^{MUT} have been continuing for decades. Due to the benefits of ACT in a subset of cancer patients, much interest is dedicated to the study of T cell receptors targeting KRAS^{MUT}^{66, 67}. Along this line, Rosenberg and colleagues⁶⁸ demonstrated the tumor regression of metastatic CRC after the administration of cytotoxic T cells targeting mutant KRAS G12D. A similar pattern in pancreatic cancer was showed in a recent study by Tran et al⁶⁹. These insights suggest that KRAS-driven tumors can be targeted efficiently by reprogramming immune program. A study by DePinho et al reinforced this therapeutic strategy by showing that inhibition of myeloid-derived suppressor cell recruitment could overcome resistance of tumors expressing KRAS^{G12D} to anti-PD-1 therapy⁴⁹. These therapeutics, however, require the expression of KRAS^{G12D} and cannot be used against non-G12C mutants. As such, efforts to seek approach that enables broad inhibition of KRAS^{MUT} or its related downstream signaling are continuing. Our work here identified a KRAS^{MUT}-activated circATXN7 program as an exploitable therapeutic approach to combat KRAS^{MUT} tumors, which did not correlate with the KRAS mutation type and appeared to be a general feature of KRAS^{MUT} tumors. In vitro and in vivo experiments showed targeting circATXN7 in T cells protected T cells from tumor-mediated AICD. Accordingly, circATXN7 ablation shifts KRAS^{MUT} tumors from immunologically “cold” to “hot” and consequently improves immunotherapeutic efficacy.”

REVIEWER COMMENTS

Reviewer #1 (Remarks to the Author):

All my comments have been addressed.

Reviewer #2 (Remarks to the Author):

The revised manuscript is improved considerably, with the addition of a large amount of new data. While most issues have been appropriately addressed, some questions remain need to be solved.

Question 2:

The authors have quantified the copy numbers of circATXN7. Results showed that 926.8 ± 148.2 and 80.8 ± 26.0 circATXN7 molecules in per CD8+ T cell from KRASMUT and KRASWT tumors, respectively. Considering that P65 known as a very abundant protein in CD8+ T cell, and >180,000 copies of P65 protein per HeLa cell were estimated (PMID: 26496610). The authors should quantify the copy number of P65 protein (rather than P65 RNA) in CD8+ T cells carefully as described in previous papers (PMID: 29706547, PMID: 33436560, and PMID: 26496610), and discuss whether the copy number of circATXN7 and P65 protein support the proposed models?

Reviewer #3 (Remarks to the Author):

The authors have addressed all the concerns raised by me and two other reviewers in the resubmitted manuscript. I think the current version is now acceptable.

Reviewer #4 (Remarks to the Author):

This reviewer thanks the authors for their significant efforts in addressing the concerns raised in the initial review by new experiments and clarifications. The manuscript now reports an important discovery in the role of circATXN7 regulating CD8+ T-cells. The results have the potential to affect immune-based therapies and are of high significance to the field.

1 **Point-by-point responses to the comments from the Reviewers**

2
3 We sincerely thank all of the Reviewers for their constructive comments and helpful
4 suggestions. We have addressed all of the raised issues at our best efforts, and hope that
5 our revised manuscript now meets your expectations.
6

7
8 =====
9 **Responses to comments (Reviewer #1)**

10 *All my comments have been addressed.*

11
12 **Reply:** Many thanks for the Reviewer's encouragement and positive comments!
13

14
15 =====
16 **Responses to comments (Reviewer #2)**

17 *The revised manuscript is improved considerably, with the addition of a large amount*
18 *of new data. While most issues have been appropriately addressed, some questions*
19 *remain need to be solved.*

20 *Question 2:*

21 *The authors have quantified the copy numbers of circATXN7. Results showed that 926.8*
22 *± 148.2 and 80.8 ± 26.0 circATXN7 molecules in per CD8+ T cell from KRASMUT and*
23 *KRASWT tumors, respectively. Considering that P65 known as a very abundant protein*
24 *in CD8+ T cell, and >180,000 copies of P65 protein per HeLa cell were estimated*
25 *(PMID: 26496610). The authors should quantify the copy number of P65 protein*
26 *(rather than P65 RNA) in CD8+ T cells carefully as described in previous papers*
27 *(PMID: 29706547, PMID: 33436560, and PMID: 26496610), and discuss whether the*
28 *copy number of circATXN7 and P65 protein support the proposed models?*

29
30 **Reply:** We appreciate the reviewer for raising this important question. In our previous
31 version, we showed that p65 was ~1000 copies at the RNA level in each tumor-specific
32 cytotoxic T lymphocyte (CTL). Mann et al demonstrated a comparatively different copy
33 number between transcripts and proteins (Cell Metab. 2014;20(6):1076-87.). Moreover, only a
34 low amount of the proteins had a significant correlation with the cognate RNA (Nat
35 Commun. 2022;13(1):7389.). As such, we fully agree with the reviewer that it is an important
36 point to estimate whether the p65 copies at the protein level can support the proposed
37 models.
38

39 Based on the reviewer's advice, we further estimated the protein abundance of p65 in
40 several cell types. To this end, the key cell types in tumor microenvironment were sorted,
41 including primary tumor cells, CD4⁺ T cells, CD8⁺ T cells, macrophages, fibroblasts,
42 NK cells, and tumor specific CTL. The results showed different cells have different
43 protein copies of p65, and that the protein copies of p65 in each CD4⁺ T cells, CD8⁺ T
44 cells, macrophages, fibroblasts, NK cells, primary tumor cells and tumor-specific CTL
45 were ~4800, ~4200, ~62,900, ~25,900, ~95,100, ~143,000 and ~3600, respectively

46 (See Figure 1# below). These findings suggested that the protein copy number was
47 cell context-dependent, but not a general feature in different types of cells.

48

49 One protein might have different abundance in different cells. Further support for this
50 possibility comes from previous studies (Nat Commun. 2021;12(1):295; Cell. 2018;173(4):906-
51 919.e13.) in which they are demonstrated that each A549 cell, and VSV-infected
52 macrophage contained 50, ~1000 copies of RIG-I, respectively. On the other hand, the
53 protein copy number range can span several orders of magnitude in one specific type
54 of cell. It has been reported that the protein copies per HeLa cell vary from 3 to > 80,
55 000, 000 (Cell. 2015;163(3):712-23.). Together, these results from published literatures as
56 well as our findings herein indicated that protein copies exhibit a cell-type specific
57 expression pattern.

58

59 Our previous revision had estimated the copies of circATXN7 in the whole indicated
60 cells. Considering the cytoplasmic localization of circATXN7 (Supplementary Fig.
61 6J), the copies of circATXN7 and p65 in the cytoplasm of each tumor-specific CTL
62 were further quantified to estimate whether the stoichiometry of these molecules could
63 support our proposed models. Results demonstrated that the cytoplasm of each tumor-
64 specific CTL from KRAS^{MUT} tumors contained 1072.4 ± 676.3 and 1978.4 ± 1122.3
65 copies of circATXN7 and p65 protein (Supplementary Fig. 9G-H), which would allow
66 for an approximately equimolar interaction. However, circATXN7 was 33.8 ± 20.3
67 copies in the cytoplasm of each tumor-specific CTL from KRAS^{WT} tumors, which was
68 significantly lower than p65 (536.1 ± 171.5 copies per cell; Supplementary Fig. 9 G-
69 H). As reported, circRNA-protein interaction is a common event (Cell. 2020;183(1):76-93.e22;
70 Proc Natl Acad Sci U S A. 2023;120(13):e2215132120; Nat Commun. 2022;13(1):7243.). In this study,
71 considering the stoichiometry of circATXN7 versus p65 and the fact that each
72 circATXN7 contains one p65-binding motif (Fig. 4H), we concluded tumor-specific
73 CTLs of KRAS^{MUT} tumors, but not those of KRAS^{WT} tumors, have sufficient
74 circATXN7 to directly bind p65 for inhibition. In light of the Reviewer's comments,
75 the following sentences have been rephrased in the revised manuscript, as follows:

76

77 (Manuscript, Results, Page 14 line 377- Page 15 line 388)

78 "Considering the cytoplasmic localization of circATXN7 (Supplementary Fig. 6J),
79 the copies of circATXN7 and p65 in the cytoplasm of each tumor-specific CTL were
80 further quantified. Results demonstrated that the cytoplasm of each tumor-specific CTL
81 from KRAS^{MUT} tumors contained 1072.4 ± 676.3 and 1978.4 ± 1122.3 copies of
82 circATXN7 and p65 protein, which would allow for an approximately equimolar
83 interaction (Supplementary Fig. 9G-H). However, circATXN7 was 33.8 ± 20.3 copies
84 in the cytoplasm of each tumor-specific CTL from KRAS^{WT} tumors, which was
85 significantly lower than p65 (536.1 ± 171.5 copies per cell; Supplementary Fig. 9G-
86 H). On the basis of the stoichiometry of circATXN7 versus p65 and the fact that each
87 circATXN7 contains one p65-binding motif, we concluded that tumor-specific CTLs
88 of KRAS^{MUT} tumors, but not those of KRAS^{WT} tumors, had sufficient circATXN7 to
89 directly bind p65 for inhibition."

90
91
92
93
94
95
96
97
98
99
100
101
102
103
104
105
106
107
108
109

(Manuscript, Discussion, Page 23 line 627- Page 24 line 643)

“The interaction between proteins and circRNAs can be often seen in the current literatures^{66, 67}. For instance, Guarnerio and colleagues found that circCsnk1g3 and circAnkib1 can interact with RIG-I at a close molar ratio in the sarcoma cells³⁸. The present study proposed a model in which circATXN7 directly binds with p65 in the tumor-specific CD8⁺ T cells of KRAS^{MUT} CRC. Furthermore, the stoichiometry of the circATXN7 versus that of p65 indicated that the interaction between circATXN7 and p65 was approximately equimolar. Although Mann et al estimated that each HeLa cell contained >180,000 copies of p65⁶⁸, this work demonstrated p65 protein was expressed in the tumor-specific CD8⁺ T cells of KRAS^{MUT} CRC at ~2000 copies per cell. These findings suggested a cell-type specific protein expression pattern. One protein might have different abundance in different cells. Additional support for this possibility comes from previous studies^{69, 70} in which they are demonstrated that each A549 cell, and VSV-infected macrophage contained 50, ~1000 copies of RIG-I, respectively. On the other hand, the protein copy number range can span several orders of magnitude in one specific type of cell⁷¹. It has been reported that the protein copies per HeLa cell vary from 3 to > 80, 000, 000⁶⁸. These findings confirmed the protein abundance was cell context-dependent, but not a general feature in different types of cells.”

110
111
112
113
114
115

Figure 1# Measurement of the copy number of p65 protein in indicated cells. Purified recombinant p65 protein was used to generate standard curves to estimate the mass of p65 in cell lysate of CD4⁺ T cells (1×10⁵), CD8⁺ T cells (1×10⁵), tumor-specific CTL (1×10⁵), macrophages (1×10⁴), fibroblasts (1×10⁴), NK cells (1×10⁴), and primary tumor cells (1×10⁴).

116
117
118
119
120
121

=====
Responses to comments (Reviewer #3)

The authors have addressed all the concerns raised by me and two other reviewers in the resubmitted manuscript. I think the current version is now acceptable.

Reply: We truly thank the Reviewer for the encouragement and comments on our work.

122

123

124 =====

125 **Responses to comments (Reviewer #4)**

126 *This reviewer thanks the authors for their significant efforts in addressing the concerns*
127 *raised in the initial review by new experiments and clarifications. The manuscript now*
128 *reports an important discovery in the role of circATXN7 regulating CD8+ T-cells. The*
129 *results have the potential to affect immune-based therapies and are of high significance*
130 *to the field.*

131

132 **Reply:** We sincerely thank you for your encouragement and helpful suggestions.

REVIEWERS' COMMENTS

Reviewer #2 (Remarks to the Author):

All my comments have been addressed. I think the current version is now acceptable.

1 **Point-by-point responses to the comments from the Reviewers**

2

3 We sincerely thank all of the Reviewers for their constructive comments and helpful
4 suggestions. We have addressed all of the raised issues at our best efforts, and hope that
5 our revised manuscript now meets your expectations.

6

7

8 =====

9 **Responses to comments (Reviewer #2)**

10 *All my comments have been addressed. I think the current version is now acceptable.*

11

12 **Reply:** Many thanks for the Reviewer's encouragement and positive comments! We
13 feel that these revisions have substantially strengthened our paper, and we are very
14 appreciative of your time and effort.